# (Almost) No Label No Cry

**Giorgio Patrini**[1,2], **Richard Nock**[1,2], **Paul Rivera**[1,2], **Tiberio Caetano**[1,3,4]

Australian National University[1], NICTA[2], University of New South Wales[3], Ambiata[4]

Sydney, NSW, Australia

`{name.surname}@anu.edu.au`

## Abstract

In Learning with Label Proportions (LLP), the objective is to learn a supervised classifier when, instead of labels, only label proportions for bags of observations are known. This setting has broad practical relevance, in particular for privacy preserving data processing. We first show that the mean operator, a statistic which aggregates all labels, is minimally sufficient for the minimization of many proper scoring losses with linear (or kernelized) classifiers *without* using labels. We provide a fast learning algorithm that estimates the mean operator via a manifold regularizer with guaranteed approximation bounds. Then, we present an iterative learning algorithm that uses this as initialization. We ground this algorithm in Rademacher-style generalization bounds that fit the LLP setting, introducing a generalization of Rademacher complexity and a Label Proportion Complexity measure. This latter algorithm optimizes tractable bounds for the corresponding bag-empirical risk. Experiments are provided on fourteen domains, whose size ranges up to ≈300K observations. They display that our algorithms are scalable and tend to consistently outperform the state of the art in LLP. Moreover, in many cases, our algorithms compete with or are just percents of AUC away from the Oracle that learns knowing all labels. On the largest domains, half a dozen proportions can suffice, *i.e.* roughly 40K times less than the total number of labels.

## 1   Introduction

Machine learning has recently experienced a proliferation of problem settings that, to some extent, enrich the classical dichotomy between *supervised* and *unsupervised learning*. Cases as multiple instance labels, noisy labels, partial labels as well as semi-supervised learning have been studied motivated by applications where fully supervised learning is no longer realistic. In the present work, we are interested in learning a binary classifier from information provided at the level of groups of instances, called *bags*. The type of information we assume available is the *label proportions per bag*, indicating the fraction of positive binary labels of its instances. Inspired by [1], we refer to this framework as Learning with Label Proportions (LLP). Settings that perform a bag-wise aggregation of labels include Multiple Instance Learning (MIL) [2]. In MIL, the aggregation is logical rather than statistical: each bag is provided with a binary label expressing an OR condition on all the labels contained in the bag. More general setting also exist [3] [4] [5].
Many practical scenarios fit the LLP abstraction. (a) Only aggregated labels can be obtained due to the physical limits of measurement tools [6] [7] [8] [9]. (b) The problem is semi- or unsupervised but domain experts have knowledge about the unlabelled samples in form of expectation, as *pseudo-measurement* [5]. (c) Labels existed once but they are now given in an aggregated fashion for privacy-preserving reasons, as in medical databases [10], fraud detection [11], house price market, election results, census data, etc. . (d) This setting also arises in computer vision [12] [13] [14].

**Related work.** Two papers independently introduce the problem, [12] and [9]. In the first the authors propose a hierarchical probabilistic model which generates labels consistent with the proportions, and make inference through MCMC sampling. Similarly, the second and its follower [6] offer a

variety of standard machine learning methods designed to generate self-consistent labels. [15] gives a Bayesian interpretation of LLP where the key distribution is estimated through an RBM. Other ideas rely on structural learning of Bayesian networks with missing data [7], and on K-MEANS clustering to solve preliminary label assignment in order to resort to fully supervised methods [13] [8]. Recent SVM implementations [11] [16] outperform most of the other known methods. Theoretical works on LLP belong to two main categories. The first contains uniform convergence results, for the estimators of label proportions [1], or the estimator of the mean operator [17]. The second contains approximation results for the classifier [17]. Our work builds upon their Mean Map algorithm, that relies on the trick that the logistic loss may be split in two, a convex part depending only on the observations, and a linear part involving a sufficient statistic for the label, the mean operator. Being able to estimate the mean operator means being able to fit a classifier without using labels. In [17], this estimation relies on a restrictive *homogeneity* assumption that the class-conditional estimation of features does not depend on the bags. Experiments display the limits of this assumption [11][16].

**Contributions.** In this paper we consider linear classifiers, but our results hold for kernelized formulations following [17]. We first show that the trick about the logistic loss can be generalized, and the mean operator is actually minimally sufficient for a wide set of "symmetric" proper scoring losses with no class-dependent misclassification cost, that encompass the logistic, square and Matsushita losses [18]. We then provide an algorithm, LMM, which estimates the mean operator via a Laplacian-based manifold regularizer without calling to the homogeneity assumption. We show that under a weak distinguishability assumption between bags, our estimation of the mean operator is all the better as the observations norm increase. This, as we show, cannot hold for the Mean Map estimator. Then, we provide a data-dependent approximation bound for our classifier with respect to the optimal classifier, that is shown to be better than previous bounds [17]. We also show that the manifold regularizer's solution is tightly related to the linear separability of the bags. We then provide an iterative algorithm, AMM, that takes as input the solution of LMM and optimizes it further over the set of consistent labelings. We ground the algorithm in a uniform convergence result involving a generalization of Rademacher complexities for the LLP setting. The bound involves a bag-empirical surrogate risk for which we show that AMM optimizes tractable bounds. All our theoretical results hold for any symmetric proper scoring loss. Experiments are provided on fourteen domains, ranging from hundreds to hundreds of thousands of examples, comparing AMM and LMM to their contenders: Mean Map, InvCal [11] and ∝SVM [16]. They display that AMM and LMM outperform their contenders, and sometimes even compete with the fully supervised learner while requiring few proportions only. Tests on the largest domains display the scalability of both algorithms. Such experimental evidence seriously questions the safety of privacy-preserving summarization of data, whenever accurate aggregates and informative individual features are available. Section (2) presents our algorithms and related theoretical results. Section (3) presents experiments. Section (4) concludes. A Supplementary Material [19] includes proofs and additional experiments.

## 2 LLP and the mean operator: theoretical results and algorithms

**Learning setting**   Hereafter, boldfaces like $\boldsymbol{p}$ denote vectors, whose coordinates are denoted $p_l$ for $l = 1, 2, ....$ For any $m \in \mathbb{N}_*$, let $[m] \doteq \{1, 2, ..., m\}$. Let $\Sigma_m \doteq \{\boldsymbol{\sigma} \in \{-1, 1\}^m\}$ and $\mathfrak{X} \subseteq \mathbb{R}^d$. Examples are couples (observation, label) $\in \mathfrak{X} \times \Sigma_1$, sampled i.i.d. according to some unknown but fixed distribution $\mathcal{D}$. Let $\mathcal{S} \doteq \{(\boldsymbol{x}_i, y_i), i \in [m]\} \sim \mathcal{D}_m$ denote a size-$m$ sample. In Learning with Label Proportions (LLP), we do not observe directly $\mathcal{S}$ but $\mathcal{S}_{|y}$, which denotes $\mathcal{S}$ with labels removed; we are given its partition in $n > 0$ bags, $\mathcal{S}_{|y} = \cup_j \mathcal{S}_j, j \in [n]$, along with their respective label proportions $\hat{\pi}_j \doteq \hat{\mathbb{P}}[y = +1|\mathcal{S}_j]$ and bag proportions $\hat{p}_j \doteq m_j/m$ with $m_j = \mathrm{card}(\mathcal{S}_j)$. (This generalizes to a cover of $\mathcal{S}$, by copying examples among bags.) The "bag assignment function" that partitions $\mathcal{S}$ is unknown but fixed. In real world domains, it would rather be known, *e.g.* state, gender, age band. A classifier is a function $h : \mathfrak{X} \to \mathbb{R}$, from a set of classifiers $\mathcal{H}$. $\mathcal{H}_L$ denotes the set of linear classifiers, noted $h_{\boldsymbol{\theta}}(\boldsymbol{x}) \doteq \boldsymbol{\theta}^\top \boldsymbol{x}$ with $\boldsymbol{\theta} \in \mathfrak{X}$. A (surrogate) loss is a function $F : \mathbb{R} \to \mathbb{R}_+$. We let $F(\mathcal{S}, h) \doteq (1/m) \sum_i F(y_i h(\boldsymbol{x}_i))$ denote the empirical surrogate risk on $\mathcal{S}$ corresponding to loss $F$. For the sake of clarity, indexes $i, j$ and $k$ respectively refer to examples, bags and features.

**The mean operator and its minimal sufficiency**   We define the (empirical) mean operator as:

$$\boldsymbol{\mu}_{\mathcal{S}} \quad \doteq \quad \frac{1}{m} \sum_i y_i \boldsymbol{x}_i \ . \tag{1}$$

---

**Algorithm 1** Laplacian Mean Map (LMM)

---

**Input** $\mathcal{S}_j, \hat{\pi}_j, j \in [n]; \gamma > 0$ (7); $\boldsymbol{w}$ (7); V (8); permissible $\phi$ (2); $\lambda > 0$;

Step 1 : let $\tilde{B}^{\pm} \leftarrow \arg\min_{X \in \mathbb{R}^{2n \times d}} \ell(L, X)$ using (7) (Lemma 2)

Step 2 : let $\tilde{\boldsymbol{\mu}}_{\mathcal{S}} \leftarrow \sum_j \hat{p}_j (\hat{\pi}_j \tilde{\boldsymbol{b}}_j^+ - (1 - \hat{\pi}_j) \tilde{\boldsymbol{b}}_j^-)$

Step 3 : let $\tilde{\boldsymbol{\theta}}_* \leftarrow \arg\min_{\boldsymbol{\theta}} F_\phi(\mathcal{S}_{|y}, \boldsymbol{\theta}, \tilde{\boldsymbol{\mu}}_{\mathcal{S}}) + \lambda \|\boldsymbol{\theta}\|_2^2$ (3)

**Return** $\tilde{\boldsymbol{\theta}}^*$

---

Table 1: Correspondence between permissible functions $\phi$ and the corresponding loss $F_\phi$.

| loss name | $F_\phi(x)$ | $-\phi(x)$ |
|---|---|---|
| logistic loss | $\log(1 + \exp(-x))$ | $-x \log x - (1-x)\log(1-x)$ |
| square loss | $(1-x)^2$ | $x(1-x)$ |
| Matsushita loss | $-x + \sqrt{1 + x^2}$ | $\sqrt{x(1-x)}$ |

The estimation of the mean operator $\boldsymbol{\mu}_{\mathcal{S}}$ appears to be a learning bottleneck in the LLP setting [17]. The fact that the mean operator is sufficient to learn a classifier without the label information motivates the notion of minimal sufficient statistic for features in this context. Let $\mathcal{F}$ be a set of loss functions, $\mathcal{H}$ be a set of classifiers, $\mathcal{I}$ be a subset of features. Some quantity $\boldsymbol{t}(\mathcal{S})$ is said to be a *minimal sufficient statistic* for $\mathcal{I}$ with respect to $\mathcal{F}$ and $\mathcal{H}$ iff: for any $F \in \mathcal{F}$, any $h \in \mathcal{H}$ and any two samples $\mathcal{S}$ and $\mathcal{S}'$, the quantity $F(\mathcal{S}, h) - F(\mathcal{S}', h)$ does not depend on $\mathcal{I}$ iff $\boldsymbol{t}(\mathcal{S}) = \boldsymbol{t}(\mathcal{S}')$. This definition can be motivated from the one in statistics by building losses from log likelihoods. The following Lemma motivates further the mean operator in the LLP setting, as it is the minimal sufficient statistic for a broad set of proper scoring losses that encompass the logistic and square losses [18]. The proper scoring losses we consider, hereafter called "symmetric" (SPSL), are twice differentiable, non-negative and such that misclassification cost is not label-dependent.

**Lemma 1** $\boldsymbol{\mu}_{\mathcal{S}}$ *is a minimal sufficient statistic for the label variable, with respect to* SPSL *and* $\mathcal{H}_L$.

([19], Subsection 2.1) This property, very useful for LLP, may also be exploited in other weakly supervised tasks [2]. Up to constant scalings that play no role in its minimization, the empirical surrogate risk corresponding to any SPSL, $F_\phi(\mathcal{S}, h)$, can be written with loss:

$$F_\phi(x) \quad \doteq \quad \frac{\phi(0) + \phi^\star(-x)}{\phi(0) - \phi(1/2)} \doteq a_\phi + \frac{\phi^\star(-x)}{b_\phi} \quad , \tag{2}$$

and $\phi$ is a *permissible* function [20, 18], *i.e.* $\mathrm{dom}(\phi) \supseteq [0, 1]$, $\phi$ is strictly convex, differentiable and symmetric with respect to $1/2$. $\phi^\star$ is the convex conjugate of $\phi$. Table 1 shows examples of $F_\phi$. It follows from Lemma 1 and its proof, that any $F_\phi(\mathcal{S}\boldsymbol{\theta})$, can be written for any $\boldsymbol{\theta} \equiv h_{\boldsymbol{\theta}} \in \mathcal{H}_L$ as:

$$F_\phi(\mathcal{S}, \boldsymbol{\theta}) \quad = \quad \frac{b_\phi}{2m}\left(\sum_i \sum_\sigma F_\phi(\sigma \boldsymbol{\theta}^\top \boldsymbol{x}_i)\right) - \frac{1}{2}\boldsymbol{\theta}^\top \boldsymbol{\mu}_{\mathcal{S}} \doteq F_\phi(\mathcal{S}_{|y}, \boldsymbol{\theta}, \boldsymbol{\mu}_{\mathcal{S}}) \quad , \tag{3}$$

where $\sigma \in \Sigma_1$.

**The Laplacian Mean Map (LMM) algorithm** The sum in eq. (3) is convex and differentiable in $\boldsymbol{\theta}$. Hence, once we have an accurate estimator of $\boldsymbol{\mu}_{\mathcal{S}}$, we can then easily fit $\boldsymbol{\theta}$ to minimize $F_\phi(\mathcal{S}_{|y}, \boldsymbol{\theta}, \boldsymbol{\mu}_{\mathcal{S}})$. This two-steps strategy is implemented in LMM in algorithm 1. $\boldsymbol{\mu}_{\mathcal{S}}$ can be retrieved from $2n$ bag-wise, label-wise unknown averages $\boldsymbol{b}_j^\sigma$:

$$\boldsymbol{\mu}_{\mathcal{S}} \quad = \quad (1/2)\sum_{j=1}^n \hat{p}_j \sum_{\sigma \in \Sigma_1} (2\hat{\pi}_j + \sigma(1 - \sigma))\boldsymbol{b}_j^\sigma \quad , \tag{4}$$

with $\boldsymbol{b}_j^\sigma \doteq \mathbb{E}_{\mathcal{S}}[\boldsymbol{x}|\sigma, j]$ denoting these $2n$ unknowns (for $j \in [n], \sigma \in \Sigma_1$), and let $\boldsymbol{b}_j \doteq (1/m_j)\sum_{\boldsymbol{x}_i \in \mathcal{S}_j} \boldsymbol{x}_i$. The $2n$ $\boldsymbol{b}_j^\sigma$s are solution of a set of $n$ identities that are (in matrix form):

$$B - \Pi^\top B^\pm \quad = \quad \mathbf{0} \quad , \tag{5}$$

where $\mathrm{B} \doteq [\boldsymbol{b}_1|\boldsymbol{b}_2|...|\boldsymbol{b}_n]^\top \in \mathbb{R}^{n\times d}$, $\Pi \doteq [\mathrm{DIAG}(\hat{\boldsymbol{\pi}})|\mathrm{DIAG}(\mathbf{1}-\hat{\boldsymbol{\pi}})]^\top \in \mathbb{R}^{2n\times n}$ and $\mathrm{B}^\pm \in \mathbb{R}^{2n\times d}$ is the matrix of unknowns:

$$\mathrm{B}^\pm \doteq \left[\underbrace{\boldsymbol{b}_1^{+1}|\boldsymbol{b}_2^{+1}|...|\boldsymbol{b}_n^{+1}}_{(\mathrm{B}^+)^\top}\middle|\underbrace{\boldsymbol{b}_1^{-1}|\boldsymbol{b}_2^{-1}|...|\boldsymbol{b}_n^{-1}}_{(\mathrm{B}^-)^\top}\right]^\top .\tag{6}$$

System (5) is underdetermined, unless one makes the homogeneity assumption that yields the Mean Map estimator [17]. Rather than making such a restrictive assumption, we regularize the cost that brings (5) with a manifold regularizer [21], and search for $\tilde{\mathrm{B}}^\pm = \arg\min_{\mathrm{X}\in\mathbb{R}^{2n\times d}}\ell(\mathrm{L},\mathrm{X})$, with:

$$\ell(\mathrm{L},\mathrm{X}) \doteq \mathrm{tr}\left((\mathrm{B}^\top - \mathrm{X}^\top\Pi)\mathrm{D}_{\boldsymbol{w}}(\mathrm{B}-\Pi^\top\mathrm{X})\right) + \gamma\,\mathrm{tr}\left(\mathrm{X}^\top\mathrm{LX}\right) ,\tag{7}$$

and $\gamma > 0$. $\mathrm{D}_{\boldsymbol{w}} \doteq \mathrm{DIAG}(\boldsymbol{w})$ is a user-fixed bias matrix with $\boldsymbol{w} \in \mathbb{R}_{+,*}^n$ (and $\boldsymbol{w} \neq \hat{\boldsymbol{p}}$ in general) and:

$$\mathrm{L} \doteq \varepsilon\mathrm{I} + \begin{bmatrix} \mathrm{L}_a & | & 0 \\ 0 & | & \mathrm{L}_a \end{bmatrix} \in \mathbb{R}^{2n\times 2n} ,\tag{8}$$

where $\mathrm{L}_a \doteq \mathrm{D} - \mathrm{V} \in \mathbb{R}^{n\times n}$ is the Laplacian of the bag similarities. V is a symmetric similarity matrix with non negative coordinates, and the diagonal matrix D satisfies $d_{jj} \doteq \sum_{j'} v_{jj'}, \forall j \in [n]$. The size of the Laplacian is $O(n^2)$, which is very small compared to $O(m^2)$ if there are not many bags. One can interpret the Laplacian regularization as smoothing the estimates of $\boldsymbol{b}_j^\sigma$ w.r.t the similarity of the respective bags.

**Lemma 2** *The solution $\tilde{\mathrm{B}}^\pm$ to $\min_{\mathrm{X}\in\mathbb{R}^{2n\times d}}\ell(\mathrm{L},\mathrm{X})$ is $\tilde{\mathrm{B}}^\pm = \left(\Pi\mathrm{D}_{\boldsymbol{w}}\Pi^\top + \gamma\mathrm{L}\right)^{-1}\Pi\mathrm{D}_{\boldsymbol{w}}\mathrm{B}$.*

([19], Subsection 2.2). This Lemma explains the role of penalty $\varepsilon\mathrm{I}$ in (8) as $\Pi\mathrm{D}_{\boldsymbol{w}}\Pi^\top$ and L have respectively $n$- and $(\geq 1)$-dim null spaces, so the inversion may not be possible. Even when this does not happen exactly, this may incur numerical instabilities in computing the inverse. For domains where this risk exists, picking a small $\varepsilon > 0$ solves the problem. Let $\tilde{\boldsymbol{b}}_j^\sigma$ denote the row-wise decomposition of $\tilde{\mathrm{B}}^\pm$ following (6), from which we compute $\tilde{\boldsymbol{\mu}}_S$ following (4) when we use these $2n$ estimates in lieu of the true $\boldsymbol{b}_j^\sigma$. We compare $\boldsymbol{\mu}_j \doteq \hat{\pi}_j\boldsymbol{b}_j^+ - (1-\hat{\pi}_j)\boldsymbol{b}_j^-$, $\forall j \in [n]$ to our estimates $\tilde{\boldsymbol{\mu}}_j \doteq \hat{\pi}_j\tilde{\boldsymbol{b}}_j^+ - (1-\hat{\pi}_j)\tilde{\boldsymbol{b}}_j^-$, $\forall j \in [n]$, granted that $\boldsymbol{\mu}_S = \sum_j \hat{p}_j\boldsymbol{\mu}_j$ and $\tilde{\boldsymbol{\mu}}_S = \sum_j \hat{p}_j\tilde{\boldsymbol{\mu}}_j$.

**Theorem 3** *Suppose that $\gamma$ satisfies $\gamma\sqrt{2} \leq ((\varepsilon(2n)^{-1}) + \max_{j\neq j'} v_{jj'})/\min_j w_j$. Let $\mathrm{M} \doteq [\boldsymbol{\mu}_1|\boldsymbol{\mu}_2|...|\boldsymbol{\mu}_n]^\top \in \mathbb{R}^{n\times d}$, $\tilde{\mathrm{M}} \doteq [\tilde{\boldsymbol{\mu}}_1|\tilde{\boldsymbol{\mu}}_2|...|\tilde{\boldsymbol{\mu}}_n]^\top \in \mathbb{R}^{n\times d}$ and $\varsigma(\mathrm{V},\mathrm{B}^\pm) \doteq ((\varepsilon(2n)^{-1}) + \max_{j\neq j'} v_{jj'})^2\|\mathrm{B}^\pm\|_F$. The following holds:*

$$\|\mathrm{M} - \tilde{\mathrm{M}}\|_F \leq \sqrt{n}\left(\sqrt{2}\min_j w_j^2\right)^{-1} \times \varsigma(\mathrm{V},\mathrm{B}^\pm) .\tag{9}$$

([19], Subsection 2.3) The multiplicative factor to $\varsigma$ in (9) is roughly $O(n^{5/2})$ when there is no large discrepancy in the bias matrix $\mathrm{D}_{\boldsymbol{w}}$, so the upperbound is driven by $\varsigma(.,.)$ when there are not many bags. We have studied its variations when the "distinguishability" between bags increases. This setting is interesting because in this case we may kill two birds in one shot, with the estimation of M *and* the subsequent learning problem potentially easier, in particular for linear separators. We consider two examples for $v_{jj'}$, the first being (half) the normalized association [22]:

$$v_{jj'}^{nc} \doteq \frac{1}{2}\left(\frac{\mathrm{ASSOC}(\mathcal{S}_j,\mathcal{S}_j)}{\mathrm{ASSOC}(\mathcal{S}_j,\mathcal{S}_j\cup\mathcal{S}_{j'})} + \frac{\mathrm{ASSOC}(\mathcal{S}_{j'},\mathcal{S}_{j'})}{\mathrm{ASSOC}(\mathcal{S}_{j'},\mathcal{S}_j\cup\mathcal{S}_{j'})}\right) = \mathrm{NASSOC}(\mathcal{S}_j,\mathcal{S}_{j'}) ,\tag{10}$$

$$v_{jj'}^{G,s} \doteq \exp(-\|\boldsymbol{b}_j-\boldsymbol{b}_{j'}\|_2/s) , s > 0 .\tag{11}$$

Here, $\mathrm{ASSOC}(\mathcal{S}_j,\mathcal{S}_{j'}) \doteq \sum_{\boldsymbol{x}\in\mathcal{S}_j,\boldsymbol{x}'\in\mathcal{S}_{j'}}\|\boldsymbol{x}-\boldsymbol{x}'\|_2^2$ [22]. To put these two similarity measures in the context of Theorem 3, consider the setting where we can make assumption (**D1**) that there exists a small constant $\kappa > 0$ such that $\|\boldsymbol{b}_j - \boldsymbol{b}_{j'}\|_2^2 \geq \kappa\max_{\sigma,j}\|\boldsymbol{b}_j^\sigma\|_2^2, \forall j, j' \in [n]$. This is a weak distinguishability property as if no such $\kappa$ exists, then the centers of distinct bags may just be confounded. Consider also the additional assumption, (**D2**), that there exists $\kappa' > 0$ such that $\max_j d_j^2 \leq \kappa', \forall j \in [n]$, where $d_j \doteq \max_{\boldsymbol{x}_i,\boldsymbol{x}'_i\in\mathcal{S}_j}\|\boldsymbol{x}_i - \boldsymbol{x}_{i'}\|_2$ is a bag's diameter. In the following Lemma, the little-oh notation is with respect to the "largest" unknown in eq. (4), *i.e.* $\max_{\sigma,j}\|\boldsymbol{b}_j^\sigma\|_2$.

---

**Algorithm 2** Alternating Mean Map (AMM$^{\text{OPT}}$)

---

**Input** LMM parameters + optimization strategy OPT $\in \{\min, \max\}$ + convergence predicate PR

Step 1 : let $\tilde{\boldsymbol{\theta}}_0 \leftarrow$ LMM(LMM parameters) and $t \leftarrow 0$

Step 2 : **repeat**

      Step 2.1 : let $\boldsymbol{\sigma}_t \leftarrow \arg \text{OPT}_{\boldsymbol{\sigma} \in \Sigma_{\tilde{\pi}}} F_\phi(\mathcal{S}_{|y}, \boldsymbol{\theta}_t, \boldsymbol{\mu}_\mathcal{S}(\boldsymbol{\sigma}))$

      Step 2.2 : let $\tilde{\boldsymbol{\theta}}_{t+1} \leftarrow \arg \min_{\boldsymbol{\theta}} F_\phi(\mathcal{S}_{|y}, \boldsymbol{\theta}, \boldsymbol{\mu}_\mathcal{S}(\boldsymbol{\sigma}_t)) + \lambda \|\boldsymbol{\theta}\|_2^2$

      Step 2.3 : let $t \leftarrow t + 1$

      **until** predicate PR is true

**Return** $\tilde{\boldsymbol{\theta}}_* \doteq \arg \min_t F_\phi(\mathcal{S}_{|y}, \tilde{\boldsymbol{\theta}}_{t+1}, \boldsymbol{\mu}_\mathcal{S}(\boldsymbol{\sigma}_t))$

---

**Lemma 4** *There exists $\varepsilon_* > 0$ such that $\forall \varepsilon \leq \varepsilon_*$, the following holds: (i) $\varsigma(\mathrm{V}^{nc}, \mathrm{B}^{\pm}) = o(1)$ under assumptions (**D1** + **D2**); (ii) $\varsigma(\mathrm{V}^{G,s}, \mathrm{B}^{\pm}) = o(1)$ under assumption (**D1**), $\forall s > 0$.*

([19], Subsection 2.4) Hence, provided a weak (**D1**) or stronger (**D1+D2**) distinguishability assumption holds, the divergence between M and M̃ gets smaller with the increase of the norm of the unknowns $\boldsymbol{b}_j^\sigma$. The proof of the Lemma suggests that the convergence may be faster for $\mathrm{V}^{G,s}$. The following Lemma shows that both similarities also partially encode the hardness of solving the classification problem with linear separators, so that the manifold regularizer "limits" the distortion of the $\tilde{\boldsymbol{b}}_{\cdot}^{\pm}$s between two bags that tend not to be linearly separable.

**Lemma 5** *Take $v_{jj'} \in \{v_{jj'}^{G,\cdot}, v_{jj'}^{nc}\}$. There exists $0 < \kappa_l < \kappa_n < 1$ such that (i) if $v_{jj'} > \kappa_n$ then $\mathcal{S}_j, \mathcal{S}_{j'}$ are not linearly separable, and if $v_{jj'} < \kappa_l$ then $\mathcal{S}_j, \mathcal{S}_{j'}$ are linearly separable.*

([19], Subsection 2.5) This Lemma is an advocacy to fit $s$ in a data-dependent way in $v_{jj'}^{G,s}$. The question may be raised as to whether finite samples approximation results like Theorem 3 can be proven for the Mean Map estimator [17]. [19], Subsection 2.6 answers by the negative.

In the Laplacian Mean Map algorithm (LMM, Algorithm 1), Steps 1 and 2 have now been described. Step 3 is a differentiable convex minimization problem for $\boldsymbol{\theta}$ that does not use the labels, so it does not present any technical difficulty. An interesting question is how much our classifier $\tilde{\boldsymbol{\theta}}_*$ in Step 3 diverges from the one that would be computed with the true expression for $\boldsymbol{\mu}_\mathcal{S}$, $\boldsymbol{\theta}_*$. It is not hard to show that Lemma 17 in Altun and Smola [23], and Corollary 9 in Quadrianto *et al.* [17] hold for LMM so that $\|\tilde{\boldsymbol{\theta}}_* - \boldsymbol{\theta}_*\|_2^2 \leq (2\lambda)^{-1}\|\tilde{\boldsymbol{\mu}}_\mathcal{S} - \boldsymbol{\mu}_\mathcal{S}\|_2^2$. The following Theorem shows a data-dependent approximation bound that can be significantly better, when it holds that $\boldsymbol{\theta}_*^\top \boldsymbol{x}_i, \tilde{\boldsymbol{\theta}}_*^\top \boldsymbol{x}_i \in \phi'([0,1]), \forall i$ ($\phi'$ is the first derivative). We call this setting *proper scoring compliance* (PSC) [18]. PSC always holds for the logistic and Matsushita losses for which $\phi'([0,1]) = \mathbb{R}$. For other losses like the square loss for which $\phi'([0,1]) = [-1,1]$, shrinking the observations in a ball of sufficiently small radius is sufficient to ensure this.

**Theorem 6** *Let $\boldsymbol{f}_k \in \mathbb{R}^m$ denote the vector encoding the $k^{th}$ feature variable in $\mathcal{S}$ : $f_{ki} = x_{ik}$ ($k \in [d]$). Let $\tilde{\mathrm{F}}$ denote the feature matrix with column-wise normalized feature vectors: $\tilde{\boldsymbol{f}}_k \doteq (d/\sum_{k'} \|\boldsymbol{f}_{k'}\|_2^2)^{(d-1)/(2d)} \boldsymbol{f}_k$. Under PSC, we have $\|\tilde{\boldsymbol{\theta}}_* - \boldsymbol{\theta}_*\|_2^2 \leq (2\lambda + q)^{-1}\|\tilde{\boldsymbol{\mu}}_\mathcal{S} - \boldsymbol{\mu}_\mathcal{S}\|_2^2$, with:*

$$q \doteq \frac{\det \tilde{\mathrm{F}}^\top \tilde{\mathrm{F}}}{m} \times \frac{2e^{-1}}{b_\phi \phi'' (\phi'^{-1}(q'/\lambda))} \quad (> 0) , \tag{12}$$

*for some $q' \in \mathbb{I} \doteq [\pm(x_* + \max\{\|\boldsymbol{\mu}_\mathcal{S}\|_2, \|\tilde{\boldsymbol{\mu}}_\mathcal{S}\|_2\})]$. Here, $x_* \doteq \max_i \|\boldsymbol{x}_i\|_2$ and $\phi'' \doteq (\phi')'$.*

([19], Subsection 2.7) To see how large $q$ can be, consider the simple case where all eigenvalues of $\tilde{\mathrm{F}}^\top \tilde{\mathrm{F}}$, $\lambda_k(\tilde{\mathrm{F}}^\top \tilde{\mathrm{F}}) \in [\lambda_\circ \pm \delta]$ for small $\delta$. In this case, $q$ is proportional to the average feature "norm":

$$\frac{\det \tilde{\mathrm{F}}^\top \tilde{\mathrm{F}}}{m} = \frac{\text{tr} (\mathrm{F}^\top \mathrm{F})}{md} + o(\delta) = \frac{\sum_i \|\boldsymbol{x}_i\|_2^2}{md} + o(\delta) .$$

**The Alternating Mean Map (AMM) algorithm** Let us denote $\Sigma_{\hat{\boldsymbol{\pi}}} \doteq \{\boldsymbol{\sigma} \in \Sigma_m : \sum_{i:\boldsymbol{x}_i \in \mathcal{S}_j} \sigma_i = (2\hat{\pi}_j - 1)m_j, \forall j \in [n]\}$ the set of labelings that are *consistent* with the observed proportions $\hat{\boldsymbol{\pi}}$, and $\boldsymbol{\mu}_\mathcal{S}(\boldsymbol{\sigma}) \doteq (1/m) \sum_i \sigma_i \boldsymbol{x}_i$ the biased mean operator computed from some $\boldsymbol{\sigma} \in \Sigma_{\hat{\boldsymbol{\pi}}}$. Notice that the true mean operator $\boldsymbol{\mu}_\mathcal{S} = \boldsymbol{\mu}_\mathcal{S}(\boldsymbol{\sigma})$ for at least one $\boldsymbol{\sigma} \in \Sigma_{\hat{\boldsymbol{\pi}}}$. The Alternating Mean Map algorithm, (AMM, Algorithm 2), starts with the output of LMM and then optimizes it further over the set of consistent labelings. At each iteration, it first picks a consistent labeling in $\Sigma_{\hat{\boldsymbol{\pi}}}$ that is the best (OPT = min) or the worst (OPT = max) for the current classifier (Step 2.1) and then fits a classifier $\tilde{\boldsymbol{\theta}}$ on the given set of labels (Step 2.2). The algorithm then iterates until a convergence predicate is met, which tests whether the difference between two values for $F_\phi(.,.,.)$ is too small (AMM$^{\text{min}}$), or the number of iterations exceeds a user-specified limit (AMM$^{\text{max}}$). The classifier returned $\tilde{\boldsymbol{\theta}}_*$ is the best in the sequence. In the case of AMM$^{\text{min}}$, it is the last of the sequence as risk $F_\phi(\mathcal{S}_{|y}, ., .)$ cannot increase. Again, Step 2.2 is a convex minimization with no technical difficulty. Step 2.1 is combinatorial. It can be solved in time almost linear in $m$ [19] (Subsection 2.8).

**Lemma 7** *The running time of Step 2.1 in* AMM *is* $\tilde{O}(m)$, *where the tilde notation hides log-terms.*

**Bag-Rademacher generalization bounds for LLP** We relate the "min" and "max" strategies of AMM by uniform convergence bounds involving the *true* surrogate risk, *i.e.* integrating the unknown distribution $\mathcal{D}$ *and* the true labels (which we may never know). Previous uniform convergence bounds for LLP focus on coarser grained problems, like the estimation of label proportions [1]. We rely on a LLP generalization of Rademacher complexity [24, 25]. Let $F : \mathbb{R} \to \mathbb{R}^+$ be a loss function and $\mathcal{H}$ a set of classifiers. The bag empirical Rademacher complexity of sample $\mathcal{S}$, $R_m^b$, is defined as $R_m^b \doteq \mathbb{E}_{\boldsymbol{\sigma} \sim \Sigma_m} \sup_{h \in \mathcal{H}} \{\mathbb{E}_{\boldsymbol{\sigma}' \sim \Sigma_{\hat{\boldsymbol{\pi}}}} \mathbb{E}_\mathcal{S}[\sigma(\boldsymbol{x})F(\sigma'(\boldsymbol{x})h(\boldsymbol{x}))]$. The usual empirical Rademacher complexity equals $R_m^b$ for $\text{card}(\Sigma_{\hat{\boldsymbol{\pi}}}) = 1$. The Label Proportion Complexity of $\mathcal{H}$ is:

$$L_{2m} \quad \doteq \quad \mathbb{E}_{\mathcal{D}_{2m}} \mathbb{E}_{\mathcal{I}_1^{/2}, \mathcal{I}_2^{/2}} \sup_{h \in \mathcal{H}} \mathbb{E}_\mathcal{S}[\sigma_1(\boldsymbol{x})(\hat{\pi}_{|2}^s(\boldsymbol{x}) - \hat{\pi}_{|1}^\ell(\boldsymbol{x}))h(\boldsymbol{x})] \ . \tag{13}$$

Here, each of $\mathcal{I}_l^{/2}, l = 1, 2$ is a random (uniformly) subset of $[2m]$ of cardinal $m$. Let $\mathcal{S}(\mathcal{I}_l^{/2})$ be the size-$m$ subset of $\mathcal{S}$ that corresponds to the indexes. Take $l = 1, 2$ and any $\boldsymbol{x}_i \in \mathcal{S}$. If $i \notin \mathcal{I}_l^{/2}$ then $\hat{\pi}_{|l}^s(\boldsymbol{x}_i) = \hat{\pi}_{|l}^\ell(\boldsymbol{x}_i)$ is $\boldsymbol{x}_i$'s bag's label proportion measured on $\mathcal{S} \backslash \mathcal{S}(\mathcal{I}_l^{/2})$. Else, $\hat{\pi}_{|2}^s(\boldsymbol{x}_i)$ is its bag's label proportion measured on $\mathcal{S}(\mathcal{I}_2^{/2})$ and $\hat{\pi}_{|1}^\ell(\boldsymbol{x}_i)$ is its label (*i.e.* a bag's label proportion that would contain only $\boldsymbol{x}_i$). Finally, $\sigma_1(\boldsymbol{x}) \doteq 2 \times \mathbb{1}_{\boldsymbol{x} \in \mathcal{S}(\mathcal{I}_1^{/2})} - 1 \in \Sigma_1$. $L_{2m}$ tends to be all the smaller as classifiers in $\mathcal{H}$ have small magnitude on bags whose label proportion is close to $1/2$.

**Theorem 8** *Suppose* $\exists h_* \geq 0$ *s.t.* $|h(\boldsymbol{x})| \leq h_*, \forall \boldsymbol{x}, \forall h$. *Then, for any loss* $F_\phi$, *any training sample of size* $m$ *and any* $0 < \delta \leq 1$, *with probability* $> 1 - \delta$, *the following bound holds over all* $h \in \mathcal{H}$:

$$\mathbb{E}_\mathcal{D}[F_\phi(yh(\boldsymbol{x}))] \quad \leq \quad \mathbb{E}_{\Sigma_{\hat{\boldsymbol{\pi}}}} \mathbb{E}_\mathcal{S}[F_\phi(\sigma(\boldsymbol{x})h(\boldsymbol{x}))] + 2R_m^b + L_{2m} + 4\left(\frac{2h_*}{b_\phi} + 1\right)\sqrt{\frac{1}{2m}\log\frac{2}{\delta}} \tag{14}$$

*Furthermore, under* PSC *(Theorem 6), we have for any* $F_\phi$:

$$R_m^b \quad \leq \quad 2b_\phi \mathbb{E}_{\Sigma_m} \sup_{h \in \mathcal{H}} \{\mathbb{E}_\mathcal{S}[\sigma(\boldsymbol{x})(\hat{\pi}(\boldsymbol{x}) - (1/2))h(\boldsymbol{x})]\} \ . \tag{15}$$

([19], Subsection 2.9) Despite similar shapes (13) (15), $R_m^b$ and $L_{2m}$ behave differently: when bags are pure ($\hat{\pi}_j \in \{0, 1\}, \forall j$), $L_{2m} = 0$. When bags are impure ($\hat{\pi}_j = 1/2, \forall j$), $R_m^b = 0$. As bags get impure, the bag-empirical surrogate risk, $\mathbb{E}_{\Sigma_{\hat{\boldsymbol{\pi}}}} \mathbb{E}_\mathcal{S}[F_\phi(\sigma(\boldsymbol{x})h(\boldsymbol{x}))]$, also tends to increase. AMM$^{\text{min}}$ and AMM$^{\text{max}}$ respectively minimize a lowerbound and an upperbound of this risk.

## 3 Experiments

**Algorithms** We compare LMM, AMM ($F_\phi$ = logistic loss) to the original MM [17], InvCal [11], conv-$\propto$SVM and alter-$\propto$SVM [16] (linear kernels). To make experiments extensive, we test several initializations for AMM that are not displayed in Algorithm 2 (Step 1): (i) the edge mean map estimator, $\tilde{\mu}_\mathcal{S}^{\text{EMM}} \doteq 1/m^2(\sum_i y_i)(\sum_i \boldsymbol{x}_i)$ (AMM$_{\text{EMM}}$), (ii) the constant estimator $\tilde{\mu}_\mathcal{S}^1 \doteq \mathbf{1}$ (AMM$_1$), and finally AMM$_{10\text{ran}}$ which runs 10 random initial models ($\|\boldsymbol{\theta}_0\|_2 \leq 1$), and selects the one with smallest risk;

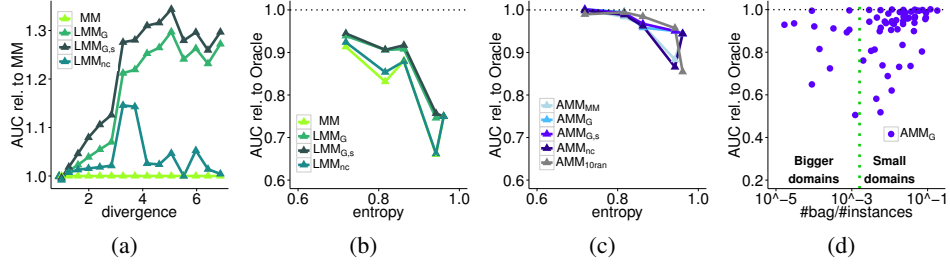

Figure 1: Relative AUC (wrt MM) as homogeneity assumption is violated (a). Relative AUC (wrt Oracle) vs entropy on *heart* for LMM(b), AMM$^{\min}$(c). Relative AUC vs $n/m$ for AMM$_{G,s}^{\min}$ (d).

Table 2: Small domains results. #win/#lose for row vs column. Bold faces means $p$-val $< .001$ for Wilcoxon signed-rank tests. Top-left subtable is for one-shot methods, bottom-right iterative ones, bottom-left compare the two. Italic is state-of-the-art. Grey cells highlight the best of all (AMM$_G^{\min}$).

| algorithm | | MM | G | LMM G,s | nc | InvCal | AMM$^{\min}$ MM | G | G,s | 10ran | AMM$^{\max}$ MM | G | G,s | 10ran | conv-$\propto$SVM |
|---|---|---|---|---|---|---|---|---|---|---|---|---|---|---|---|
| LMM | G | **36/4** | | | | | | | | | | | | | |
| | G,s | **38/3** | **30/6** | | | | | | | | | | | | |
| | nc | **28/12** | 3/37 | 2/37 | | | | | | | | | | | |
| | *InvCal* | 4/46 | 3/47 | 4/46 | 4/46 | | | | | | | | | | |
| AMM$^{\min}$ | MM | **33/16** | 26/24 | 25/25 | 32/18 | **46/4** | | | | | | | | | |
| | G | **38/11** | **35/14** | **30/20** | **37/13** | **47/3** | **31/7** | | | | | | | | |
| | G,s | **35/14** | 33/17 | **30/20** | **35/15** | **47/3** | 24/11 | 7/15 | | | | | | | |
| | 10ran | 27/22 | 24/26 | 22/28 | 26/24 | **44/6** | 20/30 | 16/34 | 19/31 | | | | | | |
| AMM$^{\max}$ | MM | 25/25 | 23/27 | 22/28 | 25/25 | **45/5** | 15/35 | 13/37 | 13/37 | 8/42 | | | | | |
| | G | 27/23 | 22/28 | 21/28 | **26/24** | **45/5** | 17/33 | 14/36 | 14/36 | 10/40 | 13/14 | | | | |
| | G,s | 25/25 | 21/29 | 22/28 | 24/26 | **45/5** | 15/35 | 13/37 | 13/37 | 12/38 | 15/22 | 16/22 | | | |
| | 10ran | 23/27 | 21/29 | 19/31 | 24/26 | **50/0** | 19/31 | 15/35 | 17/33 | 7/43 | 19/30 | 20/29 | 17/32 | | |
| SVM | *conv-$\propto$* | 21/29 | 2/48 | 2/48 | 2/48 | 2/48 | 4/46 | 3/47 | 3/47 | 4/46 | 3/47 | 3/47 | 4/46 | 0/50 | |
| | *alter-$\propto$* | 0/50 | 0/50 | 0/50 | 0/50 | 20/30 | 0/50 | 0/50 | 0/50 | 3/47 | 3/47 | 2/48 | 1/49 | 0/50 | 27/23 |

*e.g.* AMM$_{G,s}^{\min}$ wins on AMM$_G^{\min}$ 7 times, loses **15**, with 28 ties

this is the same procedure of alter-$\propto$SVM. Matrix V (eqs. (10), (11)) used is indicated in subscript: LMM/AMM$_G$, LMM/AMM$_{G,s}$, LMM/AMM$_{nc}$ respectively denote $v^{G,s}$ with $s = 1$, $v^{G,s}$ with $s$ learned on cross validation (CV; validation ranges indicated in [19]) and $v^{nc}$. For space reasons, results not displayed in the paper can be found in [19], Section 3 (including runtime comparisons, and detailed results by domain). We split the algorithms in two groups, *one-shot* and *iterative*. The latter, including AMM, (conv/alter)-$\propto$SVM, iteratively optimize a cost over labelings (always consistent with label proportions for AMM, not always for (conv/alter)-$\propto$SVM). The former (LMM, InvCal) do not and are thus much faster. Tests are done on a 4-core 3.2GHz CPUs Mac with 32GB of RAM. AMM/LMM/MM are implemented in R. Code for InvCal and $\propto$SVM is [16].

**Simulated domains, MM and the homogeneity assumption** The testing metric is the AUC. Prior to testing on our domains, we generate 16 domains that gradually move away the $b_j^\sigma$ away from each other (wrt $j$), thus violating increasingly the homogeneity assumption [17]. The degree of violation is measured as $\|B^\pm - \overline{B^\pm}\|_F$, where $\overline{B^\pm}$ is the homogeneity assumption matrix, that replaces all $b_j^\sigma$ by $b^\sigma$ for $\sigma \in \{-1, 1\}$, see eq. (5). Figure 1 (a) displays the ratios of the AUC of LMM to the AUC of MM. It shows that LMM is all the better with respect to MM as the homogeneity assumption is violated. Furthermore, learning $s$ in LMM improves the results. Experiments on the simulated domain of [16] on which MM obtains zero accuracy also display that our algorithms perform better (1 iteration only of AMM$^{\max}$ brings 100% AUC).

**Small and large domains experiments** We convert 10 small domains [19] ($m \leq 1000$) and 4 bigger ones ($m > 8000$) from UCI[26] into the LLP framework. We cast to one-against-all classification when the problem is multiclass. On large domains, the bag assignment function is inspired by [1]: we craft bags according to a selected feature value, and then we remove that feature from the data. This conforms to the idea that bag assignment is structured and non random in real-world problems. Most of our small domains, however, do not have a lot of features, so instead of clustering on one feature and then discard it, we run K-MEANS on the whole data to make the bags, for K $= n \in 2^{[5]}$.

**Small domains results** We performe 5-folds nested CV comparisons on the 10 domains = 50 AUC values for each algorithm. Table 2 synthesises the results [19], splitting one-shot and iterative algo-

Table 3: AUCs on big domains (*name*: #instances×#features). I=*cap-shape*, II=*habitat*, III=*cap-colour*, IV=*race*, V=*education*, VI=*country*, VII=*poutcome*, VIII=*job* (number of bags); for each feature, the best result over one-shot, and over iterative algorithms is bold faced.

| algorithm | | *mushroom*: 8124 × 108 | | | *adult*: 48842 × 89 | | | *marketing*: 45211 × 41 | | | *census*: 299285 × 381 | | |
|---|---|---|---|---|---|---|---|---|---|---|---|---|---|
| | | I(6) | II(7) | III(10) | IV(5) | V(16) | VI(42) | V(4) | VII(4) | VIII(12) | IV(5) | VIII(9) | VI(42) |
| | EMM | 55.61 | 59.80 | 76.68 | 43.91 | 47.50 | 66.61 | **63.49** | **54.50** | 44.31 | 56.05 | 56.25 | 57.87 |
| | MM | 51.99 | **98.79** | 5.02 | 80.93 | 76.65 | 74.01 | 54.64 | 50.71 | 49.70 | 75.21 | **90.37** | 75.52 |
| | LMM$_G$ | 73.92 | 98.57 | 14.70 | 81.79 | 78.40 | 78.78 | 54.66 | 51.00 | 51.93 | 75.80 | 71.75 | **76.31** |
| | LMM$_{G,s}$ | **94.91** | 98.24 | **89.43** | **84.89** | **78.94** | **80.12** | 49.27 | 51.00 | **65.81** | **84.88** | 60.71 | 69.74 |
| AMM$^{min}$ | AMM$_{EMM}$ | 85.12 | 99.45 | 69.43 | 49.97 | 56.98 | 70.19 | 61.39 | 55.73 | 43.10 | 87.86 | 87.71 | 40.80 |
| | AMM$_{MM}$ | 89.81 | 99.01 | 15.74 | **83.73** | 77.39 | 80.67 | 52.85 | **75.27** | 58.19 | 89.68 | 84.91 | 68.36 |
| | AMM$_G$ | 89.18 | 99.45 | 50.44 | 83.41 | **82.55** | **81.96** | 51.61 | 75.16 | 57.52 | 87.61 | 88.28 | 76.99 |
| | AMM$_{G,s}$ | 89.24 | **99.57** | 3.28 | 81.18 | 78.53 | **81.96** | 52.03 | 75.16 | 53.98 | **89.93** | 83.54 | 52.13 |
| | AMM$_1$ | **95.90** | 98.49 | 97.31 | 81.32 | 75.80 | 80.05 | 65.13 | 64.96 | 66.62 | 89.09 | **88.94** | 56.72 |
| AMM$^{max}$ | AMM$_{EMM}$ | 93.04 | 3.32 | 26.67 | 54.46 | 69.63 | 56.62 | 51.48 | 55.63 | 57.48 | 71.20 | 77.14 | 66.71 |
| | AMM$_{MM}$ | 59.45 | 55.16 | **99.70** | 82.57 | 71.63 | 81.39 | 48.46 | 51.34 | 56.90 | 50.75 | 66.76 | 58.67 |
| | AMM$_G$ | 95.50 | 65.32 | 99.30 | 82.75 | 72.16 | 81.39 | 50.58 | 47.27 | 34.29 | 48.32 | 67.54 | **77.46** |
| | AMM$_{G,s}$ | 95.84 | 65.32 | 84.26 | 82.69 | 70.95 | 81.39 | **66.88** | 47.27 | 34.29 | 80.33 | 74.45 | 52.70 |
| | AMM$_1$ | 95.01 | 73.48 | 1.29 | 75.22 | 67.52 | 77.67 | 66.70 | 61.16 | **71.94** | 57.97 | 81.07 | 53.42 |
| | Oracle | 99.82 | 99.81 | 99.8 | 90.55 | 90.55 | 90.50 | 79.52 | 75.55 | 79.43 | 94.31 | 94.37 | 94.45 |

rithms. LMM$_{G,s}$ outperforms all one-shot algorithms. LMM$_G$ and LMM$_{G,s}$ are competitive with many iterative algorithms, but lose against their AMM counterpart, which proves that additional optimization over labels is beneficial. AMM$_G$ and AMM$_{G,s}$ are confirmed as the best variant of AMM, the first being the best in this case. Surprisingly, all mean map algorithms, even one-shots, are clearly superior to ∝SVMs. Further results [19] reveal that ∝SVM performances are dampened by learning classifiers with the "inverted polarity" — *i.e.* flipping the sign of the classifier improves its performances. Figure 1 (b, c) presents the AUC relative to the Oracle (which learns the classifier knowing all labels and minimizing the logistic loss), as a function of the Gini entropy of bag assignment, $gini(\mathcal{S}) \doteq 4\mathbb{E}_j[\hat{\pi}_j(1 - \hat{\pi}_j)]$. For an entropy close to 1, we were expecting a drop in performances. The unexpected [19] is that on some domains, large entropies ($\geq .8$) do not prevent AMM$^{min}$ to compete with the Oracle. No such pattern clearly emerges for ∝SVM and AMM$^{max}$ [19].

**Big domains results** We adopt a 1/5 hold-out method. Scalability results [19] display that every method using $v^{nc}$ and ∝SVM are not scalable to big domains; in particular, the estimated time for a single run of alter-∝SVM is >100 hours on the adult domain. Table 3 presents the results on the big domains, distinguishing the feature used for bag assignment. Big domains confirm the efficiency of LMM+AMM. No approach clearly outperforms the rest, although LMM$_{G,s}$ is often the best one-shot.

**Synthesis** Figure 1 (d) gives the AUCs of AMM$_G^{min}$ over the Oracle for *all* domains [19], as a function of the "degree of supervision", $n/m$ (=1 if the problem is fully supervised). Noticeably, on 90% of the runs, AMM$_G^{min}$ gets an AUC representing at least 70% of the Oracle's. Results on big domains can be remarkable: on the *census* domain with bag assignment on *race*, 5 proportions are sufficient for an AUC 5 points below the Oracle's — which learns with 200K labels.

## 4 Conclusion

In this paper, we have shown that efficient learning in the LLP setting is possible, for general loss functions, via the mean operator and without resorting to the homogeneity assumption. Through its estimation, the sufficiency allows one to resort to standard learning procedures for binary classification, practically implementing a *reduction* between machine learning problems [27]; hence the mean operator estimation may be a viable shortcut to tackle other weakly supervised settings [2] [3] [4] [5]. Approximation results and generalization bounds are provided. Experiments display results that are superior to the state of the art, with algorithms that scale to big domains at affordable computational costs. Performances sometimes compete with the Oracle's — that learns knowing all labels —, even on big domains. Such experimental finding poses severe implications on the reliability of privacy-preserving aggregation techniques with simple group statistics like proportions.

### Acknowledgments

NICTA is funded by the Australian Government through the Department of Communications and the Australian Research Council through the ICT Centre of Excellence Program. The first author would like to acknowledge that part of this research was conducted during his internship at the Commonwealth Bank of Australia. We thank A. Menon and D. García-García for useful discussions.

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
