[Supplementary Material]

# (Almost) No Label No Cry - Supplementary Material

**Giorgio Patrini**[1,2], **Richard Nock**[1,2], **Paul Rivera**[1,2], **Tiberio Caetano**[1,3,4]
Australian National University[1], NICTA[2], University of New South Wales[3], Ambiata[4]
Sydney, NSW, Australia
{name.surname}@anu.edu.au

## 1   Table of contents

## 2 Supplementary Material on Proofs

### 2.1 Proof of Lemma 1

For any SPSL $F(\mathcal{S}, h)$, we can write it as ([1], Lemma 1, [2]):

$$
\begin{aligned}
F(\mathcal{S}, h) &= F_\phi(\mathcal{S}, h) \\
&\doteq \frac{1}{m} \sum_i D_\phi(y_i' \| \phi'^{-1}(h(\boldsymbol{x}_i))) \ ,
\end{aligned}
\tag{1}
$$

where $y_i' = 1$ iff $y_i = 1$ and 0 otherwise, $\phi$ is permissible and $D_\phi$ is the Bregman divergence with generator $\phi$ [2]. It also holds that: $D_\phi(y_i' \| \phi'^{-1}(h(\boldsymbol{x}_i))) = b_\phi F_\phi(yh(\boldsymbol{x}))$ with:

$$
F_\phi(x) \doteq \frac{\phi^\star(-x) + \phi(0)}{\phi(0) - \phi(1/2)} = a_\phi + \frac{\phi^\star(-x)}{b_\phi} \ ,
\tag{2}
$$

and $\phi^\star$ is the convex conjugate of $\phi$, *i.e.* $\phi^\star(x) \doteq x\phi'^{-1}(x) - \phi(\phi'^{-1}(x))$. Furthermore, for any permissible $\phi$, the conjex conjugate $\phi^\star(x)$ verifies the property

$$
\phi^\star(-x) = \phi^\star(x) - x \ ,
\tag{3}
$$

and so we get that:

$$
\begin{aligned}
F(\mathcal{S}, h) &= \frac{1}{m} \sum_i D_\phi(y_i' \| \phi'^{-1}(h(\boldsymbol{x}_i))) \\
&= \frac{b_\phi}{m} \sum_i F_\phi(y_i h(\boldsymbol{x}_i)) \\
&= \frac{b_\phi}{2m} \left( \sum_i F_\phi(y_i h(\boldsymbol{x}_i)) + \sum_i F_\phi(y_i h(\boldsymbol{x}_i)) \right) \\
&= \frac{b_\phi}{2m} \left( \sum_i F_\phi(y_i h(\boldsymbol{x}_i)) + \sum_i F_\phi(-y_i h(\boldsymbol{x}_i)) - \frac{1}{b_\phi} \sum_i y_i h(\boldsymbol{x}_i) \right) \\
&= \frac{b_\phi}{2m} \sum_{y \in \{-1,+1\}} \sum_i F_\phi(yh(\boldsymbol{x}_i)) - \frac{1}{2m} \sum_i y_i h(\boldsymbol{x}_i) \\
&= \frac{b_\phi}{2m} \sum_{\sigma \in \{-1,+1\}} \sum_i F_\phi(\sigma h(\boldsymbol{x}_i)) - \frac{1}{2} h\left( \frac{1}{m} \sum_i y_i \boldsymbol{x}_i \right) \\
&= \frac{b_\phi}{2m} \sum_{\sigma \in \{-1,+1\}} \sum_i F_\phi(\sigma h(\boldsymbol{x}_i)) - \frac{1}{2} h(\boldsymbol{\mu}_\mathcal{S}) \ .
\end{aligned}
$$

$$\tag{4}$$
$$\tag{5}$$
$$\tag{6}$$

(4) holds because of (3), (5) holds because $h$ is linear. So for any samples $\mathcal{S}$ and $\mathcal{S}$ with respective size $m$ and $m'$, we have (again using the property that $h$ is linear):

$$
\begin{aligned}
F(\mathcal{S}, h) - F(\mathcal{S}', h) &= \frac{b_\phi}{2} \sum_{\sigma \in \{-1,+1\}} \left( \frac{1}{m} \sum_{\boldsymbol{x} \in \mathcal{S}_1} F_\phi(\sigma h(\boldsymbol{x}_i)) - \frac{1}{m'} \sum_{\boldsymbol{x} \in \mathcal{S}_2} F_\phi(\sigma h(\boldsymbol{x}_i)) \right) \\
&\quad + \frac{1}{2} h(\boldsymbol{\mu}_{\mathcal{S}_2} - \boldsymbol{\mu}_{\mathcal{S}_1}) \ ,
\end{aligned}
\tag{7}
$$

which yields the statement of the Lemma.

### 2.2 Proof of Lemma 2

Using the fact that $\mathrm{D}_{\boldsymbol{w}}$ and $\mathrm{L}$ are symmetric, we have:

$$
\begin{aligned}
\frac{\partial \ell(\mathrm{L}, \mathrm{X})}{\partial \mathrm{X}} &\\
&= -2 \frac{\partial}{\partial \mathrm{X}} \mathrm{tr}\left( \mathrm{B}^\top \mathrm{D}_{\boldsymbol{w}} \Pi^\top \mathrm{X} \right) + \frac{\partial}{\partial \mathrm{X}} \mathrm{tr}\left( \mathrm{X}^\top \Pi \mathrm{D}_{\boldsymbol{w}} \Pi^\top \mathrm{X} \right) + \gamma \frac{\partial}{\partial \mathrm{X}} \mathrm{tr}\left( \mathrm{X}^\top \mathrm{L} \mathrm{X} \right) \\
&= -2 \Pi \mathrm{D}_{\boldsymbol{w}} \mathrm{B} + 2 \Pi \mathrm{D}_{\boldsymbol{w}} \Pi^\top \mathrm{X} + 2\gamma \mathrm{L} \mathrm{X} = 0 \ ,
\end{aligned}
$$

out of which $\tilde{\mathrm{B}}^\pm$ follows in Lemma 2.

## 2.3 Proof of Theorem 3

We let $\Pi_o \doteq [\text{DIAG}(\hat{\boldsymbol{\pi}})|\text{DIAG}(\hat{\boldsymbol{\pi}} - \mathbf{1})]^\top N$ an orthonormal system ($n_{jj} = (\hat{\pi}_j^2 + (1 - \hat{\pi}_j)^2)^{-1/2}, \forall j \in [n]$ and 0 otherwise). Let $\mathbb{K}_{\Pi_o}$ be the $n$-dim subspace of $\mathbb{R}^d$ generated by $\Pi_o$. The proof of Theorem (3) exploits the following Lemma, which assumes that $\varepsilon$ is any $> 0$ real for L in (8) (main file) to be $\succ 0$. When $\varepsilon = 0$, the result of Theorem (3) still holds but follows a different proof.

**Lemma 1** *Let* $A \doteq \Pi D_w \Pi^\top$ *and* L *defined as in (8) (main paper). Denote for short*

$$U \doteq \left(L^{-1}A + \gamma^{-1}I\right)^{-1} . \tag{8}$$

*Suppose there exists* $\xi > 0$ *such that for any* $\boldsymbol{x} \in \mathbb{R}^{2n}$, *the projection of* $U\boldsymbol{x}$ *in* $\mathbb{K}_{\Pi_o}$, $\boldsymbol{x}_{U,o}$, *satisfies*

$$\|\boldsymbol{x}_{U,o}\|_2 \leq \xi\|\boldsymbol{x}\|_2 . \tag{9}$$

*Then:*

$$\|M - \tilde{M}\|_F \leq \gamma\xi\|B^\pm\|_F . \tag{10}$$

**Proof** Combining Lemma 2 and (5), we get

$$
\begin{aligned}
B^\pm - \tilde{B}^\pm &= -\left((A + \gamma L)^{-1} A - I\right) B^\pm \\
&= \left((\gamma L)^{-1}A + I\right)^{-1} B^\pm .
\end{aligned} \tag{11}
$$

Define the following permutation matrix:

$$C \doteq \begin{bmatrix} 0 & | & I \\ I & | & 0 \end{bmatrix} \in \mathbb{R}^{2n \times 2n} . \tag{12}$$

$A \doteq \Pi D_w \Pi^\top$ is not invertible but diagonalisable. Its (orthonormal) eigenvectors can be partitioned in two matrices $P_o$ and P such that:

$$
\begin{aligned}
P_o &\doteq [\text{DIAG}(\hat{\boldsymbol{\pi}} - \mathbf{1})|\text{DIAG}(\hat{\boldsymbol{\pi}})]^\top N = C\Pi_o \in \mathbb{R}^{2n \times n} \text{ (eigenvalues 0) }, \tag{13} \\
P &\doteq \Pi N \in \mathbb{R}^{2n \times n} \text{ (eigenvalues } w_j(\hat{\pi}_j^2 + (1 - \hat{\pi}_j)^2), \forall j) . \tag{14}
\end{aligned}
$$

We have:

$$
\begin{aligned}
M - \tilde{M} &= P_o^\top CB^\pm - P_o^\top C\tilde{B}^\pm \\
&= P_o^\top C \left((\gamma L)^{-1}A + I\right)^{-1} B^\pm \\
&= \Pi_o^\top \left((\gamma L)^{-1}A + I\right)^{-1} B^\pm \tag{15} \\
&= \gamma \Pi_o^\top \left(L^{-1}A + \gamma^{-1}I\right)^{-1} B^\pm . \tag{16}
\end{aligned}
$$

Eq. (15) follows from the fact that C is idempotent. Plugging Frobenius norm in (16), we obtain

$$
\begin{aligned}
\|M - \tilde{M}\|_F^2 &= \gamma^2 \|\Pi_o^\top \left(L^{-1}A + \gamma^{-1}I\right)^{-1} B^\pm\|_F^2 \\
&= \gamma^2 \sum_{k=1}^d \|\Pi_o^\top \left(L^{-1}A + \gamma^{-1}I\right)^{-1} \boldsymbol{b}_k^\pm\|_2^2 \\
&\leq \gamma^2\xi^2 \sum_{k=1}^d \|\boldsymbol{b}_k^\pm\|_2^2 \tag{17} \\
&= \gamma^2\xi^2 \|B^\pm\|_F^2 ,
\end{aligned}
$$

which yields (10). In (17), $\boldsymbol{b}_k^\pm$ denotes *column* $k$ in $B^\pm$. Ineq. (17) makes use of assumption (9). ∎

To ensure $\|\boldsymbol{x}_{U,o}\|_2 \leq \xi\|\boldsymbol{x}\|_2$, it is sufficient that $\|U\boldsymbol{x}\|_2 \leq \xi\|\boldsymbol{x}\|_2$, and since $\|U\boldsymbol{x}\|_2 \leq \|U\|_F\|\boldsymbol{x}\|_2$, it is sufficient to show that

$$\left\|U_\xi^{-1}\right\|_F^2 \leq 1 , \tag{18}$$

with $U_\xi \doteq L_\xi^{-1} A + \xi \gamma^{-1} I$, for relevant choices of $\xi$. We have let $L_\xi \doteq (1/\xi)L$. Let $0 \leq \lambda_1(.) \leq ... \leq \lambda_{2n}(.)$ denote the ordered eigenvalues of a positive-semidefinite matrix in $\mathbb{R}^{2n \times 2n}$. It follows that, since $L$ is symmetric positive definite, we have

$$\lambda_j(L_\xi^{-1} A) \geq \frac{\lambda_j(A)}{\lambda_{2n}(L_\xi)} \ (\geq 0) \ , \forall j \in [2n] \ .$$

We have used eq. (13). Weyl's Theorem then brings:

$$\lambda_j(U_\xi^{-1}) \leq \frac{\lambda_{2n}(L_\xi)}{\lambda_j(A) + \xi\gamma^{-1}\lambda_{2n}(L_\xi)} \leq \begin{cases} \xi^{-1}\gamma & \text{if} \quad j \in [n] \\ \frac{\lambda_{2n}(L_\xi)}{\lambda_j(A)} & \text{otherwise} \end{cases} \ . \tag{19}$$

Gershgorin's Theorem brings $\lambda_{2n} \leq (1/\xi)(\varepsilon + \max_j \sum_{j'} |l_{jj'}|)$, and furthermore the eigenvalues of $A$ satisfy $\lambda_j \geq w_j/2, \forall j \geq n+1$. We thus have:

$$\left\| U_\xi^{-1} \right\|_F^2 \leq \frac{n\gamma^2}{\xi^2} + \frac{4n \left( \varepsilon + \max_j \sum_{j'} |l_{jj'}| \right)^2}{\xi^2 \min_j w_j^2} \ . \tag{20}$$

In (19) and (20), we have used the eigenvalues of $A$ given in eqs (13) and (14). Assuming:

$$\gamma \leq \frac{\xi}{\sqrt{2n}} \ , \tag{21}$$

a sufficient condition for the right-hand side of (20) to be $\leq 1$ is that

$$\xi \geq \frac{\varepsilon + \max_j \sum_{j'} |l_{jj'}|}{2\sqrt{n} \min_j w_j} \ . \tag{22}$$

To finish up the proof, recall that $L = D - V$ with $d_{jj} \doteq \sum_{j,j'} v_{jj'}$ and the coordinates $v_{jj'} \geq 0$. Hence,

$$\begin{aligned} \sum_{j'} |l_{jj'}| &= 2 \sum_{j \neq j'} v_{jj'} \\ &\leq 2n \max_{j \neq j'} v_{jj'}, \forall j \in [n] \ . \end{aligned}$$

The proof is finished by plugging this upperbound in (22) to choose $\xi$, then taking the maximal value for $\gamma$ in (21) and finally solving the upperbound in (10). This ends the proof of Theorem 3.

## 2.4 Proof of Lemma 4

We first consider the normalized association criterion in (10):

$$\begin{aligned} v_{jj'}^N &\doteq \frac{1}{2} \left( \frac{\text{ASSOC}(\mathcal{S}_j, \mathcal{S}_j)}{\text{ASSOC}(\mathcal{S}_j, \mathcal{S}_j \cup \mathcal{S}_{j'})} + \frac{\text{ASSOC}(\mathcal{S}_{j'}, \mathcal{S}_{j'})}{\text{ASSOC}(\mathcal{S}_{j'}, \mathcal{S}_j \cup \mathcal{S}_{j'})} \right) \ , \\ \text{ASSOC}(\mathcal{S}_j, \mathcal{S}_{j'}) &\doteq \sum_{\boldsymbol{x} \in \mathcal{S}_j, \boldsymbol{x}' \in \mathcal{S}_{j'}} \| \boldsymbol{x} - \boldsymbol{x}' \|_2^2 \ . \end{aligned} \tag{23}$$

Remark that

$$
\begin{aligned}
\|\boldsymbol{b}_j - \boldsymbol{b}_{j'}\|_2^2 &= \left\| \frac{1}{m_j} \sum_{\boldsymbol{x}_i \in \mathcal{S}_j} \boldsymbol{x}_i - \frac{1}{m_{j'}} \sum_{\boldsymbol{x}_{i'} \in \mathcal{S}_{j'}} \boldsymbol{x}_{i'} \right\|_2^2 \\
&= \frac{1}{m_j^2} \left\| \sum_{\boldsymbol{x}_i \in \mathcal{S}_j} \boldsymbol{x}_i \right\|_2^2 + \frac{1}{m_{j'}^2} \left\| \sum_{\boldsymbol{x}_{i'} \in \mathcal{S}_{j'}} \boldsymbol{x}_{i'} \right\|_2^2 - \frac{2}{m_j m_{j'}} \left( \sum_{\boldsymbol{x}_i \in \mathcal{S}_j} \boldsymbol{x}_i \right)^{\top} \left( \sum_{\boldsymbol{x}_{i'} \in \mathcal{S}_{j'}} \boldsymbol{x}_{i'} \right) \\
&= \frac{1}{m_j^2} \left\| \sum_{\boldsymbol{x}_i \in \mathcal{S}_j} \boldsymbol{x}_i \right\|_2^2 + \frac{1}{m_{j'}^2} \left\| \sum_{\boldsymbol{x}_{i'} \in \mathcal{S}_{j'}} \boldsymbol{x}_{i'} \right\|_2^2 - \frac{2}{m_j m_{j'}} \sum_{\boldsymbol{x}_i \in \mathcal{S}_j, \boldsymbol{x}_{i'} \in \mathcal{S}_{j'}} \boldsymbol{x}_i^{\top} \boldsymbol{x}_{i'} \\
&\leq \frac{1}{m_j} \sum_{\boldsymbol{x}_i \in \mathcal{S}_j} \|\boldsymbol{x}_i\|_2^2 + \frac{1}{m_{j'}} \sum_{\boldsymbol{x}_{i'} \in \mathcal{S}_{j'}} \|\boldsymbol{x}_{i'}\|_2^2 - \frac{2}{m_j m_{j'}} \sum_{\boldsymbol{x}_i \in \mathcal{S}_j, \boldsymbol{x}_{i'} \in \mathcal{S}_{j'}} \boldsymbol{x}_i^{\top} \boldsymbol{x}_{i'} \qquad (24) \\
&= \frac{1}{m_j m_{j'}} \sum_{\boldsymbol{x}_i \in \mathcal{S}_j, \boldsymbol{x}_{i'} \in \mathcal{S}_{j'}} \|\boldsymbol{x}_i - \boldsymbol{x}_{i'}\|_2^2 \\
&\quad + \underbrace{\frac{m_{j'} - 1}{m_j m_{j'}} \sum_{\boldsymbol{x}_i \in \mathcal{S}_j} \|\boldsymbol{x}_i\|_2^2 + \frac{m_j - 1}{m_j m_{j'}} \sum_{\boldsymbol{x}_{i'} \in \mathcal{S}_{j'}} \|\boldsymbol{x}_{i'}\|_2^2 - \frac{1}{m_j m_{j'}} \sum_{\boldsymbol{x}_i \in \mathcal{S}_j, \boldsymbol{x}_{i'} \in \mathcal{S}_{j'}} \boldsymbol{x}_i^{\top} \boldsymbol{x}_{i'}}_{\doteq a} \\
&\leq \frac{2}{m_j m_{j'}} \sum_{\boldsymbol{x}_i \in \mathcal{S}_j, \boldsymbol{x}_{i'} \in \mathcal{S}_{j'}} \|\boldsymbol{x}_i - \boldsymbol{x}_{i'}\|_2^2 \qquad (25) \\
&= \frac{2}{m_j m_{j'}} \operatorname{ASSOC}(\mathcal{S}_j, \mathcal{S}_{j'}) \ . \qquad (26)
\end{aligned}
$$

Eq. (24) exploits the fact that $\left( \sum_{j=1}^n a_j \right)^2 \leq n \left( \sum_{j=1}^n a_j^2 \right)$ and eq. (25) exploits the fact that $a \leq (m_j m_{j'})^{-1} \sum_{\boldsymbol{x}_i \in \mathcal{S}_j, \boldsymbol{x}_{i'} \in \mathcal{S}_{j'}} \|\boldsymbol{x}_i - \boldsymbol{x}_{i'}\|_2^2$. We thus have:

$$
\begin{aligned}
\frac{\operatorname{ASSOC}(\mathcal{S}_j, \mathcal{S}_j)}{\operatorname{ASSOC}(\mathcal{S}_j, \mathcal{S}_j \cup \mathcal{S}_{j'})} &= \frac{\operatorname{ASSOC}(\mathcal{S}_j, \mathcal{S}_j)}{\operatorname{ASSOC}(\mathcal{S}_j, \mathcal{S}_j) + \operatorname{ASSOC}(\mathcal{S}_j, \mathcal{S}_{j'})} \\
&\leq \frac{\operatorname{ASSOC}(\mathcal{S}_j, \mathcal{S}_j)}{\operatorname{ASSOC}(\mathcal{S}_j, \mathcal{S}_j) + \frac{m_j m_{j'}}{2} \|\boldsymbol{b}_j - \boldsymbol{b}_{j'}\|_2^2} \qquad (27) \\
&\leq \frac{\kappa' m_j}{\kappa' m_j + \frac{m_j m_{j'}}{2} \|\boldsymbol{b}_j - \boldsymbol{b}_{j'}\|_2^2} \qquad (28) \\
&= \frac{1}{1 + \frac{m_{j'}}{2\kappa'} \|\boldsymbol{b}_j - \boldsymbol{b}_{j'}\|_2^2} \ . \qquad (29)
\end{aligned}
$$

Eq. (27) uses (26) and eq. (28) uses assumption (**D2**). Eq. (28) also holds when permuting $j$ and $j'$, so we get:

$$
\begin{aligned}
\varsigma(\mathrm{V}^{NC}, \mathrm{B}^{\pm}) &\leq \max_{j \neq j'} \left( \frac{\varepsilon}{2n} + \frac{1}{1 + \frac{m_j}{2\kappa'} \|\boldsymbol{b}_j - \boldsymbol{b}_{j'}\|_2^2} + \frac{1}{1 + \frac{m_{j'}}{2\kappa'} \|\boldsymbol{b}_j - \boldsymbol{b}_{j'}\|_2^2} \right)^2 \|\mathrm{B}^{\pm}\|_F \\
&\leq \left( \frac{\varepsilon}{2n} + \frac{1}{1 + \frac{\min_j m_j}{2\kappa'} \min_{j,j'} \|\boldsymbol{b}_j - \boldsymbol{b}_{j'}\|_2^2} \right)^2 \|\mathrm{B}^{\pm}\|_F \\
&\leq \left( \frac{\varepsilon^2}{2n^2} + 2 \left( \frac{1}{1 + \frac{\min_j m_j}{2\kappa'} \min_{j,j'} \|\boldsymbol{b}_j - \boldsymbol{b}_{j'}\|_2^2} \right)^2 \right) \|\mathrm{B}^{\pm}\|_F \qquad (30) \\
&\leq \frac{\varepsilon^2}{2n^2} d \max_{\sigma,j} \|\boldsymbol{b}_j^{\sigma}\|_2 + \frac{4\kappa' d \max_{\sigma,j} \|\boldsymbol{b}_j^{\sigma}\|_2}{\min_{j,j'}^2 \|\boldsymbol{b}_j - \boldsymbol{b}_{j'}\|_2^2} \\
&\leq \frac{\varepsilon^2}{2n^2} d \max_{\sigma,j} \|\boldsymbol{b}_j^{\sigma}\|_2 + \frac{4\kappa' d}{\kappa^2 \max_{\sigma,j} \|\boldsymbol{b}_j^{\sigma}\|_2} \\
&= f^{NC} \left( \max_{\sigma,j} \|\boldsymbol{b}_j^{\sigma}\|_2 \right) \\
&= o(1) , \qquad (31)
\end{aligned}
$$

where the last inequality uses assumption (**D1**), and (30) uses the property that $(a+b)^2 \leq 2a^2 + 2b^2$. We have let

$$
f^{NC}(x) \doteq \frac{\varepsilon^2}{2n^2} dx + \frac{4\kappa' d}{\kappa x} , \qquad (32)
$$

which is indeed $o(1)$ if $\varepsilon = o(n^2/\sqrt{x})$. This proves the Lemma for $\varsigma(\mathrm{V}^{NC}, \mathrm{B}^{\pm})$. The case of $\varsigma(\mathrm{V}^{G,s}, \mathrm{B}^{\pm})$ is easier, as

$$
\begin{aligned}
\exp \left( -\frac{\|\boldsymbol{b}_j - \boldsymbol{b}_{j'}\|_2}{s} \right) &\leq \exp \left( -\frac{\min_{j'',j'''} \|\boldsymbol{b}_{j''} - \boldsymbol{b}_{j'''}\|_2}{s} \right) \\
&\leq \exp \left( -\frac{\kappa}{s} \max_{\sigma,j} \|\boldsymbol{b}_j^{\sigma}\|_2 \right) ,
\end{aligned}
$$

from assumption (**D1**) alone, which gives

$$
\begin{aligned}
\varsigma(\mathrm{V}^{G,s}, \mathrm{B}^{\pm}) &\leq \|\mathrm{B}^{\pm}\|_F \left( \frac{\varepsilon}{2n} + \exp \left( -\frac{\kappa}{s} \max_{\sigma,j} \|\boldsymbol{b}_j^{\sigma}\|_2 \right) \right)^2 \\
&\leq \|\mathrm{B}^{\pm}\|_F \left( \frac{\varepsilon^2}{2n^2} + 2 \exp \left( -\frac{2\kappa}{s} \max_{\sigma,j} \|\boldsymbol{b}_j^{\sigma}\|_2 \right) \right) \\
&\leq d \max_{\sigma,j} \|\boldsymbol{b}_j^{\sigma}\|_2 \left( \frac{\varepsilon^2}{2n^2} + 2 \exp \left( -\frac{2\kappa}{s} \max_{\sigma,j} \|\boldsymbol{b}_j^{\sigma}\|_2 \right) \right) \\
&= f^G \left( \max_{\sigma,j} \|\boldsymbol{b}_j^{\sigma}\|_2 \right) \\
&= o(1) , \qquad (33)
\end{aligned}
$$

as claimed. We have let $f^G(x) \doteq \frac{\varepsilon^2}{2n^2} dx + dx \exp(-2\kappa x/s)$, which is indeed $o(1)$ if $\varepsilon = o(n^2/\sqrt{x})$. Remark that we shall have in general $f^G(x) \leq f^{NC}(x)$ and even $f^G(x) = o(f^{NC}(x))$ if $\varepsilon = 0$, so we may expect better convergence in the case of $\mathrm{V}^{G,s}$ as $\max_{\sigma,j} \|\boldsymbol{b}_j^{\sigma}\|_2$ grows.

## 2.5 Proof of Lemma 5

We first restate the Lemma in a more explicit way, that shall provide explicit values for $\kappa_l$ and $\kappa_n$.

**Lemma 2** *There exist $\kappa_{jj'}$ and $s_{jj'}$ depending on $d_j, d_{j'}$, and $\kappa'_{jj'} > 1$ depending on $m_j, m_{j'}$, such that:*

- If $v_{jj'}^{G,s_{jj'}} > \exp(-1/4)$ then $\mathcal{S}_j, \mathcal{S}_{j'}$ are not linearly separable;

- If $v_{jj'}^{G,s_{jj'}} < \exp(-64)$ then $\mathcal{S}_j, \mathcal{S}_{j'}$ are linearly separable;

- If $v_{jj'}^{NC} > \kappa_{jj'}$ then $\mathcal{S}_j, \mathcal{S}_{j'}$ are not linearly separable;

- If $v_{jj'}^{NC} < \kappa_{jj'}/\kappa'_{jj'}$ then $\mathcal{S}_j, \mathcal{S}_{j'}$ are linearly separable.

**Proof** We first consider the normalized association criterion in (10), and we prove the Lemma for the following expressions of $\kappa_{jj'}$ and $\kappa'_{jj'}$:

$$\kappa_{jj'} \doteq \frac{16}{2 + \frac{d_{jj'}^2}{2d_{j'}^2}} + \frac{16}{2 + \frac{d_{jj'}^2}{2d_j^2}} \ , \tag{34}$$

$$\kappa'_{jj'} \doteq 512 \max\{m_j, m_{j'}\} \ , \tag{35}$$

with $d_{jj'} \doteq \max\{d_j, d_{j'}\}$ and $d_j \doteq \max_{\boldsymbol{x}, \boldsymbol{x}' \in \mathcal{S}_j} \|\boldsymbol{x} - \boldsymbol{x}'\|_2, \forall j \neq j' \in [n]$. For any bag $\mathcal{S}_j$, we let $(\boldsymbol{b}_j^\star, r_j) \doteq MEB(\mathcal{S}_j)$ denote the minimum enclosing ball (MEB) for bag $\mathcal{S}_j$ and distance $L_2$, that is, $r_j$ is the smallest unique real such that

$$\exists! \boldsymbol{b}_j^\star : d(\boldsymbol{x}, \boldsymbol{b}_j^\star) \doteq \|\boldsymbol{x} - \boldsymbol{b}_j^\star\|_2 \leq r_j, \forall \boldsymbol{x} \in \mathcal{S}_j \ .$$

We have let $d(\boldsymbol{x}, \boldsymbol{b}_j^\star) \doteq \|\boldsymbol{x} - \boldsymbol{b}_j^\star\|_2$. We are going to prove a first result involving the MEBs of $\mathcal{S}_j$ and $\mathcal{S}_{j'}$, and then will translate the result to the Lemma's statement. The following properties follows from standard properties of MEBs and the fact that $d(.,.)$ is a distance (they hold for any $j \neq j'$):

(a) $d(\boldsymbol{x}, \boldsymbol{x}') \leq 2r_j \ , \forall \boldsymbol{x}, \boldsymbol{x}' \in \mathcal{S}_j$;

(b) If bags $\mathcal{S}_j$ and $\mathcal{S}_{j'}$ are linearly separable, then $\forall \boldsymbol{x} \in \mathrm{CO}(\mathcal{S}_j), \exists \boldsymbol{x}' \in \mathcal{S}_{j'}$ such that $d(\boldsymbol{x}, \boldsymbol{x}') \geq \max\{r_j, r_{j'}\}$; here, "CO" denotes the convex closure;

(c) If bags $\mathcal{S}_j$ and $\mathcal{S}_{j'}$ are linearly separable, then $d(\boldsymbol{b}_j, \boldsymbol{b}_{j'}) \geq \max\{r_j, r_{j'}\}$, where $\boldsymbol{b}_j$ and $\boldsymbol{b}_{j'}$ are the bags average;

(d) $\forall \boldsymbol{x} \in \mathcal{S}_j, \exists \boldsymbol{x}' \in \mathcal{S}_j$ s.t. $d(\boldsymbol{x}, \boldsymbol{x}') \geq r_j$;

(e) $d(\boldsymbol{x}, \boldsymbol{x}') \leq 2\max\{r_j, r_{j'}\} + d(\boldsymbol{b}_j^\star, \boldsymbol{b}_{j'}^\star), \forall \boldsymbol{x} \in \mathrm{CO}(\mathcal{S}_j), \forall \boldsymbol{x}' \in \mathrm{CO}(\mathcal{S}_{j'})$.

Let us define

$$\mathrm{ASSOC}(\mathcal{S}_j, \mathcal{S}_{j'}) \doteq \sum_{\boldsymbol{x} \in \mathcal{S}_j, \boldsymbol{x}' \in \mathcal{S}_{j'}} d^2(\boldsymbol{x}, \boldsymbol{x}') \ . \tag{36}$$

We remark that, assuming that each bag contains at least two elements without loss of generality:

$$v_{jj'}^{NC} = \frac{1}{2} \left( \frac{1}{1 + \frac{\mathrm{ASSOC}(\mathcal{B}_j, \mathcal{B}_{j'})}{\mathrm{ASSOC}(\mathcal{B}_j, \mathcal{B}_j)}} + \frac{1}{1 + \frac{\mathrm{ASSOC}(\mathcal{B}_j, \mathcal{B}_{j'})}{\mathrm{ASSOC}(\mathcal{B}_{j'}, \mathcal{B}_{j'})}} \right) \ . \tag{37}$$

We have $\mathrm{ASSOC}(\mathcal{S}_j, \mathcal{S}_j) \leq 4m_j r_j^2$ and $\mathrm{ASSOC}(\mathcal{S}_{j'}, \mathcal{S}_{j'}) \leq 4m_{j'} r_{j'}^2$ (because of (a)), and also $\mathrm{ASSOC}(\mathcal{S}_j, \mathcal{S}_{j'}) \geq \max\{m_j, m_{j'}\} \max\{r_j^2, r_{j'}^2\}$ when $\mathcal{S}_j$ and $\mathcal{S}_{j'}$ are linearly separable (because of (b)), which yields in this case

$$v_{jj'}^{NC} \leq \frac{1}{2 + \frac{\max\{m_j, m_{j'}\} \max\{r_j^2, r_{j'}^2\}}{2m_j r_j^2}} + \frac{1}{2 + \frac{\max\{m_j, m_{j'}\} \max\{r_j^2, r_{j'}^2\}}{2m_{j'} r_{j'}^2}}$$

$$\leq \frac{1}{2 + \frac{\max\{r_j^2, r_{j'}^2\}}{2r_j^2}} + \frac{1}{2 + \frac{\max\{r_j^2, r_{j'}^2\}}{2r_{j'}^2}} \ . \tag{38}$$

Let us name $\kappa_{jj'}^\circ$ the right-hand side of (38). It follows that when $v_{jj'}^{NC} > \kappa_{jj'}^\circ$, $\mathcal{S}_j$ and $\mathcal{S}_{j'}$ are not linearly separable.

On the other hand, we have $\text{ASSOC}(\mathcal{S}_j, \mathcal{S}_j) \geq m_j r_j^2$ and $\text{ASSOC}(\mathcal{S}_{j'}, \mathcal{S}_{j'}) \geq m_{j'} r_{j'}^2$ (because of (d)), and also

$$
\begin{aligned}
\text{ASSOC}(\mathcal{S}_j, \mathcal{S}_{j'}) &\leq m_j m_{j'} (2\max\{r_j, r_{j'}\} + d(\boldsymbol{b}_j^\star, \boldsymbol{b}_{j'}^\star))^2 \\
&\leq m_j m_{j'} (4\max\{r_j^2, r_{j'}^2\} + 2d^2(\boldsymbol{b}_j^\star, \boldsymbol{b}_{j'}^\star)) \ ,
\end{aligned}
\tag{39}
$$

because of (e) and the fact that $(a+b)^2 \leq 2a^2 + 2b^2$. It follows that $\forall j \neq j'$:

$$
v_{jj'}^{NC} \geq \frac{1}{2 + \frac{2m_{j'}(4\max\{r_j^2, r_{j'}^2\} + 2d^2(\boldsymbol{b}_j^\star, \boldsymbol{b}_{j'}^\star))}{r_j^2}} + \frac{1}{2 + \frac{2m_j(4\max\{r_j^2, r_{j'}^2\} + 2d^2(\boldsymbol{b}_j^\star, \boldsymbol{b}_{j'}^\star))}{r_{j'}^2}} \ .
\tag{40}
$$

For any $j \neq j'$, when $d^2(\boldsymbol{b}_j^\star, \boldsymbol{b}_{j'}^\star) \leq 4\max\{r_j^2, r_{j'}^2\}$, then we have from (40):

$$
\begin{aligned}
v_{jj'}^{NC} &\geq \frac{1}{2 + \frac{16m_{j'}\max\{r_j^2, r_{j'}^2\}}{r_j^2}} + \frac{1}{2 + \frac{16m_j\max\{r_j^2, r_{j'}^2\}}{r_{j'}^2}} \\
&> \kappa_{jj'}^\circ / (32\max\{m_j, m_{j'}\}) \ .
\end{aligned}
\tag{41}
$$

Hence, when $v_{jj'}^{NC} \leq \kappa_{jj'}^\circ / (32\max\{m_j, m_{j'}\})$, it implies $d(\boldsymbol{b}_j^\star, \boldsymbol{b}_{j'}^\star) > 2\max\{r_j, r_{j'}\}$, implying $d(\boldsymbol{b}_j^\star, \boldsymbol{b}_{j'}^\star) > r_j + r_{j'}$, which is a sufficient condition for the linear separability of $\mathcal{S}_j$ and $\mathcal{S}_{j'}$.

So, we can relate the linear separability of $\mathcal{S}_j$ and $\mathcal{S}_{j'}$ to the value of $v_{jj'}^{NC}$ with respect to $\kappa_{jj'}^\circ$ defined in (38). To remove the dependence in the MEB parameters and obtain the statement of the Lemma, we just have to remark that $d_j^2/4 \leq r_j^2 \leq 4d_j^2, \forall j \in [n]$, which yields $\kappa_{jj'}/16 \leq \kappa_{jj'}^\circ \leq \kappa_{jj'}$. Hence, when $v_{jj'}^{NC} > \kappa_{jj'}$, it follows that $v_{jj'}^{NC} > \kappa_{jj'}^\circ$ and $\mathcal{S}_j$ and $\mathcal{S}_{j'}$ are not linearly separable. On the other hand, when $v_{jj'}^{NC} \leq \kappa_{jj'}/(16 \times 32\max\{m_j, m_{j'}\}) = \kappa_{jj'}/\kappa_{jj'}'$, then $v_{jj'}^{NC} \leq \kappa_{jj'}^\circ / (32\max\{m_j, m_{j'}\})$ and the bags $\mathcal{S}_j$ and $\mathcal{S}_{j'}$ are linearly separable. This achieves the proof of Lemma 5 for the normalized association criterion in (10).

The proof for $v_{jj'}^{G,s}$ is shorter, and we prove it for

$$
s_{j,j'} = \max\{d_j, d_{j'}\} \ .
\tag{42}
$$

We have $(1/2)\max\{d_j, d_{j'}\} \leq \max\{r_j, r_{j'}\} \leq 2\max\{d_j, d_{j'}\}$. Hence, because of (c) above, if $\mathcal{S}_j$ and $\mathcal{S}_{j'}$ are linearly separable, then $v_{jj'}^{G,s} \leq 1/e^{1/4}$; so, when $v_{jj'}^{G,s} > 1/e^{1/4}$, the two bags are not linearly separable. On the other hand, if $d(\boldsymbol{b}_j^\star, \boldsymbol{b}_{j'}^\star) \leq 2\max\{r_j, r_{j'}\}$, then because of (e) above $d(\boldsymbol{b}_j, \boldsymbol{b}_{j'}) \leq 4\max\{r_j, r_{j'}\} \leq 8\max\{d_j, d_{j'}\}$, and so $v_{jj'}^{G,s} \geq 1/e^{64}$. This implies that if $v_{jj'}^{G,s} < 1/e^{64}$, then $d(\boldsymbol{b}_j^\star, \boldsymbol{b}_{j'}^\star) > 2\max\{r_j, r_{j'}\} \geq r_j + r_{j'}$, and thus the two bags are linearly separable, as claimed.

This achieves the proof of Lemma 2. ∎

This achieves the proof of Lemma 5.

### 2.6 Mean Map estimator's Lemma and Proof

It is not hard to check that the randomized procedure that builds $\tilde{\mu}_{\mathcal{S}}^{\text{RAND}} \doteq y\boldsymbol{x}$ for some random $\boldsymbol{x} \in \mathcal{S}$ and $y \in \{-1, 1\}$ guarantees $O(2 + \gamma)$ approximability when some bags are close to the convex hull of $\mathcal{S}$, for small $\gamma > 0$. Hence, the Mean Map estimation of $\boldsymbol{\mu}_{\mathcal{S}}$ can be very poor in that respect.

**Lemma 3** *For any $\gamma > 0$, the Mean Map estimator $\tilde{\mu}_{\mathcal{S}}^{\text{MM}}$ cannot guarantee $\|\tilde{\mu}_{\mathcal{S}}^{\text{MM}} - \mu_{\mathcal{S}}\|_2 / \max_{\sigma, j} \|\boldsymbol{b}_j^\sigma\|_2 \leq 2 - \gamma$, even when (D1 + D2) hold.*

**Proof** Let $x > 0, \epsilon \in (0, 1), p \in (0, 1), p \neq 1/2$. We create a dataset from four observations, $\{(x_1 = 0, 1), (x_2 = 0, -1), (x_3 = x, 1), (x_4 = x, -1)\}$. There are two bags, $\mathcal{S}_1$ takes $1 - \epsilon$ of $x_2$ and $\epsilon$ of $x_1$. $\mathcal{S}_2$ takes $\epsilon$ of $x_4$ and $1 - \epsilon$ of $x_3$. The label-wise estimators $\tilde{\mu}^\sigma$ of [3] are solution of

$$
\begin{aligned}
\begin{bmatrix} \tilde{\mu}^1 \\ \tilde{\mu}^{-1} \end{bmatrix} &= \left( \begin{bmatrix} 1 - \epsilon & \epsilon \\ \epsilon & 1 - \epsilon \end{bmatrix}^\top \begin{bmatrix} 1 - \epsilon & \epsilon \\ \epsilon & 1 - \epsilon \end{bmatrix} \right)^{-1} \begin{bmatrix} 1 - \epsilon & \epsilon \\ \epsilon & 1 - \epsilon \end{bmatrix}^\top \begin{bmatrix} x \\ 0 \end{bmatrix} \\
&= \frac{1}{1 - 2\epsilon} \begin{bmatrix} (1 - \epsilon)x \\ \epsilon x \end{bmatrix}
\end{aligned}
\tag{43}
$$

On the other hand, the true quantities are:

$$\begin{bmatrix} \mu^1 \\ \mu^{-1} \end{bmatrix} = \begin{bmatrix} (1-\epsilon)x \\ \epsilon x \end{bmatrix} . \tag{44}$$

We now mix classes in $\mathcal{S}$ and pick bag proportions $q \doteq \mathbb{P}_{\mathcal{S}}[\mathcal{S}_1]$ and $1 - q = \mathbb{P}_{\mathcal{S}}[\mathcal{S}_2]$. We have the class proportions defined by $\mathbb{P}_{\mathcal{S}}[y = +1] = \epsilon q + (1-\epsilon)(1-q) \doteq p$. Then

$$\begin{aligned} |\tilde{\mu}_{\mathcal{S}} - \mu_{\mathcal{S}}| &= \left| p(1-\epsilon)\left( \frac{1}{1-2\epsilon} - 1 \right) x - (1-p)\epsilon \left( \frac{1}{1-2\epsilon} - 1 \right) x \right| \\ &= \frac{2\epsilon|p - \epsilon|}{1 - 2\epsilon} x \\ &= 2\epsilon(1-q)x . \end{aligned} \tag{45}$$

Furthermore, $\max_i |b_i^\sigma| = x$. We get

$$\frac{|\tilde{\mu}_{\mathcal{S}} - \mu_{\mathcal{S}}|}{\max_i |b_i^\sigma|} = 2\epsilon(1-q) . \tag{46}$$

Picking $\epsilon$ and $(1-q)$ both $> \sqrt{1 - (\gamma/2)}$ is sufficient to have eq. (46) $> 2 - \gamma$ for any $\gamma > 0$. Remark that both assumptions (**D1**) and (**D2**) hold for any $\kappa < 1$ and any $\kappa' > 0$. ∎

## 2.7 Proof of Theorem 6

The proof of the Theorem involves two Lemmata, the first of which is of independent interest and holds for any convex twice differentiable function $F$, and not just any $F_\phi$. So, let us define:

$$F(\mathcal{S}_{|y}, \boldsymbol{\theta}, \boldsymbol{\mu}) = \frac{b}{2m}\left( \sum_i \sum_\sigma F(\sigma\boldsymbol{\theta}^\top \boldsymbol{x}_i) \right) - \frac{1}{2}\boldsymbol{\theta}^\top \boldsymbol{\mu} . \tag{47}$$

where $b$ is any fixed positive real. Define also the regularized loss:

$$F(\mathcal{S}_{|y}, \boldsymbol{\theta}, \boldsymbol{\mu}, \lambda) \doteq F(\mathcal{S}_{|y}, \boldsymbol{\theta}, \boldsymbol{\mu}) + \lambda\|\boldsymbol{\theta}\|_2^2 . \tag{48}$$

Let $\boldsymbol{f}_k \in \mathbb{R}^m$ denote the vector encoding the $k^{th}$ variable in $\mathcal{S} : f_{ki} = x_{ik}$. For any $k \in [d]$, let

$$\tilde{\boldsymbol{f}}_k \doteq \left( \frac{d}{\sum_k \|\boldsymbol{f}_k\|_2^2} \right)^{\frac{d-1}{2d}} \boldsymbol{f}_k \tag{49}$$

denote a normalization of vectors $\boldsymbol{f}_k$ in the sense that

$$\begin{aligned} \frac{1}{d}\sum_k \|\tilde{\boldsymbol{f}}_k\|_2^2 &= \frac{1}{d}\left( \frac{d}{\sum_k \|\boldsymbol{f}_k\|_2^2} \right)^{1 - \frac{1}{d}} \sum_k \|\boldsymbol{f}_k\|_2^2 \\ &= \left( \frac{1}{d}\sum_k \|\boldsymbol{f}_k\|_2^2 \right)^{\frac{1}{d}} . \end{aligned} \tag{50}$$

Let $\tilde{\mathsf{V}}$ collect all vectors $\tilde{\boldsymbol{f}}_k$ in column and $\mathsf{V}$ collect all vectors $\boldsymbol{f}_k$ in column. Without loss of generality, we assume $\mathsf{V}^\top \mathsf{V} \succ 0$, *i.e.* $\mathsf{V}^\top \mathsf{V}$ positive definite (*i.e.* no feature is a linear combination of the others), implying, because the columns of $\tilde{\mathsf{V}}$ are just positive rescaling of the columns of $\mathsf{V}$, that $\tilde{\mathsf{V}}^\top \tilde{\mathsf{V}} \succ 0$ as well. We use $\mathsf{V}$ instead of $F$ as in the main paper, in order not to counfound with the general convex surrogate notation $F$ that we use here.

**Lemma 4** *Given any two $\boldsymbol{\mu}$ and $\boldsymbol{\mu}'$, let $\boldsymbol{\theta}_*$ and $\boldsymbol{\theta}'_*$ be the respective minimizers of $F(\mathcal{S}_{|y}, ., \boldsymbol{\mu}, \lambda)$ and $F(\mathcal{S}_{|y}, ., \boldsymbol{\mu}', \lambda)$. Suppose there exists $F''_\circ > 0$ such that surrogate $F$ satisfies*

$$F''(\pm(\alpha\boldsymbol{\theta}_* + (1-\alpha)\boldsymbol{\theta}'_*)^\top \boldsymbol{x}_i) \geq F''_\circ , \forall \alpha \in [0,1], \forall i \in [m] . \tag{51}$$

*Then the following holds:*

$$\|\boldsymbol{\theta}_* - \boldsymbol{\theta}'_*\|_2 \leq \frac{1}{2\lambda + \frac{2}{em}F''_\circ \mathrm{vol}^2(\tilde{\mathsf{V}})} \|\boldsymbol{\mu} - \boldsymbol{\mu}'\|_2 , \tag{52}$$

*where $\mathrm{vol}(\tilde{\mathsf{V}}) \doteq \sqrt{\det \tilde{\mathsf{V}}^\top \tilde{\mathsf{V}}}$ denote the volume of the (row/column) system of $\tilde{\mathsf{V}}$.*

**Proof** Our proof begins following the same first steps as the proof of Lemma 17 in [4], adding the steps that handle the lowerbound on $F''$. Consider the following auxiliary function $A_F(\tau)$:

$$A_F(\tau) \;\doteq\; \left(\nabla F(\mathcal{S}_{|y}, \boldsymbol{\theta}_*, \boldsymbol{\mu}) - \nabla F(\mathcal{S}_{|y}, \boldsymbol{\theta}'_*, \boldsymbol{\mu}')\right)^\top (\tau - \boldsymbol{\theta}'_*) + \lambda\|\tau - \boldsymbol{\theta}'_*\|_2^2 \;, \qquad (53)$$

where the gradient $\nabla$ of $F$ is computed with respect to parameter $\boldsymbol{\theta}$. The gradient of $A_F(.)$ is:

$$\nabla A_F(\tau) \;=\; \nabla F(\mathcal{S}_{|y}, \boldsymbol{\theta}_*, \boldsymbol{\mu}) - \nabla F(\mathcal{S}_{|y}, \boldsymbol{\theta}'_*, \boldsymbol{\mu}') + 2\lambda(\tau - \boldsymbol{\theta}'_*) \;, \qquad (54)$$

The gradient of $A_F$ satisfies

$$\begin{aligned}
\nabla A_F(\boldsymbol{\theta}_*) &= \nabla F(\mathcal{S}_{|y}, \boldsymbol{\theta}_*, \boldsymbol{\mu}, \lambda) - \nabla F(\mathcal{S}_{|y}, \boldsymbol{\theta}'_*, \boldsymbol{\mu}', \lambda) \\
&= \mathbf{0} \;, 
\end{aligned} \qquad (55)$$

as both gradients in the right are $\mathbf{0}$ because of the optimality of $\boldsymbol{\theta}_*$ and $\boldsymbol{\theta}'_*$ with respect to $F(\mathcal{S}_{|y}, ., \boldsymbol{\mu}, \lambda)$ and $F(\mathcal{S}_{|y}, ., \boldsymbol{\mu}', \lambda)$. The Hessian H of $A_F$ is $\mathrm{H} A_F(\tau) = 2\lambda \mathrm{I} \succeq 0$ and so $A_F$ is convex and is thus minimal at $\tau = \boldsymbol{\theta}_*$. Finally, $A_F(\boldsymbol{\theta}'_*) = 0$. It comes thus $A_F(\boldsymbol{\theta}_*) \leq 0$, which yields equivalently:

$$\begin{aligned}
0 \;\geq\; & \left(\nabla F(\mathcal{S}_{|y}, \boldsymbol{\theta}_*, \boldsymbol{\mu}) - \nabla F(\mathcal{S}_{|y}, \boldsymbol{\theta}'_*, \boldsymbol{\mu}')\right)^\top (\boldsymbol{\theta}_* - \boldsymbol{\theta}'_*) + \lambda\|\boldsymbol{\theta}_* - \boldsymbol{\theta}'_*\|_2^2 \\
=\; & \left(\frac{b}{2m}\sum_y\sum_i \nabla F(y\boldsymbol{\theta}_*^\top \boldsymbol{x}_i) - \frac{1}{2}\boldsymbol{\mu} - \frac{b}{2m}\sum_y\sum_i \nabla F(y\boldsymbol{\theta}'^\top_* \boldsymbol{x}_i) + \frac{1}{2}\boldsymbol{\mu}'\right)^\top (\boldsymbol{\theta}_* - \boldsymbol{\theta}'_*) \\
& + \lambda\|\boldsymbol{\theta}_* - \boldsymbol{\theta}'_*\|_2^2 \\
=\; & \frac{b}{2m}\underbrace{\left(\sum_y\sum_i \nabla F(y\boldsymbol{\theta}_*^\top \boldsymbol{x}_i) - \sum_y\sum_i \nabla F(y\boldsymbol{\theta}'^\top_* \boldsymbol{x}_i)\right)^\top (\boldsymbol{\theta}_* - \boldsymbol{\theta}'_*)}_{\doteq a} \\
& -\frac{1}{2}(\boldsymbol{\mu} - \boldsymbol{\mu}')^\top (\boldsymbol{\theta}_* - \boldsymbol{\theta}'_*) + \lambda\|\boldsymbol{\theta}_* - \boldsymbol{\theta}'_*\|_2^2 \;.
\end{aligned} \qquad (56)$$

Let us lowerbound $a$. We have $\nabla F(y\boldsymbol{\theta}_*^\top \boldsymbol{x}) = yF'(y\boldsymbol{\theta}_*^\top \boldsymbol{x})\boldsymbol{x}$, and a Taylor expansion brings that for any $\boldsymbol{\theta}_*, \boldsymbol{\theta}'_*$, there exists some $\alpha \in [0,1]$ such that, defining

$$u_{\alpha,i} \;\doteq\; y(\alpha\boldsymbol{\theta}_* + (1-\alpha)\boldsymbol{\theta}'_*)^\top \boldsymbol{x}_i \;, \qquad (57)$$

we have:

$$F'(y\boldsymbol{\theta}_*^\top \boldsymbol{x}_i) \;=\; F'(y\boldsymbol{\theta}'^\top_* \boldsymbol{x}_i) + y(\boldsymbol{\theta}_* - \boldsymbol{\theta}'_*)^\top \boldsymbol{x}_i F''(u_{\alpha,i}) \;. \qquad (58)$$

We thus get:

$$\begin{aligned}
a \;=\; & \left(\sum_y\sum_i \nabla F(y\boldsymbol{\theta}_*^\top \boldsymbol{x}_i) - \sum_y\sum_i \nabla F(y\boldsymbol{\theta}'^\top_* \boldsymbol{x}_i)\right)^\top (\boldsymbol{\theta}_* - \boldsymbol{\theta}'_*) \\
=\; & \left(\sum_y\sum_i y(F'(y\boldsymbol{\theta}_*^\top \boldsymbol{x}_i) - F'(y\boldsymbol{\theta}'^\top_* \boldsymbol{x}_i))\boldsymbol{x}_i\right)^\top (\boldsymbol{\theta}_* - \boldsymbol{\theta}'_*) \\
=\; & \left(\sum_y\sum_i (\boldsymbol{\theta}_* - \boldsymbol{\theta}'_*)^\top \boldsymbol{x}_i F''(u_{\alpha,i})\boldsymbol{x}_i\right)^\top (\boldsymbol{\theta}_* - \boldsymbol{\theta}'_*) \\
=\; & 2\sum_i ((\boldsymbol{\theta}_* - \boldsymbol{\theta}'_*)^\top \boldsymbol{x}_i)^2 F''(u_{\alpha,i}) \\
\geq\; & 2F''_\circ \sum_i ((\boldsymbol{\theta}_* - \boldsymbol{\theta}'_*)^\top \boldsymbol{x}_i)^2 \qquad (59) \\
=\; & 2F''_\circ (\boldsymbol{\theta}_* - \boldsymbol{\theta}'_*)^\top \mathrm{S}\mathrm{S}^\top (\boldsymbol{\theta}_* - \boldsymbol{\theta}'_*) \;, \qquad (60)
\end{aligned}$$

where matrix $\mathrm{S} \in \mathbb{R}^{d \times m}$ is formed by the observations of $\mathcal{S}_{|y}$ in columns, and ineq. (59) comes from (51). Define $\mathrm{T} \doteq (d/\sum_i \|\boldsymbol{x}_i\|_2^2)\mathrm{S}\mathrm{S}^\top$. Its trace satisfies $\mathrm{tr}(\mathrm{T}) = d$. Let $\lambda_d \geq \lambda_{d-1} \geq ... \geq \lambda_1 > 0$

denote eigenvalues of $\mathrm{T}$, with $\lambda_1$ strictly positive because $\mathrm{SS}^\top = \mathrm{V}^\top \mathrm{V} \succ 0$. The AGH inequality brings:

$$
\begin{aligned}
\prod_2^d \lambda_k \;&\leq\; \left( \frac{1}{d-1} \sum_{k=2}^d \lambda_k \right)^{d-1} &&(61)\\
&= \left( \frac{\mathrm{tr}\,(\mathrm{T}) - \lambda_1}{d-1} \right)^{d-1}\\
&= \left( \frac{d - \lambda_1}{d-1} \right)^{d-1}\\
&\leq \left( \frac{d}{d-1} \right)^{d-1} . &&(62)
\end{aligned}
$$

Multiplying both side by $\lambda_1$ and rearranging yields:

$$
\lambda_1 \;\geq\; \left( \frac{d-1}{d} \right)^{d-1} \det \mathrm{T} \tag{63}
$$

Let $\lambda_\circ > 0$ denote the minimal eigenvalue of $\mathrm{SS}^\top$. It satisfies $\lambda_\circ = (\sum_i \|\boldsymbol{x}_i\|_2^2 / d)\lambda_1$ and thus it comes from ineq. (63):

$$
\begin{aligned}
\lambda_\circ \;&\geq\; \left( \frac{d-1}{d} \right)^{d-1} \left( \frac{d}{\sum_i \|\boldsymbol{x}_i\|_2^2} \right)^{d-1} \det \mathrm{SS}^\top\\
&= \left( \frac{d-1}{d} \right)^{d-1} \det \left[ \left( \frac{d}{\sum_i \|\boldsymbol{x}_i\|_2^2} \right)^{1-\frac{1}{d}} \mathrm{SS}^\top \right]\\
&= \left( \frac{d-1}{d} \right)^{d-1} \det \tilde{\mathrm{V}}^\top \tilde{\mathrm{V}} &&(64)\\
&= \left( \frac{d-1}{d} \right)^{d-1} \mathrm{vol}^2(\tilde{\mathrm{V}}) &&(65)\\
&\geq \frac{1}{e} \mathrm{vol}^2(\tilde{\mathrm{V}}) . &&(66)
\end{aligned}
$$

We have used notation $\mathrm{vol}(\tilde{\mathrm{V}}) \doteq \sqrt{\det \tilde{\mathrm{V}}^\top \tilde{\mathrm{V}}}$. Since $(\boldsymbol{\theta}_* - \boldsymbol{\theta}'_*)^\top \mathrm{SS}^\top (\boldsymbol{\theta}_* - \boldsymbol{\theta}'_*) \geq \lambda_\circ \|\boldsymbol{\theta}_* - \boldsymbol{\theta}'_*\|_2^2$, combining (60) with (66) yields the following lowerbound on $a$:

$$
a \;\geq\; \frac{2}{e} F''_\circ \mathrm{vol}^2(\tilde{\mathrm{V}}) \|\boldsymbol{\theta}_* - \boldsymbol{\theta}'_*\|_2^2 . \tag{67}
$$

Going back to (56), we get

$$
\lambda \|\boldsymbol{\theta}_* - \boldsymbol{\theta}'_*\|_2^2 - \frac{1}{2} (\boldsymbol{\mu} - \boldsymbol{\mu}')^\top (\boldsymbol{\theta}_* - \boldsymbol{\theta}'_*) + \frac{b}{em} F''_\circ \mathrm{vol}^2(\tilde{\mathrm{V}}) \|\boldsymbol{\theta}_* - \boldsymbol{\theta}'_*\|_2^2 \;\leq\; 0 .
$$

Since $(\boldsymbol{\mu} - \boldsymbol{\mu}')^\top (\boldsymbol{\theta}_* - \boldsymbol{\theta}'_*) \leq \|\boldsymbol{\mu} - \boldsymbol{\mu}'\|_2 \|\boldsymbol{\theta}_* - \boldsymbol{\theta}'_*\|_2$, we get after chaining the inequalities and solving for $\|\boldsymbol{\theta}_* - \boldsymbol{\theta}'_*\|_2$:

$$
\|\boldsymbol{\theta}_* - \boldsymbol{\theta}'_*\|_2 \;\leq\; \frac{1}{2\lambda + \frac{2}{em} F''_\circ \mathrm{vol}^2(\tilde{\mathrm{V}})} \|\boldsymbol{\mu} - \boldsymbol{\mu}'\|_2 ,
$$

as claimed. ∎

The second Lemma is used to (51) when $F(x) = F_\phi$. Notice that we cannot rely on strong convexity arguments on $F_\phi$, as this do not hold in general. The Lemma is stated in a more general setting than for just $F = F_\phi$.

**Lemma 5** *Fix $\lambda, b > 0$, and let $x_* \doteq \max_i \|\boldsymbol{x}_i\|_2$. Suppose that $\|\boldsymbol{\mu}\|_2 \leq \mu_*$ for some $\mu > 0$. Let*

$$F(\mathcal{S}_{|y}, \boldsymbol{\theta}, \boldsymbol{\mu}, \lambda) = \frac{b}{2m}\left(\sum_i \sum_\sigma F(\sigma\boldsymbol{\theta}^\top \boldsymbol{x}_i)\right) - \frac{1}{2}\boldsymbol{\theta}^\top\boldsymbol{\mu} + \lambda\|\boldsymbol{\theta}\|_2^2 \ , \tag{68}$$

*and let $\boldsymbol{\theta}_* \doteq \arg\min_{\boldsymbol{\theta}} F(\mathcal{S}_{|y}, \boldsymbol{\theta}, \boldsymbol{\mu}, \lambda)$. Suppose that $F(.)$ is L-Lipschitz. Then*

$$\|\boldsymbol{\theta}_*\|_2 \leq \frac{bLx_* + \mu_*}{\lambda} \ . \tag{69}$$

**Proof** Let us define a shrinking of the optimal solution $\boldsymbol{\theta}_*$, $\boldsymbol{\theta}_\alpha \doteq \alpha\boldsymbol{\theta}_*$ for $\alpha \in (0,1)$. We have

$$\begin{aligned}
F(\mathcal{S}_{|y}, \boldsymbol{\theta}_\alpha, \boldsymbol{\mu}, \lambda) &= \frac{b}{2m}\left(\sum_i \sum_\sigma F(\sigma\boldsymbol{\theta}_\alpha^\top \boldsymbol{x}_i)\right) - \frac{1}{2}\boldsymbol{\theta}_\alpha^\top\boldsymbol{\mu} + \lambda\|\boldsymbol{\theta}_\alpha\|_2^2 \\
&= \frac{b}{2m}\left(\sum_i \sum_\sigma F(\sigma\alpha\boldsymbol{\theta}_*^\top \boldsymbol{x}_i)\right) - \frac{\alpha}{2}\boldsymbol{\theta}_*^\top\boldsymbol{\mu} + \lambda\alpha^2\|\boldsymbol{\theta}_*\|_2^2 \\
&\leq \frac{b}{2m}\left(\sum_i \sum_\sigma F(\sigma\boldsymbol{\theta}_*^\top \boldsymbol{x}_i) + L\left|\sigma\alpha\boldsymbol{\theta}_*^\top \boldsymbol{x}_i - \sigma\boldsymbol{\theta}_*^\top \boldsymbol{x}_i\right|\right) + -\frac{\alpha}{2}\boldsymbol{\theta}_*^\top\boldsymbol{\mu} \\
&\qquad + \lambda\alpha^2\|\boldsymbol{\theta}_*\|_2^2 \tag{70} \\
&= \frac{b}{2m}\left(\sum_i \sum_\sigma F(\sigma\boldsymbol{\theta}_*^\top \boldsymbol{x}_i)\right) + \frac{bK(1-\alpha)}{m}\sum_i |\boldsymbol{\theta}_*^\top \boldsymbol{x}_i| - \frac{\alpha}{2}\boldsymbol{\theta}_*^\top\boldsymbol{\mu} \\
&\qquad + \lambda\alpha^2\|\boldsymbol{\theta}_*\|_2^2 \ , \tag{71}
\end{aligned}$$

where (70) holds because $F$ is $L$-Lipschitz. To have eq. (71) smaller than $F(\mathcal{S}_{|y}, \boldsymbol{\theta}_*, \boldsymbol{\mu}, \lambda)$, we need equivalently:

$$\frac{bL(1-\alpha)}{m}\sum_i |\boldsymbol{\theta}_*^\top \boldsymbol{x}_i| - \frac{\alpha}{2}\boldsymbol{\theta}_*^\top\boldsymbol{\mu} + \lambda\alpha^2\|\boldsymbol{\theta}_*\|_2^2 \leq -\frac{1}{2}\boldsymbol{\theta}_*^\top\boldsymbol{\mu} + \lambda\|\boldsymbol{\theta}_*\|_2^2 \ ,$$

that is:

$$\frac{bL(1-\alpha)}{m}\sum_i |\boldsymbol{\theta}_*^\top \boldsymbol{x}_i| + \frac{1-\alpha}{2}\boldsymbol{\theta}_*^\top\boldsymbol{\mu} \leq \lambda(1-\alpha^2)\|\boldsymbol{\theta}_*\|_2^2 \ ,$$

and to find an $\alpha \in (0,1)$ such that this holds, because of Cauchy-Schwartz inequality, it is sufficient that $(1-\alpha)(bLx_* + \mu) \leq \lambda(1-\alpha^2)\|\boldsymbol{\theta}_*\|_2$, *i.e.*:

$$\|\boldsymbol{\theta}_*\|_2 \geq \frac{bLx_* + \|\boldsymbol{\mu}\|_2}{\lambda(1+\alpha)} \ .$$

Hence, whenever $\|\boldsymbol{\theta}_*\|_2 > (bLx_* + \|\boldsymbol{\mu}\|_2)/\lambda$, there is a shrinking of the optimal solution to eq. (68) that further decreases the risk, thus contradicting its optimality. This ends the proof of Lemma 5. ∎

Notice that Lemma 5 does not require $F(x)$ to be convex, nor differentiable. To use this Lemma, remark that for any $F_\phi$,

$$F_\phi'(x) = -\frac{1}{b_\phi}(\phi^\star)'(-x) = -\frac{1}{b_\phi}(\phi')^{-1}(-x) \in [-1/b_\phi, 0] \ , \tag{72}$$

for any $x \in \phi'([0,1])$ [1], and thus $F_\phi$ is $1/b_\phi$-Lipschitz. Finally, considering (51), for any $\alpha \in [0,1]$

$$\begin{aligned}
|\pm(\alpha\boldsymbol{\theta}_* + (1-\alpha)\boldsymbol{\theta}_*')^\top \boldsymbol{x}_i| &\leq (\alpha\|\boldsymbol{\theta}_*\|_2 + (1-\alpha)\|\boldsymbol{\theta}_*'\|_2)x_* \\
&\leq \frac{x_* + \alpha\|\boldsymbol{\mu}\|_2 + (1-\alpha)\|\boldsymbol{\mu}'\|_2}{\lambda} \tag{73} \\
&\leq \frac{x_* + \max\{\|\boldsymbol{\mu}\|_2, \|\boldsymbol{\mu}'\|_2\}}{\lambda} \ , \tag{74}
\end{aligned}$$

where ineq. (73) uses Lemma 5 with $b = 1/K = b_\phi$. $\boldsymbol{\mu}$ and $\boldsymbol{\mu}'$ are the parameters of $F(\mathcal{S}_{|y}, ., \boldsymbol{\mu}, \lambda)$ and $F(\mathcal{S}_{|y}, ., \boldsymbol{\mu}', \lambda)$ in Lemma 4.

---

**Algorithm 1** Label Assignation (LA)

---

**Input** $\boldsymbol{\theta} \in \mathbb{R}^d$, a bag $\mathcal{B} = \{\boldsymbol{x}_i \in \mathbb{R}^d, i = 1, 2, ..., m\}$, bag size $m^+ \in [m]$;
**If** $\mathcal{B} = \emptyset$ **then** stop
**Else if** $m^+ \notin (m)$ **then** $y_i \leftarrow \mathrm{I}(m^+ = m) - \mathrm{I}(m^+ = 0), \forall i = 1, 2, ..., m$
**Else**
  Step 1 : $i^* \leftarrow \arg\max_i |\boldsymbol{\theta}^\top \boldsymbol{x}_i|$
  Step 2 : $y_{i^*} \leftarrow \mathrm{sign}(\boldsymbol{\theta}^\top \boldsymbol{x}_{i^*})$
  Step 3 : LA$(\boldsymbol{\theta}, \mathcal{B} \backslash \{\boldsymbol{x}_{i^*}\}, m^+ - \mathrm{I}(y_{i^*} = 1))$

---

Now, going back to the parameters of Theorem 6, we make the change $\boldsymbol{\mu} \to \boldsymbol{\mu}_{\mathrm{S}}$ and $\boldsymbol{\mu}' \to \tilde{\boldsymbol{\mu}}_{\mathrm{S}}$ and obtain the statement of the Theorem for interval

$$\mathbb{I} \quad = \quad [\pm(x_* + \max\{\|\boldsymbol{\mu}_{\mathrm{S}}\|_2, \|\tilde{\boldsymbol{\mu}}_{\mathrm{S}}\|_2\})] \quad . \tag{75}$$

This achieves the proof of Theorem 6.

### 2.8 Proof of Lemma 7

We make the proof for optimization strategy OPT $=$ min. The case OPT $=$ max flips the choice of the label in Step 2. To minimize $F_\phi(\mathcal{S}_{|y}, \boldsymbol{\theta}_t, \boldsymbol{\mu}_{\mathrm{S}}(\boldsymbol{\sigma}))$ over $\boldsymbol{\sigma} \in \Sigma_{\hat{\boldsymbol{\pi}}}$, we just have to find $\boldsymbol{\sigma}_* \in \arg\max_{\boldsymbol{\sigma} \in \Sigma_{\hat{\boldsymbol{\pi}}}} \boldsymbol{\theta}^\top \sum_i \sigma_i \boldsymbol{x}_i$, and we can do that bag-wise. Algorithm 1 presents the labeling (notation $(m) \doteq \{1, 2, ..., m-1\}$). Remark that the time complexity for one bag is $O(m_j \log m_j)$ due to the ordering (Step 1), so the overall complexity is indeed $O(m \max_i \log m_i)$.

**Lemma 6** *Let* $\boldsymbol{\sigma}_* \doteq \{\sigma_1^*, \sigma_2^*, ..., \sigma_m^*\}$ *be the set of labels obtained after running* LA$(\boldsymbol{\theta}, \mathcal{S}_j, m_j^+)$ *for* $j = 1, 2, ..., n$. *Then* $\boldsymbol{\sigma}_* \in \arg\max_{\boldsymbol{\sigma} \in \Sigma_{\hat{\boldsymbol{\pi}}}} \boldsymbol{\theta}^\top \sum_i \sigma_i \boldsymbol{x}_i$.

**Proof** The total edge, $\boldsymbol{\theta}^\top \sum_i \sigma_i \boldsymbol{x}_i$ (for any $\boldsymbol{\sigma} \in \Sigma_{\hat{\boldsymbol{\pi}}}$), can be summable bag-wise wrt the coordinates of $\boldsymbol{\sigma}$. Consider thus the optimal set $\{\boldsymbol{\sigma}^\star\}_{\mathcal{B}} \doteq \arg\max_{\boldsymbol{\sigma} \in \{-1,1\}^{m'} : \mathbf{1}^\top \boldsymbol{\sigma} = 2m^+ - m'} \boldsymbol{\theta}^\top \sum_{\boldsymbol{x}_i \in \mathcal{B}} \sigma_i \boldsymbol{x}_i$, for some bag $\mathcal{B} = \{\boldsymbol{x}_i, i = 1, 2, ..., m'\}$, with constraint $m^+ \in [m']$. This set contains the label assignment $\boldsymbol{\sigma}_*$ returned by LA$(\boldsymbol{\theta}, \mathcal{B}, m^+)$, a property that follows from two simple observations:

**P1** Consider any observation $\boldsymbol{x}_i$ of bag $\mathcal{B}$; for any optimal labeling $\boldsymbol{\sigma}^\star$ of $\mathcal{B}$, let $m'^+ \doteq m^+ - \mathrm{I}(\sigma_i^\star = 1)$. Define the set $\{\boldsymbol{\sigma}'^\star\}_i$ of optimal labelings of $\mathcal{B} \backslash \{\boldsymbol{x}_i\}$ with constraint $m'^+ \doteq m^+ - \mathrm{I}(\sigma_i^\star = 1)$. Then this set coincides with the set created by taking the elements of $\{\boldsymbol{\sigma}^\star\}_{\mathcal{B}}$ to which we drop coordinate $i$. This follows from the per-observation summability of the total edge wrt labels.

**P2** Assume $m^+ \in (m')$. $\forall i^* \in \arg\max_i |\boldsymbol{\theta}^\top \boldsymbol{x}_i|$, there exists an optimal assignment $\boldsymbol{\sigma}^\star$ such that $\sigma_{i^*}^\star = \mathrm{sign}(\boldsymbol{\theta}^\top \boldsymbol{x}_{i^*})$. Otherwise, starting from any optimal assignment $\boldsymbol{\sigma}^\star$, we can flip the label of $\boldsymbol{x}_{i^*}$ and the label of any other $\boldsymbol{x}_i$ for which $\sigma_i^\star \neq \sigma_{i^*}^\star$, and get a label assignment that satisfies constraint $m^+$ and cannot be worse than $\boldsymbol{\sigma}^\star$, and is thus optimal, a contradiction.

Hence, LA$(\boldsymbol{\theta}, \mathcal{B}, m^+)$ picks at each iteration a label that matches one in a subset of optimal labelings, and the recursive call preserves the subset of optimal labelings. Since when $m^+ \notin (m)$ the solution returned by LA$(\boldsymbol{\theta}, \mathcal{B}, m^+)$ is obviously optimal, we end up when the current $\mathcal{B}$ is empty with $\boldsymbol{\sigma}_* \in \arg\max_{\boldsymbol{\sigma} \in \Sigma_{\hat{\boldsymbol{\pi}}}} \boldsymbol{\theta}^\top \sum_i \sigma_i \boldsymbol{x}_i$, as claimed. $\blacksquare$

### 2.9 Proof of Theorem 8

We prove separately Eqs (14) and (15).

### 2.9.1 Proof of eq. (14)

**Notations** : unless explicitly stated, all samples like $\mathcal{S}$ and $\mathcal{S}'$ are of size $m$. To make the reading of our expectations clear and simple, we shall write $\mathbb{E}_{\mathcal{D}}$ for $\mathbb{E}_{(\boldsymbol{x},y)\sim\mathcal{D}}$, $\mathbb{E}_{\Sigma_m}$ for $\mathbb{E}_{\boldsymbol{\sigma}\sim\Sigma_m}$, $\mathbb{E}_{\mathcal{S}}$ for $\mathbb{E}_{(\boldsymbol{x},y)\sim\mathcal{S}}$, $\mathbb{E}_{\mathcal{D}'_m}$ for $\mathbb{E}_{\mathcal{S}'\sim\mathcal{D}}$ and $\mathbb{E}_{\mathcal{D}_m}$ for $\mathbb{E}_{\mathcal{S}\sim\mathcal{D}}$.

We now proceed to the proof, that follows the same main steps as that of Theorem 5 in [5]. For any $q \in [0,1]$, let us define the convex combination:

$$F_\phi(q, h(\boldsymbol{x})) \;\doteq\; q F_\phi(h(\boldsymbol{x})) + (1-q)F_\phi(-h(\boldsymbol{x})) \;. \tag{76}$$

It follows that

$$\mathbb{E}_{\Sigma_{\hat{\boldsymbol{\pi}}}}\mathbb{E}_{\mathcal{S}}[F_\phi(\sigma(\boldsymbol{x})h(\boldsymbol{x}))] \;=\; \mathbb{E}_{\mathcal{S}}[F_\phi(\hat{\pi}(\boldsymbol{x}), h(\boldsymbol{x}))] \;, \tag{77}$$

with $\hat{\pi}(\boldsymbol{x})$ the label proportion of the bag to which $\boldsymbol{x}$ belongs in $\mathcal{S}$. We also have $\forall h$,

$$\mathbb{E}_{\mathcal{D}}[F_\phi(yh(\boldsymbol{x}))] \;\leq\; \mathbb{E}_{\mathcal{S}}[F_\phi(\hat{\pi}(\boldsymbol{x}), h(\boldsymbol{x}))] + \Lambda(\mathcal{S}) \;, \tag{78}$$

with

$$\Lambda(\mathcal{S}) \;\doteq\; \sup_g \left\{ \mathbb{E}_{\mathcal{D}}[F_\phi(yg(\boldsymbol{x}))] - \mathbb{E}_{\mathcal{S}}[F_\phi(\hat{\pi}(\boldsymbol{x}), g(\boldsymbol{x}))] \right\} \;. \tag{79}$$

Let us bound the deviations of $\Lambda(\mathcal{S})$ around its expectation on the sampling of $\mathcal{S}$, using the independent bounded differences inequality (IBDI, [6]). for which we need to upperbound the maximum difference for the supremum term computed over two samples $\mathcal{S}$ and $\mathcal{S}'$ of the same size, such that $\mathcal{S}'$ is $\mathcal{S}$ with one example replaced. We have:

$$|\Lambda(\mathcal{S}) - \Lambda(\mathcal{S}')| \;\leq\; |\mathbb{E}_{\mathcal{S}}[F_\phi(\hat{\pi}(\boldsymbol{x}), g(\boldsymbol{x}))] - \mathbb{E}_{\mathcal{S}'}[F_\phi(\hat{\pi}'(\boldsymbol{x}), g(\boldsymbol{x}))]| \;, \tag{80}$$

with $\hat{\boldsymbol{\pi}}$ and $\hat{\boldsymbol{\pi}}'$ denoting the corresponding label proportions in $\mathcal{S}$ and $\mathcal{S}'$. Let $\{\boldsymbol{x}_1\} = \mathcal{S}\backslash\mathcal{S}'$ and $\{\boldsymbol{x}_2\} = \mathcal{S}'\backslash\mathcal{S}$. Let $\boldsymbol{x}_1 \in \mathcal{S}_j$ and $\boldsymbol{x}_2 \in \mathcal{S}'_{j'}$ for some bags $j$ and $j'$. Upperbound (80) depends only on bags $j$ and $j'$. For any $\boldsymbol{x} \in (\mathcal{S}_j \cup \mathcal{S}_{j'})\backslash\{\boldsymbol{x}_1, \boldsymbol{x}_2\}$, eqs. (2) and (3) bring:

$$F_\phi(\hat{\pi}(\boldsymbol{x}), g(\boldsymbol{x})) - F_\phi(\hat{\pi}'(\boldsymbol{x}), g(\boldsymbol{x})) \;\leq\; \frac{|F_\phi(g(\boldsymbol{x})) - F_\phi(-g(\boldsymbol{x}))|}{m(\boldsymbol{x})}$$

$$= \frac{|g(\boldsymbol{x})|}{b_\phi m(\boldsymbol{x})} \tag{81}$$

$$\leq \frac{h_*}{b_\phi m(\boldsymbol{x})} \;, \tag{82}$$

where $m(\boldsymbol{x})$ is the size of the bag to which it belongs in $\mathcal{S}$, plus 1 iff it is bag $j'$ and $j' \neq j$, minus 1 iff it is bag $j$ and $j' \neq j$. Furthermore, (2) and (3) also bring:

$$F_\phi(\hat{\pi}(\boldsymbol{x}), g(\boldsymbol{x})) = F_\phi(|g(\boldsymbol{x})|) + \frac{1}{b_\phi}((1 - \hat{\pi}(\boldsymbol{x}))1_{g(\boldsymbol{x})>0} + \hat{\pi}(\boldsymbol{x})(1 - 1_{g(\boldsymbol{x})>0}))|g(\boldsymbol{x})|$$

$$\leq F_\phi(0) + \frac{1}{b_\phi}((1 - \hat{\pi}(\boldsymbol{x}))1_{g(\boldsymbol{x})>0} + \hat{\pi}(\boldsymbol{x})(1 - 1_{g(\boldsymbol{x})>0}))h^*$$

$$\leq F_\phi(0) + \frac{h^*}{b_\phi} \;, \forall \boldsymbol{x} \in \mathcal{S} \;.$$

Also, it comes from its definition that:

$$F_\phi(0) = \frac{1}{b_\phi}(0\phi'^{-1}(0) - \phi(\phi'^{-1}(0)))$$

$$= \frac{-\phi(1/2)}{b_\phi} = 1 \;. \tag{83}$$

We obtain that:

$$|\Lambda(\mathcal{S}) - \Lambda(\mathcal{S}')| \;\leq\; \frac{1}{m}\left(1 + \frac{h^*}{b_\phi} + 1 + \frac{h^*}{b_\phi}\right) + \frac{1}{m}\sum_{\boldsymbol{x}\in(\mathcal{S}_j\cup\mathcal{S}_{j'})\backslash\{\boldsymbol{x}_1,\boldsymbol{x}_2\}} \frac{h_*}{b_\phi m(\boldsymbol{x})}$$

$$\leq \frac{Q_1}{m} \;, \tag{84}$$

where

$$Q_1 \doteq 2\left(\frac{2h_*}{b_\phi} + 1\right) \ .$$ (85)

So the IBDI yields that with probability $\leq \delta/2$ over the sampling of $\mathcal{S}$,

$$\Lambda(\mathcal{S}) \geq \mathbb{E}_{\mathcal{D}_m} \sup_g \{\mathbb{E}_{\mathcal{D}}[F_\phi(yg(\boldsymbol{x}))] - \mathbb{E}_{\mathcal{S}}[F_\phi(\hat{\pi}(\boldsymbol{x}), g(\boldsymbol{x}))]\} + Q_1\sqrt{\frac{1}{2m}\log\frac{2}{\delta}} \ ,$$ (86)

We now upperbound the expectation in (86). Using the convexity of the supremum, we have

$$\mathbb{E}_{\mathcal{D}_m} \sup_g \{\mathbb{E}_{\mathcal{D}}[F_\phi(yg(\boldsymbol{x}))] - \mathbb{E}_{\mathcal{S}}[F_\phi(\hat{\pi}(\boldsymbol{x}), g(\boldsymbol{x}))]\}$$

$$= \mathbb{E}_{\mathcal{D}_m} \sup_g \{\mathbb{E}_{\mathcal{D}'_m}[F_\phi(yg(\boldsymbol{x}))] - \mathbb{E}_{\mathcal{S}}[F_\phi(\hat{\pi}(\boldsymbol{x}), g(\boldsymbol{x}))]\}$$

$$\leq \mathbb{E}_{\mathcal{D}_m,\mathcal{D}'_m} \sup_g \{\mathbb{E}_{\mathcal{S}'}[F_\phi(yg(\boldsymbol{x}))] - \mathbb{E}_{\mathcal{S}}[F_\phi(\hat{\pi}(\boldsymbol{x}), g(\boldsymbol{x}))]\} \ .$$ (87)

Consider any set $\mathcal{S} \sim \mathcal{D}_{2m}$, and let $\mathfrak{I}'^2 \subseteq [2m]$ be a subset of $m$ indices, picked uniformly at random among all $\binom{2m}{m}$ possible choices. For any $\mathfrak{I} \subseteq [2m]$, let $\mathcal{S}(\mathfrak{I})$ denote the subset of examples whose index matches $\mathfrak{I}$, and for any $\boldsymbol{x} \in \mathcal{S}(\mathfrak{I})$, let $\hat{\pi}(\boldsymbol{x}|\mathcal{S}(\mathfrak{I}))$ denote its bag proportion in $\mathcal{S}(\mathfrak{I})$. For any $\mathfrak{I}'^2_l$ indexed by $l \geq 1$ and any $\boldsymbol{x} \in \mathcal{S}$, let:

$$\hat{\pi}^s_{|l}(\boldsymbol{x}) \doteq \begin{cases} \hat{\pi}(\boldsymbol{x}|\mathcal{S}(\mathfrak{I}'^2_l)) & \text{if } \boldsymbol{x} \in \mathcal{S}(\mathfrak{I}'^2_l) \\ \hat{\pi}(\boldsymbol{x}|\mathcal{S}\backslash\mathcal{S}(\mathfrak{I}'^2_l)) & \text{otherwise} \end{cases}$$ (88)

denote the label proportions induced by the split of $\mathcal{S}$ in two subsamples $\mathcal{S}(\mathfrak{I}'^2_l)$ and $\mathcal{S}\backslash\mathcal{S}(\mathfrak{I}'^2_l)$. Let

$$\hat{\pi}^\ell_{|l}(\boldsymbol{x}) \doteq \begin{cases} y & \text{if } \boldsymbol{x} \in \mathcal{S}(\mathfrak{I}'^2_l) \\ \hat{\pi}(\boldsymbol{x}|\mathcal{S}\backslash\mathcal{S}(\mathfrak{I}'^2_l)) & \text{otherwise} \end{cases} \ ,$$ (89)

where $y$ is the true label of $\boldsymbol{x}$. Let $\sigma_l(\boldsymbol{x}) \doteq 2 \times 1_{\boldsymbol{x}\in\mathcal{S}(\mathfrak{I}'^2_l)} - 1$. The Label Proportion Complexity (LPC) $L_{2m}$ quantifies the discrepance between these two estimators. When each bag in $\mathcal{S}$ has label proportion zero or one, each term factoring classifier $h$ in eq. (13) (main file) is zero, so $L_{2m} = 0$.

**Lemma 7** *The following holds true:*

$$\mathbb{E}_{\mathcal{D}_m,\mathcal{D}'_m} \sup_g \{\mathbb{E}_{\mathcal{S}'}[F_\phi(yg(\boldsymbol{x}))] - \mathbb{E}_{\mathcal{S}}[F_\phi(\hat{\pi}(\boldsymbol{x}), g(\boldsymbol{x}))]\}$$

$$\leq 2\mathbb{E}_{\mathcal{D}_m,\Sigma_m} \sup_h \{\mathbb{E}_{\mathcal{S}}[\sigma(\boldsymbol{x})F_\phi(\hat{\pi}(\boldsymbol{x}), h(\boldsymbol{x}))]\} + L_{2m} \ .$$ (90)

**Proof** For any $\boldsymbol{\sigma} \in \Sigma_m$ and any sets $\mathcal{S} = \{\boldsymbol{x}_1, \boldsymbol{x}_2, ..., \boldsymbol{x}_m\}$ and $\mathcal{S}' = \{\boldsymbol{x}'_1, \boldsymbol{x}'_2, ..., \boldsymbol{x}'_m\}$ of size $m$, denote

$$\mathcal{S}_{\boldsymbol{\sigma}} \doteq \{\boldsymbol{x}'_i \text{ iff } \sigma_i = 1, \boldsymbol{x}_i \text{ otherwise}\} \ ,$$
$$\mathcal{S}_{\overline{\boldsymbol{\sigma}}} \doteq \{\boldsymbol{x}'_i \text{ iff } \sigma_i = -1, \boldsymbol{x}_i \text{ otherwise}\} = (\mathcal{S} \cup \mathcal{S}')\backslash\mathcal{S}_{\boldsymbol{\sigma}} \ .$$ (91)

and

$$\hat{\pi}_*(\boldsymbol{x}) \doteq \begin{cases} \hat{\pi}_{\boldsymbol{\sigma}}(\boldsymbol{x}) & \text{if } \boldsymbol{x} \in \mathcal{S}_{\boldsymbol{\sigma}} \ , \\ \hat{\pi}_{\overline{\boldsymbol{\sigma}}}(\boldsymbol{x}) & \text{otherwise} \end{cases} \ ,$$ (92)

where $\hat{\pi}_{\boldsymbol{\sigma}}(.)$ denote the label proportions in $\mathcal{S}_{\boldsymbol{\sigma}}$ and $\hat{\pi}_{\overline{\boldsymbol{\sigma}}}(.)$ denote the label proportions in $\mathcal{S}_{\overline{\boldsymbol{\sigma}}}$. Let $\hat{\pi}(.)$ denote the label proportions in $\mathcal{S}$, $\hat{\pi}'(.)$ denote the label proportions in $\mathcal{S}'$ (we know each bag to which each example in $\mathcal{S}'$ belongs to, so we can compute these estimators), We have

$$\mathbb{E}_{\mathcal{D}_m,\mathcal{D}'_m} \sup_h \{\mathbb{E}_{\mathcal{S}'}[F_\phi(yh(\boldsymbol{x}))] - \mathbb{E}_{\mathcal{S}}[F_\phi(\hat{\pi}(\boldsymbol{x}), h(\boldsymbol{x}))]\}$$

$$= \mathbb{E}_{\mathcal{D}_m,\mathcal{D}'_m} \sup_h \left\{\mathbb{E}_{\mathcal{S}'}[F_\phi(\hat{\pi}'(\boldsymbol{x}), h(\boldsymbol{x}))] - \mathbb{E}_{\mathcal{S}}[F_\phi(\hat{\pi}(\boldsymbol{x}), h(\boldsymbol{x}))] - \frac{1}{b_\phi} \times \Delta_1\right\}$$

$$= \mathbb{E}_{\mathcal{D}_m,\mathcal{D}'_m} \sup_h \left\{\mathbb{E}_{\mathcal{S}_{\boldsymbol{\sigma}}}[\sigma(\boldsymbol{x})F_\phi(\hat{\pi}^l(\boldsymbol{x}), h(\boldsymbol{x}))] - \mathbb{E}_{\mathcal{S}_{\overline{\boldsymbol{\sigma}}}}[\sigma(\boldsymbol{x})F_\phi(\hat{\pi}^r(\boldsymbol{x}), h(\boldsymbol{x}))] - \frac{1}{b_\phi} \times \Delta_1\right\}$$ (93)

with

$$\Delta_1 \;\doteq\; \mathbb{E}_{\mathcal{S}'}[((1-\hat{\pi}'(\boldsymbol{x}))1_{y=1} - \hat{\pi}'(\boldsymbol{x})1_{y=-1})h(\boldsymbol{x})] \; ; \tag{94}$$

$$\hat{\pi}^l(\boldsymbol{x}) \;\doteq\; \frac{1}{2}\left((1+\sigma(\boldsymbol{x}))\hat{\pi}'(\boldsymbol{x}) + (1-\sigma(\boldsymbol{x}))\hat{\pi}(\boldsymbol{x})\right) \;,$$

$$\hat{\pi}^r(\boldsymbol{x}) \;\doteq\; \frac{1}{2}\left((1+\sigma(\boldsymbol{x}))\hat{\pi}(\boldsymbol{x}) + (1-\sigma(\boldsymbol{x}))\hat{\pi}'(\boldsymbol{x})\right) \;. \tag{95}$$

We also have from eq. (2) and (3):

$$\mathbb{E}_{\mathcal{S}_{\boldsymbol{\sigma}}}[\sigma(\boldsymbol{x})F_\phi(\hat{\pi}^l(\boldsymbol{x}), h(\boldsymbol{x}))] \;=\; \mathbb{E}_{\mathcal{S}_{\boldsymbol{\sigma}}}[\sigma(\boldsymbol{x})F_\phi(\hat{\pi}_{\boldsymbol{\sigma}}(\boldsymbol{x}), h(\boldsymbol{x}))] - \frac{1}{b_\phi} \times \Delta_2 \;, \tag{96}$$

$$\mathbb{E}_{\mathcal{S}_{\overline{\boldsymbol{\sigma}}}}[\sigma(\boldsymbol{x})F_\phi(\hat{\pi}^r(\boldsymbol{x}), h(\boldsymbol{x}))] \;=\; \mathbb{E}_{\mathcal{S}_{\overline{\boldsymbol{\sigma}}}}[\sigma(\boldsymbol{x})F_\phi(\hat{\pi}_{\overline{\boldsymbol{\sigma}}}(\boldsymbol{x}), h(\boldsymbol{x}))] - \frac{1}{b_\phi} \times \Delta_3 \;, \tag{97}$$

with

$$\Delta_2 \;\doteq\; \mathbb{E}_{\mathcal{S}_{\boldsymbol{\sigma}}}[\sigma(\boldsymbol{x})(\hat{\pi}^l(\boldsymbol{x}) - \hat{\pi}_{\boldsymbol{\sigma}}(\boldsymbol{x}))h(\boldsymbol{x})] \;, \tag{98}$$

$$\Delta_3 \;\doteq\; \mathbb{E}_{\mathcal{S}_{\overline{\boldsymbol{\sigma}}}}[\sigma(\boldsymbol{x})(\hat{\pi}^r(\boldsymbol{x}) - \hat{\pi}_{\overline{\boldsymbol{\sigma}}}(\boldsymbol{x}))h(\boldsymbol{x})] \;. \tag{99}$$

We also have:

$$\Delta_3 - \Delta_2 - \Delta_1 \;=\; \mathbb{E}_{\mathcal{S}'}[(\hat{\pi}_*(\boldsymbol{x}) - 1_{y=1})h(\boldsymbol{x})] + \mathbb{E}_{\mathcal{S}}[(\hat{\pi}(\boldsymbol{x}) - \hat{\pi}_*(\boldsymbol{x}))h(\boldsymbol{x})]$$
$$\doteq\; \Delta_4 \;. \tag{100}$$

Putting eqs (93), (96), (97) and (100) altogether, we get, after introducing Rademacher variables:

$$\mathbb{E}_{\mathcal{D}_m, \mathcal{D}'_m, \Sigma_m} \sup_h \{\mathbb{E}_{\mathcal{S}'}[F_\phi(yh(\boldsymbol{x}))] - \mathbb{E}_{\mathcal{S}}[F_\phi(\hat{\pi}(\boldsymbol{x}), h(\boldsymbol{x}))]\}$$

$$= \; \mathbb{E}_{\mathcal{D}_m, \mathcal{D}'_m, \Sigma_m} \sup_h \{\mathbb{E}_{\mathcal{S}_{\boldsymbol{\sigma}}}[\sigma(\boldsymbol{x})F_\phi(\hat{\pi}_{\boldsymbol{\sigma}}(\boldsymbol{x}), h(\boldsymbol{x}))] - \mathbb{E}_{\mathcal{S}_{\overline{\boldsymbol{\sigma}}}}[\sigma(\boldsymbol{x})F_\phi(\hat{\pi}_{\overline{\boldsymbol{\sigma}}}(\boldsymbol{x}), h(\boldsymbol{x}))] + \Delta_4\}$$

$$\leq \; \mathbb{E}_{\mathcal{D}_m, \mathcal{D}'_m, \Sigma_m} \sup_h \{\mathbb{E}_{\mathcal{S}_{\boldsymbol{\sigma}}}[\sigma(\boldsymbol{x})F_\phi(\hat{\pi}_{\boldsymbol{\sigma}}(\boldsymbol{x}), h(\boldsymbol{x}))] - \mathbb{E}_{\mathcal{S}_{\overline{\boldsymbol{\sigma}}}}[\sigma(\boldsymbol{x})F_\phi(\hat{\pi}_{\overline{\boldsymbol{\sigma}}}(\boldsymbol{x}), h(\boldsymbol{x}))]\}$$

$$+\mathbb{E}_{\mathcal{D}_m, \mathcal{D}'_m, \Sigma_m} \sup_h \{\mathbb{E}_{\mathcal{S}'}[(\hat{\pi}_*(\boldsymbol{x}) - 1_{y=1})h(\boldsymbol{x})] + \mathbb{E}_{\mathcal{S}}[(\hat{\pi}(\boldsymbol{x}) - \hat{\pi}_*(\boldsymbol{x}))h(\boldsymbol{x})]\}$$

$$= \; \mathbb{E}_{\mathcal{D}_m, \mathcal{D}'_m, \Sigma_m} \sup_h \{\mathbb{E}_{\mathcal{S}'}[\sigma(\boldsymbol{x})F_\phi(\hat{\pi}'(\boldsymbol{x}), h(\boldsymbol{x}))] - \mathbb{E}_{\mathcal{S}}[\sigma(\boldsymbol{x})F_\phi(\hat{\pi}(\boldsymbol{x}), h(\boldsymbol{x}))]\}$$

$$+\mathbb{E}_{\mathcal{D}_m, \mathcal{D}'_m, \Sigma_m} \sup_h \{\mathbb{E}_{\mathcal{S}'}[(\hat{\pi}_*(\boldsymbol{x}) - 1_{y=1})h(\boldsymbol{x})] + \mathbb{E}_{\mathcal{S}}[(\hat{\pi}(\boldsymbol{x}) - \hat{\pi}_*(\boldsymbol{x}))h(\boldsymbol{x})]\} \tag{101}$$

$$\leq \; 2\mathbb{E}_{\mathcal{D}_m, \Sigma_m} \sup_h \{\mathbb{E}_{\mathcal{S}}[\sigma(\boldsymbol{x})F_\phi(\hat{\pi}(\boldsymbol{x}), h(\boldsymbol{x}))]\}$$

$$+\mathbb{E}_{\mathcal{D}_m, \mathcal{D}'_m, \Sigma_m} \sup_h \{\mathbb{E}_{\mathcal{S}'}[(\hat{\pi}_*(\boldsymbol{x}) - 1_{y=1})h(\boldsymbol{x})] + \mathbb{E}_{\mathcal{S}}[(\hat{\pi}(\boldsymbol{x}) - \hat{\pi}_*(\boldsymbol{x}))h(\boldsymbol{x})]\} \;. \tag{102}$$

Eq. (101) holds because the distribution of the supremum is the same. We also have:

$$\mathbb{E}_{\mathcal{D}_m, \mathcal{D}'_m, \Sigma_m} \sup_h \{\mathbb{E}_{\mathcal{S}'}[(\hat{\pi}_*(\boldsymbol{x}) - 1_{y=1})h(\boldsymbol{x})] + \mathbb{E}_{\mathcal{S}}[(\hat{\pi}(\boldsymbol{x}) - \hat{\pi}_*(\boldsymbol{x}))h(\boldsymbol{x})]\}$$

$$= \; \mathbb{E}_{\mathcal{D}_m, \mathcal{D}'_m, \Sigma_m} \sup_h \{\mathbb{E}_{\mathcal{S}}[(\hat{\pi}(\boldsymbol{x}) - \hat{\pi}_*(\boldsymbol{x}))h(\boldsymbol{x})] - \mathbb{E}_{\mathcal{S}'}[(1_{y=1} - \hat{\pi}_*(\boldsymbol{x}))h(\boldsymbol{x})]\}$$

$$= \; \mathbb{E}_{\mathcal{D}_{2m}} \mathbb{E}_{\mathcal{J}_1^{/2}, \mathcal{J}_2^{/2}} \sup_h \mathbb{E}_{\mathcal{S}}[\sigma_1(\boldsymbol{x})(\hat{\pi}_{|2}^s(\boldsymbol{x}) - \hat{\pi}_{|1}^\ell(\boldsymbol{x}))h(\boldsymbol{x})] \tag{103}$$

$$= \; L_{2m} \;. \tag{104}$$

Eq. (103) holds because swapping the sample does not make any difference in the outer expectation, as each couple of swapped samples is generated with the same probability without swapping. Putting altogether (102) and (104) ends the proof of Lemma 7. ∎

We now bound the deviations of $\mathbb{E}_{\Sigma_m} \sup_h \{\mathbb{E}_{\mathcal{S}}[\sigma(\boldsymbol{x})F_\phi(\hat{\pi}(\boldsymbol{x}), h(\boldsymbol{x}))]\}$ with respect to its expectation over the sampling of $\mathcal{S}$, $\mathbb{E}_{\mathcal{D}_m, \Sigma_m} \sup_h \{\mathbb{E}_{\mathcal{S}}[\sigma(\boldsymbol{x})F_\phi(\hat{\pi}(\boldsymbol{x}), h(\boldsymbol{x}))]\}$. To do that, we use a third time the IBDI and compute an upperbound for

$$\left| \begin{array}{l} \mathbb{E}_{\Sigma_m} \sup_g \{\mathbb{E}_{\mathcal{S}_1}[\sigma(\boldsymbol{x})F_\phi(\hat{\pi}(\boldsymbol{x}), h(\boldsymbol{x}))]\} \\ -\mathbb{E}_{\Sigma_m} \sup_g \{\mathbb{E}_{\mathcal{S}_2}[\sigma(\boldsymbol{x})F_\phi(\hat{\pi}(\boldsymbol{x}), h(\boldsymbol{x}))]\} \end{array} \right|$$

$$\leq \; \mathbb{E}_{\Sigma_m} \left[ \left| \begin{array}{l} \sup_g \{\mathbb{E}_{\mathcal{S}_1}[\sigma(\boldsymbol{x})F_\phi(\hat{\pi}(\boldsymbol{x}), h(\boldsymbol{x}))]\} \\ -\sup_g \{\mathbb{E}_{\mathcal{S}_2}[\sigma(\boldsymbol{x})F_\phi(\hat{\pi}(\boldsymbol{x}), h(\boldsymbol{x}))]\} \end{array} \right| \right] \tag{105}$$

$$\leq \; \max_{\Sigma_m} \left[ \left| \begin{array}{l} \sup_g \{\mathbb{E}_{\mathcal{S}_1}[\sigma(\boldsymbol{x})F_\phi(\hat{\pi}(\boldsymbol{x}), h(\boldsymbol{x}))]\} \\ -\sup_g \{\mathbb{E}_{\mathcal{S}_2}[\sigma(\boldsymbol{x})F_\phi(\hat{\pi}(\boldsymbol{x}), h(\boldsymbol{x}))]\} \end{array} \right| \right] \leq \frac{Q_1}{m} \;, \tag{106}$$

where $Q_1$ is defined in eq. (85). Eq. (105) holds because of the triangular inequality. Ineq. (106) holds because $|\sigma(.)| = 1$. So with probability $\leq \delta/2$ over the sampling of $\mathcal{S}$,

$$\mathbb{E}_{\Sigma_m} \sup_h \{\mathbb{E}_\mathcal{S}[\sigma(\boldsymbol{x})F_\phi(\hat{\pi}(\boldsymbol{x}), h(\boldsymbol{x}))]\}$$

$$\leq \quad \mathbb{E}_{\mathcal{D}_m, \Sigma_m} \sup_h \{\mathbb{E}_\mathcal{S}[\sigma(\boldsymbol{x})F_\phi(\hat{\pi}(\boldsymbol{x}), h(\boldsymbol{x}))]\} - Q_1 \sqrt{\frac{1}{2m} \log \frac{2}{\delta}} \ , \qquad (107)$$

where $Q_1$ is defined via (84). We obtain that with probability $> 1 - ((\delta/2) + (\delta/2)) = 1 - \delta$, the following holds $\forall h$:

$$
\begin{aligned}
\mathbb{E}_\mathcal{D}[F_\phi(yh(\boldsymbol{x}))] \quad &\leq \quad \mathbb{E}_\mathcal{S}[F_\phi(\hat{\pi}(\boldsymbol{x}), h(\boldsymbol{x}))] + \Lambda(\mathcal{S}) \ \text{ (see (78) and (79))} \\
&\leq \quad \mathbb{E}_\mathcal{S}[F_\phi(\hat{\pi}(\boldsymbol{x}), h(\boldsymbol{x}))] + \mathbb{E}_{\mathcal{D}_m} \sup_g \{\mathbb{E}_\mathcal{D}[F_\phi(yg(\boldsymbol{x}))] - \mathbb{E}_\mathcal{S}[F_\phi(\hat{\pi}(\boldsymbol{x}), g(\boldsymbol{x}))]\} \\
&\qquad + Q_1 \sqrt{\frac{1}{2m} \log \frac{2}{\delta}} \ \text{ (from (86))} \\
&\leq \quad \mathbb{E}_\mathcal{S}[F_\phi(\hat{\pi}(\boldsymbol{x}), h(\boldsymbol{x}))] + \mathbb{E}_{\mathcal{D}_m, \mathcal{D}'_m} \sup_g \{\mathbb{E}_{\mathcal{S}'}[F_\phi(yg(\boldsymbol{x}))] - \mathbb{E}_\mathcal{S}[F_\phi(\hat{\pi}(\boldsymbol{x}), g(\boldsymbol{x}))]\} \\
&\qquad + Q_1 \sqrt{\frac{1}{2m} \log \frac{2}{\delta}} \ \text{ (from (87))} \\
&\leq \quad \mathbb{E}_\mathcal{S}[F_\phi(\hat{\pi}(\boldsymbol{x}), h(\boldsymbol{x}))] + 2\mathbb{E}_{\mathcal{D}_m, \Sigma_m} \sup_g \{\mathbb{E}_\mathcal{S}[\sigma(\boldsymbol{x})F_\phi(\hat{\pi}(\boldsymbol{x}), g(\boldsymbol{x}))]\} + L_{2m} \\
&\qquad + Q_1 \sqrt{\frac{1}{2m} \log \frac{2}{\delta}} \ \text{ (Lemma (7))} \\
&\leq \quad \mathbb{E}_\mathcal{S}[F_\phi(\hat{\pi}(\boldsymbol{x}), h(\boldsymbol{x}))] + 2\mathbb{E}_{\Sigma_m} \sup_h \{\mathbb{E}_\mathcal{S}[\sigma(\boldsymbol{x})F_\phi(\hat{\pi}(\boldsymbol{x}), h(\boldsymbol{x}))]\} + L_{2m} \\
&\qquad + 2Q_1 \sqrt{\frac{1}{2m} \log \frac{2}{\delta}} \ \text{ (from (107))} \\
&= \quad \mathbb{E}_{\Sigma_{\hat{\pi}}} \mathbb{E}_\mathcal{S}[F_\phi(\sigma(\boldsymbol{x})h(\boldsymbol{x}))] + 2\hat{R}_m^b + L_{2m} + 4\left(\frac{2h_*}{b_\phi} + 1\right) \sqrt{\frac{1}{2m} \log \frac{2}{\delta}} \ ,
\end{aligned}
$$

as claimed.

### 2.9.2 Proof of eq. (15)

We have $F'_\phi(x) = -(1/b_\phi))(\phi^\star)'(-x) = -(1/b_\phi)(\phi')^{-1}(-x) \in [-1/b_\phi, 0]$, and thus $F_\phi$ is $1/b_\phi$-Lipschitz, so Theorem 4.12 in [7] brings:

$$
\begin{aligned}
R_m^b(F, \eta) \quad &= \quad \mathbb{E}_{\boldsymbol{\sigma} \sim \Sigma_m} \sup_{h \in \mathcal{H}} \{\mathbb{E}_{i \sim [m]}[\sigma_i \mathbb{E}_{\boldsymbol{\sigma}' \sim \Sigma_{\hat{\pi}}}[F_\phi(\sigma'_i h(\boldsymbol{x}_i) - \eta)]]\} \\
&\leq \quad b_\phi \mathbb{E}_{\boldsymbol{\sigma} \sim \Sigma_m} \sup_{h \in \mathcal{H}} \{\mathbb{E}_{i \sim [m]}[\sigma_i \mathbb{E}_{\boldsymbol{\sigma}' \sim \Sigma_{\hat{\pi}}}[\sigma'_i h(\boldsymbol{x}_i) - \eta]]\} \\
&= \quad b_\phi \mathbb{E}_{\boldsymbol{\sigma} \sim \Sigma_m} \sup_{h \in \mathcal{H}} \{\mathbb{E}_{i \sim [m]}[\sigma_i \mathbb{E}_{\boldsymbol{\sigma}' \sim \Sigma_{\hat{\pi}}}[\sigma'_i h(\boldsymbol{x}_i)]]\} \\
&= \quad b_\phi \mathbb{E}_{\boldsymbol{\sigma} \sim \Sigma_m} \sup_{h \in \mathcal{H}} \{\mathbb{E}_{i \sim [m]}[\sigma_i (2\hat{\pi}(\boldsymbol{x}_i) - 1)h(\boldsymbol{x}_i)]\} \ ,
\end{aligned}
$$

as claimed.

## 3 Supplementary Material on Experiments

### 3.1 Full Experimental Setup

All mean operator algorithms have been coded in R. For $\propto$SVM and InvCal, we used a Matlab[1] implementation from the authors of [8]. The range of parameters for cross validation are $\lambda = \lambda' m$ with $\lambda' \in \{0\} \cup 10^{\{0,1,2\}}$, $\gamma \in 10^{-\{2,1,0\}}$, $\sigma \in 2^{-\{2,1,0\}}$ for mean operator algorithms. We run all

experiments with $D_w = I$ and $\varepsilon = 0$. Since we test on similar domains (6 are actually the same), ranges for InvCal and $\propto$SVM are taken from [8]. To avoid an additional source of complexity in the analysis, we cross-validate all hyper-parameters using the knowledge of all labels of the validation sets; notice that labels at validation time generally would not be accessible in a real world application.

## 3.2 Simulated Domain for Violation of Homogeneity Assumption

The synthetic data generated for this test consists on 16 classification problems, each one formed by 16 bags of 100 two-dimensional normal samples. The distribution generating the first dataset satisfies the homogeneity assumption (Figure 1 (a)). Then, we gradually change the position of the class-conditional bag-conditional means on one linear direction (to the right on Figure 1 (b) and (c)), with different offsets for different bags. In Figure 1 we give a graphical explanation of the process on 3 bags.

Figure 1: Violation of homogeneity assumption

## 3.3 Simulated Domain from [8]

The MM algorithm was shown to learn a model with zero accuracy prediction on the toy domain of [8]. We report here in Table 1 performance of all mean operator algorithms measured in transductive setting, training with cross-validation. Although none of the distances used in our experiments in LMM leads reasonable accuracy in the toy dataset, AMM$^{max}$ initialised with *any* starting point learns *in one step* a model which perfectly classifies all the instances. We also notice that EMM returns an optimal classifier by itself (not reported in Table 1).

Table 1: AUC on the toy dataset of [8]

|                | AMM$^{min}$ | AMM$^{max}$ |
|----------------|-------------|-------------|
| EMM            | 100.00      | 100.00      |
| MM             | 8.46        | 100.00      |
| LMM$_G$        | 8.46        | 100.00      |
| LMM$_{G,s}$    | 8.46        | 100.00      |
| LMM$_{nc}$     | 8.46        | 100.00      |
| 1              | 8.46        | 100.00      |
| 10ran          | 100.00      | 100.00      |

## 3.4 Additional Tests on alter-$\propto$SVM [8]

In our experiments, we observe that AUCs achieved by $\propto$SVM can be very high, but it is also often *below* 0.5; in those cases the algorithm outputs models which are worse than random and the average performance over the 5 test splits drops. We are able to reproduce the same behaviour on the *heart*

dataset provided by the authors in a demo for alter-$\propto$SVM; this also proves our bag assignment for LLP simulation does not introduce the issue. In a first test, we randomly select 3/4 of the dataset, and randomly assign instances to 4 bags of fixed size 64, following [8]. We repeat the training split 50 times with $C = C_p = 1$, as in the demo, and we measure AUCs on the same training set. As expected, a consistent number of run (22%) ends up producing AUC smaller than 0.5. We display in Figure 2 (a) the AUC's density profile, which shows a relevant mass around 0.25; notice also the two distribution modes look symmetric around 0.5.

In a second test, we investigate further measuring pairs of training set AUC and losses obtained by the same execution of the algorithm. In this case, we run over all parameters ranges defined in $\propto$SVM's paper, and do not pick the model that minimizes the loss over the 10 random runs, but record losses of all. Figures 2 (b) and (c) show scatter plots relative to two chosen training set splits. We observe that loss minimization can lead both to high and low AUCs, with few points close to 0.5. A possible explanation might be in the inverted polarity of the learnt linear classifier; inverted polarity in this contest means having a model which would achieve better performance classifying instances labels opposite to the ones predicted. We conclude that optimizing $\propto$SVM's loss in some cases might be equivalent to train a max-margin separator of the unlabelled data, exploiting only weakly the information given by the label proportions. This would give a heuristic understanding of the frequent symmetrical behaviour of the AUC.

Figure 2: alter-$\propto$SVM: empirical distribution of AUC (a), and relationship between loss and AUC in two different train spit (b)(c)

### 3.5  Scalability

Figure 8 (a) shows runtime of learning (including cross-validation) of MM and LMM with regard to the number of bags – which is the natural parameter of time complexity for our Laplacian-based methods. Although the 3 layers of cross-validation of LMM$_{G,s}$, LMM$_{nc}$ results the only method clearly not scalable. Figure 8 (b) presents how our one-shots algorithms scale on all small domains as a function of problem size. Runtime is averaged over the different bag assignments. The same plot is given in Figure 8 (c) for iterative algorithms, in particular AMM$^{min}$ and (alter/conv)-$\propto$SVM. All curves are completed with measurements on bigger domains when available. Runtime of SVMs is not directly comparable with our methods. This is due to both (a) the implementation on different programming languages and (b) to the fact that the code provided implements kernel SVM, even for linear kernels, which is a big overhead in computation and memory access. Nevertheless, the high growth rate of conv-$\propto$SVM makes the algorithm not suitable for large datasets. Noticeably, even if alter-$\propto$SVM does not show such behaviour, we are not able to run it on our bigger domains, since it requires approximately 10 hours to run on a training set split with fixed parameters.

### 3.6  Full Results on Small Domains

Finally we report details about all experiments run on the 10 small domains (Table 2). In the following Tables, columns show the number of bags generated through K-MEANS. Each cell contains

Figure 3: Learning runtime of LMM for bags number (a), and for domain size one-shot (b) and iterative methods (c)

Table 2: Small domains size

| dataset | instances | feature |
|---|---|---|
| *arrhythmia* | 452 | 297 |
| *australian* | 690 | 39 |
| *breastw* | 699 | 11 |
| *colic* | 368 | 83 |
| *german* | 1000 | 27 |
| *heart* | 270 | 14 |
| *ionosphere* | 351 | 37 |
| *vertebral column* | 620 | 9 |
| *vote* | 435 | 49 |
| *wine* | 178 | 16 |

average AUC over 5 test splits and standard deviation; runtime in second is in the separated column. Best performing algorithm and ones not worse than 0.1 AUC are bold faced. Comparisons are made in the respective top/bottom sub-tables, which group one-shot and iterative algorithms. We use ↑ to highlight runs which achieve AUCs greater or equal than the Oracles.

## Table 3: *arrhythmia*

| algorithm | 2 bags AUC | time(s) | 4 bags AUC | time(s) | 8 bags AUC | time(s) | 16 bags AUC | time(s) | 32 bags AUC | time(s) |
|---|---|---|---|---|---|---|---|---|---|---|
| EMM | **70.91 ± 6.81** | 2 | 50.55 ± 7.54 | 2 | 50.31 ± 7.55 | 2 | 47.03 ± 6.60 | 2 | 52.34 ± 7.25 | 2 |
| MM | 64.99 ± 2.99 | 2 | 60.48 ± 7.28 | 1 | 68.17 ± 5.95 | 2 | 70.01 ± 9.33 | 2 | 72.85 ± 9.49 | 2 |
| $LMM_G$ | 64.99 ± 2.99 | 18 | 68.10 ± 4.43 | 17 | 71.53 ± 2.36 | 20 | 72.06 ± 7.62 | 18 | 76.29 ± 7.91 | 20 |
| $LMM_{G,s}$ | 64.99 ± 2.99 | 49 | **68.34 ± 3.95** | 49 | 71.53 ± 2.36 | 54 | 72.06 ± 7.62 | 52 | 76.29 ± 7.91 | 57 |
| $LMM_{nc}$ | 64.99 ± 2.99 | 83 | 61.19 ± 7.53 | 83 | 70.21 ± 5.17 | 119 | 70.89 ± 9.86 | 267 | 73.82 ± 9.29 | 854 |
| InvCal | 64.75 ± 3.04 | 17 | 66.12 ± 260 | 17 | 60.87 ± 3.54 | 17 | 44.46 ± 3.36 | 17 | 56.36 ± 5.26 | 17 |
| $AMM^{min}$ $AMM_{EMM}$ | 59.54 ± 7.52 | 9 | 52.65 ± 3.10 | 8 | 63.46 ± 10.37 | 8 | 67.85 ± 9.56 | 8 | 75.65 ± 8.81 | 8 |
| $AMM_{MM}$ | 57.29 ± 5.95 | 7 | 60.00 ± 7.96 | 4 | 70.12 ± 6.46 | 4 | 73.66 ± 8.86 | 5 | 78.36 ± 8.53 | 5 |
| $AMM_G$ | 58.15 ± 6.83 | 31 | 68.80 ± 2.15 | 28 | **73.08 ± 2.92** | 30 | 74.54 ± 7.98 | 29 | **80.32 ± 8.08** | 30 |
| $AMM_{G,s}$ | 56.67 ± 4.66 | 92 | 69.83 ± 2.69 | 84 | **73.08 ± 2.92** | 88 | 73.34 ± 7.62 | 88 | **80.32 ± 8.08** | 91 |
| $AMM_{nc}$ | 57.29 ± 5.95 | 97 | 59.71 ± 8.39 | 90 | 71.43 ± 6.21 | 126 | 73.49 ± 8.95 | 274 | 78.04 ± 8.26 | 862 |
| $AMM_1$ | **65.80 ± 6.92** | 5 | **70.00 ± 5.89** | 4 | 68.17 ± 7.19 | 4 | 69.93 ± 4.27 | 4 | 72.31 ± 5.02 | 5 |
| $AMM_{10ran}$ | 54.09 ± 12.03 | 30 | 55.78 ± 17.36 | 32 | 66.38 ± 7.32 | 51 | 66.89 ± 6.75 | 51 | 73.61 ± 5.15 | 57 |
| $AMM^{max}$ $AMM_{EMM}$ | 50.59 ± 5.97 | 41 | 59.32 ± 5.82 | 41 | 60.85 ± 5.43 | 37 | 60.38 ± 4.08 | 41 | 58.31 ± 8.40 | 40 |
| $AMM_{MM}$ | 62.08 ± 9.46 | 45 | 46.86 ± 3.90 | 34 | 67.28 ± 8.92 | 33 | 74.04 ± 9.46 | 35 | 71.00 ± 7.65 | 38 |
| $AMM_G$ | 62.08 ± 9.46 | 141 | 62.27 ± 8.14 | 128 | 65.78 ± 3.92 | 118 | 64.64 ± 10.26 | 121 | 73.07 ± 6.72 | 124 |
| $AMM_{G,s}$ | 62.08 ± 9.46 | 414 | 63.13 ± 5.17 | 380 | 63.85 ± 7.00 | 346 | 65.49 ± 10.62 | 354 | 73.05 ± 6.70 | 374 |
| $AMM_{nc}$ | 62.08 ± 9.46 | 206 | 55.57 ± 6.07 | 182 | 64.30 ± 6.24 | 207 | **76.33 ± 3.96** | 362 | 70.82 ± 4.23 | 965 |
| $AMM_1$ | 60.53 ± 9.79 | 31 | 54.14 ± 13.28 | 34 | 67.45 ± 3.91 | 32 | 55.85 ± 8.96 | 35 | 61.26 ± 6.95 | 38 |
| $AMM_{10ran}$ | 49.79 ± 8.14 | 307 | 55.37 ± 14.62 | 370 | 53.78 ± 5.13 | 301 | 60.62 ± 8.04 | 322 | 64.20 ± 2.84 | 338 |
| SVM alter-∝ | 49.24 ± 3.92 | 96 | 57.10 ± 2.71 | 100 | 56.38 ± 2.73 | 104 | 35.31 ± 1.30 | 114 | 38.68 ± 6.10 | 125 |
| conv-∝ | 54.15 ± 2.22 | 2054 | 34.82 ± 3.20 | 2078 | 38.31 ± 8.24 | 2168 | 61.96 ± 1.10 | 1930 | 48.77 ± 5.73 | 2004 |
| Oracle | 99.99 ± 0.02 | 2 | 99.98 ± 0.05 | 2 | 99.94 ± 0.13 | 2 | 100.00 ± 0.00 | 2 | 99.97 ± 0.07 | 2 |

## Table 4: *australian*

| algorithm | 2 bags AUC | time(s) | 4 bags AUC | time(s) | 8 bags AUC | time(s) | 16 bags AUC | time(s) | 32 bags AUC | time(s) |
|---|---|---|---|---|---|---|---|---|---|---|
| EMM | 66.48 ± 3.16 | <1 | 64.67 ± 4.22 | <1 | 63.56 ± 4.00 | <1 | 64.17 ± 4.80 | <1 | 63.14 ± 5.41 | <1 |
| MM | **81.08 ± 1.66** | <1 | 87.11 ± 2.68 | <1 | 87.49 ± 2.86 | 1 | 87.36 ± 2.22 | <1 | 89.53 ± 2.13 | 2 |
| $LMM_G$ | **81.08 ± 1.66** | 4 | 87.09 ± 2.82 | 4 | **87.81 ± 3.16** | 5 | 88.46 ± 2.50 | 6 | 89.69 ± 2.68 | 8 |
| $LMM_{G,s}$ | **81.08 ± 1.66** | 14 | **87.81 ± 3.08** | 15 | 87.88 ± 3.21 | 19 | **89.18 ± 2.05** | 20 | **90.80 ± 2.53** | 27 |
| $LMM_{nc}$ | **81.08 ± 1.66** | 57 | 87.02 ± 2.72 | 49 | 87.46 ± 3.03 | 57 | 88.06 ± 2.31 | 90 | 89.41 ± 2.41 | 217 |
| Invcal | 19.67 ± 2.23 | 5 | 59.50 ± 5.86 | 5 | 60.83 ± 3.17 | 5 | 60.83 ± 5.27 | 5 | 51.81 ± 4.72 | 5 |
| $AMM^{min}$ $AMM_{EMM}$ | 86.65 ± 2.06 | 4 | **86.59 ± 3.08** | 4 | 86.50 ± 4.11 | 4 | 89.51 ± 2.48 | 6 | 88.85 ± 4 | 6 |
| $AMM_{MM}$ | **87.54 ± 3.84** | 3 | 84.35 ± 3.63 | 4 | **86.99 ± 3.87** | 4 | 89.43 ± 1.34 | 4 | 89.55 ± 3.18 | 5 |
| $AMM_G$ | **87.54 ± 3.84** | 10 | 84.79 ± 3.17 | 14 | 86.78 ± 4.21 | 14 | 89.52 ± 2.18 | 14 | 89.88 ± 2.78 | 18 |
| $AMM_{G,s}$ | **87.54 ± 3.84** | 30 | 85.12 ± 3.75 | 39 | 86.75 ± 4.19 | 43 | **90.37 ± 1.67** | 43 | 89.95 ± 2.80 | 54 |
| $AMM_{nc}$ | **87.54 ± 3.84** | 63 | 85.10 ± 3.55 | 57 | 86.63 ± 4.02 | 66 | 89.00 ± 1.83 | 97 | **90.11 ± 2.93** | 227 |
| $AMM_1$ | 72.60 ± 5.70 | 2 | 85.04 ± 2.53 | 3 | **86.89 ± 3.73** | 4 | 88.91 ± 2.32 | 4 | 88.98 ± 3.00 | 4 |
| $AMM_{10ran}$ | 79.21 ± 5.07 | 27 | 80.97 ± 2.27 | 31 | 85.08 ± 3.30 | 34 | 89.19 ± 1.81 | 46 | 87.70 ± 2.68 | 47 |
| $AMM^{max}$ $AMM_{EMM}$ | 80.09 ± 3.99 | 17 | 71.46 ± 1.85 | 16 | 73.41 ± 6.07 | 16 | 73.25 ± 3.33 | 18 | 81.73 ± 3.60 | 19 |
| $AMM_{MM}$ | 86.83 ± 4.26 | 20 | 72.96 ± 4.26 | 15 | 70.25 ± 4.65 | 16 | 73.89 ± 5.77 | 18 | 75.91 ± 3.50 | 21 |
| $AMM_G$ | 86.83 ± 4.26 | 61 | 73.32 ± 1.95 | 48 | 71.16 ± 4.94 | 51 | 73.57 ± 6.86 | 55 | 75.25 ± 3.18 | 63 |
| $AMM_{G,s}$ | 86.83 ± 4.26 | 181 | 73.25 ± 2.03 | 143 | 71.19 ± 4.91 | 153 | 74.77 ± 6.85 | 163 | 75.25 ± 3.18 | 188 |
| $AMM_{nc}$ | 86.83 ± 4.26 | 114 | 73.74 ± 2.48 | 92 | 70.36 ± 5.16 | 102 | 75.16 ± 5.71 | 138 | 76.44 ± 2.74 | 272 |
| $AMM_1$ | 69.57 ± 3.99 | 15 | 73.12 ± 3.41 | 15 | 68.25 ± 2.80 | 16 | 71.02 ± 5.46 | 17 | 81.70 ± 3.02 | 19 |
| $AMM_{10ran}$ | 77.82 ± 9.12 | 192 | 68.82 ± 4.73 | 138 | 73.58 ± 4.29 | 146 | 72.21 ± 9.35 | 164 | 74.16 ± 5.25 | 188 |
| SVM alter-∝ | 53.26 ± 2.07 | 25 | 51.08 ± 2.35 | 27 | 50.90 ± 1.63 | 31 | 48.29 ± 4.51 | 38 | 41.66 ± 5.11 | 64 |
| conv-∝ | 77.80 ± 6.16 | 3924 | 66.14 ± 4.68 | 3790 | 57.94 ± 18.54 | 3244 | 61.37 ± 21.17 | 3327 | 63.73 ± 11.33 | 3603 |
| Oracle | 92.81 ± 2.89 | <1 | 92.68 ± 2.24 | <1 | 92.44 ± 3.01 | ,1 | 92.61 ± 2.03 | <1 | 92.99 ± 3.58 | <1 |

## Table 5: *breastw*

| algorithm | 2 bags AUC | time(s) | 4 bags AUC | time(s) | 8 bags AUC | time(s) | 16 bags AUC | time(s) | 32 bags AUC | time(s) |
|---|---|---|---|---|---|---|---|---|---|---|
| EMM | 48.65 ± 7.54 | <1 | 71.45 ± 16.59 | <1 | 61.68 ± 7.47 | <1 | 34.88 ± 12.33 | <1 | 47.50 ± 22.77 | <1 |
| MM | **99.42 ± 0.44** | 2 | **99.30 ± 0.39** | <1 | **99.28 ± 0.25** | <1 | **99.28 ± 0.37** | <1 | 99.18 ± 0.47 | 1 |
| $LMM_G$ | **99.42 ± 0.44** | 6 | **99.33 ± 0.38** | 3 | **99.28 ± 0.25** | 3 | 99.35 ± 0.39 | 3 | 99.22 ± 0.46 | 4 |
| $LMM_{G,s}$ | **99.42 ± 0.44** | 20 | **99.34 ± 0.39** | 10 | **99.37 ± 0.24 ↑** | 11 | 99.36 ± 0.38 | 12 | 99.23 ± 0.44 | 15 |
| $LMM_{nc}$ | **99.42 ± 0.44** | 41 | 99.29 ± 0.40 | 39 | 99.27 ± 0.25 | 41 | 99.30 ± 0.38 | 59 | 99.20 ± 0.47 | 125 |
| Invcal | 19.67 ± 2.23 | 5 | 59.50 ± 5.86 | 5 | 68 ± 5.27 | 5 | 60.83 ± 3.17 | 5 | 51.81 ± 4.72 | 5 |
| $AMM^{min}$ $AMM_{EMM}$ | **99.37 ± 0.42** | 1 | **99.33 ± 0.39** | 1 | 99.17 ± 0.54 | 1 | **99.34 ± 0.40** | 2 | 99.29 ± 0.49 | 2 |
| $AMM_{MM}$ | **99.34 ± 0.46** | 2 | **99.36 ± 0.37** | 1 | **99.36 ± 0.27 ↑** | 2 | 99.29 ± 0.41 | 2 | 99.29 ± 0.48 | 2 |
| $AMM_G$ | **99.34 ± 0.46** | 8 | **99.30 ± 0.37 ↑** | 5 | **99.36 ± 0.27 ↑** | 6 | 99.29 ± 0.41 | 7 | 99.30 ± 0.49 | 8 |
| $AMM_{G,s}$ | **99.34 ± 0.46** | 23 | **99.30 ± 0.37 ↑** | 16 | **99.36 ± 0.27 ↑** | 19 | 99.29 ± 0.41 | 20 | 99.30 ± 0.49 | 25 |
| $AMM_{nc}$ | **99.34 ± 0.46** | 43 | **99.31 ± 0.35** | 41 | **99.36 ± 0.27 ↑** | 44 | 99.29 ± 0.41 | 62 | 99.29 ± 0.48 | 129 |
| $AMM_1$ | **99.35 ± 0.45** | <1 | **99.32 ± 0.37** | 1 | 99.20 ± 0.45 | 1 | 99.30 ± 0.42 | 1 | **99.31 ± 0.48** | 2 |
| $AMM_{10ran}$ | **99.36 ± 0.45** | 8 | 99.11 ± 0.56 | 9 | 99.26 ± 0.35 | 11 | 99.28 ± 0.43 | 11 | **99.32 ± 0.49 ↑** | 14 |
| $AMM^{max}$ $AMM_{EMM}$ | **99.42 ± 0.55** | 6 | 99.02 ± 0.66 | 6 | **99.32 ± 0.25 ↑** | 6 | **99.43 ± 0.30 ↑** | 7 | **99.40 ± 0.38 ↑** | 9 |
| $AMM_{MM}$ | 99.01 ± 1.12 | 6 | 99.00 ± 0.64 | 6 | **99.32 ± 0.35 ↑** | 6 | 99.37 ± 0.38 | 7 | **99.39 ± 0.39 ↑** | 9 |
| $AMM_G$ | 99.01 ± 1.12 | 20 | 98.99 ± 0.64 | 17 | **99.33 ± 0.35 ↑** | 18 | 99.37 ± 0.38 | 21 | **99.41 ± 0.39 ↑** | 27 |
| $AMM_{G,s}$ | 99.01 ± 1.12 | 60 | 98.99 ± 0.64 | 52 | 99.19 ± 0.45 | 55 | 99.37 ± 0.38 | 63 | **99.41 ± 0.39 ↑** | 82 |
| $AMM_{nc}$ | 99.01 ± 1.12 | 55 | 98.99 ± 0.64 | 53 | **99.32 ± 0.35 ↑** | 56 | 99.37 ± 0.39 | 76 | **99.40 ± 0.38 ↑** | 148 |
| $AMM_1$ | 99.09 ± 1.08 | 5 | 99.09 ± 0.46 | 5 | **99.29 ± 0.26** | 5 | 99.37 ± 0.38 | 6 | **99.40 ± 0.38 ↑** | 8 |
| $AMM_{10ran}$ | 98.97 ± 1.29 | 47 | 98.58 ± 0.75 | 48 | **99.39 ± 0.27 ↑** | 52 | 99.37 ± 0.38 | 61 | **99.36 ± 0.41 ↑** | 81 |
| SVM alter-∝ | 68.63 ± 17.63 | 24 | 93.24 ± 4.43 | 25 | 75.17 ± 7.19 | 33 | 90.11 ± 2.58 | 42 | 18.23 ± 5.67 | 82 |
| conv-∝ | **99.41 ± 0.48** | 3346 | 56.33 ± 4.28 | 3043 | 77.71 ± 15.51 | 2800 | 32.90 ± 7.24 | 3036 | 67.21 ± 8.19 | 2037 |
| Oracle | 99.48 ± 0.41 | <1 | 99.53 ± 0.41 | <1 | 99.31 ± 0.37 | <1 | 99.43 ± 0.39 | <1 | 99.32 ± 0.44 | <1 |

## Table 6: *colic*

| algorithm | 2 bags AUC | time(s) | 4 bags AUC | time(s) | 8 bags AUC | time(s) | 16 bags AUC | time(s) | 32 bags AUC | time(s) |
|---|---|---|---|---|---|---|---|---|---|---|
| EMM | $60.69 \pm 11.30$ | <1 | $51.83 \pm 6.36$ | <1 | $52.99 \pm 5.37$ | <1 | $53.83 \pm 11.49$ | <1 | $52.95 \pm 13.28$ | <1 |
| MM | $\mathbf{62.00 \pm 6.44}$ | <1 | $70.48 \pm 7.43$ | <1 | $67.13 \pm 9.85$ | 2 | $72.60 \pm 9.35$ | 1 | $72.05 \pm 3.38$ | 1 |
| $LMM_G$ | $\mathbf{62.00 \pm 6.44}$ | 7 | $70.37 \pm 7.47$ | 6 | $72.15 \pm 8.51$ | 8 | $75.96 \pm 10.38$ | 8 | $75.47 \pm 3.59$ | 9 |
| $LMM_{G,s}$ | $\mathbf{62.00 \pm 6.44}$ | 20 | $\mathbf{72.10 \pm 6.26}$ | 20 | $\mathbf{75.08 \pm 7.14}$ | 28 | $\mathbf{78.54 \pm 10.20}$ | 26 | $\mathbf{76.43 \pm 3.10}$ | 27 |
| $LMM_{nc}$ | $\mathbf{62.00 \pm 6.44}$ | 31 | $70.45 \pm 7.46$ | 33 | $68.38 \pm 9.69$ | 52 | $74.04 \pm 10.02$ | 112 | $72.87 \pm 3.20$ | 345 |
| Invcal | $38.73 \pm 5.43$ | 6 | $65.87 \pm 6.70$ | 6 | $59.30 \pm 3.28$ | 6 | $61.54 \pm 4.17$ | 6 | $59.53 \pm 10.00$ | 6 |
| $AMM^{min}\ AMM_{EMM}$ | $59.12 \pm 8.86$ | 3 | $56.23 \pm 8.49$ | 3 | $70.93 \pm 10.31$ | 3 | $\mathbf{78.22 \pm 6.00}$ | 3 | $74.22 \pm 6.35$ | 4 |
| $AMM^{min}\ AMM_{MM}$ | $\mathbf{77.44 \pm 3.16}$ | 2 | $78.84 \pm 6.95$ | 3 | $69.46 \pm 6.44$ | 4 | $71.93 \pm 7.61$ | 4 | $81.44 \pm 5.18$ | 4 |
| $AMM^{min}\ AMM_G$ | $\mathbf{77.44 \pm 3.16}$ | 11 | $\mathbf{79.41 \pm 2.23}$ | 12 | $72.62 \pm 5.42$ | 14 | $77.80 \pm 8.11$ | 14 | $\mathbf{84.05 \pm 2.33}$ | 16 |
| $AMM^{min}\ AMM_{G,s}$ | $\mathbf{77.44 \pm 3.16}$ | 34 | $\mathbf{79.41 \pm 2.23}$ | 36 | $71.19 \pm 5.38$ | 41 | $76.71 \pm 6.70$ | 40 | $83.27 \pm 3.14$ | 47 |
| $AMM^{min}\ AMM_{nc}$ | $\mathbf{77.44 \pm 3.16}$ | 36 | $78.33 \pm 7.35$ | 38 | $70.95 \pm 4.69$ | 57 | $74.67 \pm 9.10$ | 117 | $79.86 \pm 4.87$ | 352 |
| $AMM^{min}\ AMM_1$ | $38.69 \pm 7.18$ | 1 | $56.07 \pm 14.68$ | 2 | $\mathbf{75.14 \pm 4.78}$ | 2 | $75.36 \pm 5.64$ | 3 | $77.51 \pm 5.00$ | 3 |
| $AMM^{min}\ AMM_{10ran}$ | $37.63 \pm 4.19$ | 10 | $77.75 \pm 5.66$ | 12 | $74.95 \pm 5.64$ | 15 | $76.59 \pm 10.81$ | 17 | $78.94 \pm 4.17$ | 23 |
| $AMM^{max}\ AMM_{EMM}$ | $50.94 \pm 6.54$ | 9 | $62.44 \pm 9.94$ | 9 | $57.53 \pm 13.37$ | 15 | $53.63 \pm 14.71$ | 17 | $67.63 \pm 5.63$ | 19 |
| $AMM^{max}\ AMM_{MM}$ | $43.05 \pm 14.65$ | 8 | $75.40 \pm 4.64$ | 9 | $63.72 \pm 14.41$ | 16 | $55.37 \pm 10.19$ | 18 | $69.49 \pm 3.17$ | 20 |
| $AMM^{max}\ AMM_G$ | $43.05 \pm 14.65$ | 28 | $78.19 \pm 5.93$ | 31 | $63.14 \pm 7.53$ | 51 | $61.32 \pm 5.69$ | 57 | $68.21 \pm 9.35$ | 62 |
| $AMM^{max}\ AMM_{G,s}$ | $43.05 \pm 14.65$ | 84 | $77.91 \pm 6.36$ | 91 | $62.57 \pm 6.11$ | 151 | $64.42 \pm 10.77$ | 168 | $69.47 \pm 6.40$ | 184 |
| $AMM^{max}\ AMM_{nc}$ | $42.92 \pm 14.74$ | 52 | $73.74 \pm 7.21$ | 57 | $60.39 \pm 12.21$ | 94 | $62.46 \pm 15.13$ | 162 | $68.63 \pm 2.37$ | 381 |
| $AMM^{max}\ AMM_1$ | $51.92 \pm 19.91$ | 7 | $59.89 \pm 10.79$ | 8 | $58.76 \pm 12.16$ | 14 | $62.31 \pm 13.32$ | 17 | $68.25 \pm 6.42$ | 18 |
| $AMM^{max}\ AMM_{10ran}$ | $56.39 \pm 10.26$ | 60 | $71.28 \pm 8.76$ | 68 | $65.01 \pm 13.85$ | 114 | $69.59 \pm 9.96$ | 139 | $74.40 \pm 5.54$ | 159 |
| SVM alter-$\propto$ | $46.33 \pm 2.73$ | 18 | $50.82 \pm 1.21$ | 19 | $60.84 \pm 5.51$ | 23 | $62.20 \pm 3.79$ | 32 | $57.04 \pm 10.10$ | 49 |
| SVM conv-$\propto$ | $25.27 \pm 3.45$ | 1438 | $35.96 \pm 9.34$ | 1460 | $50.31 \pm 5.57$ | 1439 | $35.46 \pm 9.11$ | 1423 | $50.13 \pm 8.34$ | 1427 |
| Oracle | $86.19 \pm 4.23$ | <1 | $87.80 \pm 2.50$ | <1 | $87.05 \pm 6.05$ | <1 | $86.53 \pm 7.15$ | <1 | $87.97 \pm 2.02$ | <1 |

## Table 7: *german*

| algorithm | 2 bags AUC | time(s) | 4 bags AUC | time(s) | 8 bags AUC | time(s) | 16 bags AUC | time(s) | 32 bags AUC | time(s) |
|---|---|---|---|---|---|---|---|---|---|---|
| EMM | $47.90 \pm 4.51$ | <1 | $50.11 \pm 5.17$ | <1 | $46.02 \pm 5.88$ | <1 | $50.94 \pm 1.61$ | <1 | $51.02 \pm 2.55$ | <1 |
| MM | $\mathbf{61.07 \pm 5.57}$ | <1 | $62.09 \pm 4.00$ | <1 | $65.50 \pm 6.54$ | 2 | $65.61 \pm 6.05$ | 2 | $66.96 \pm 4.56$ | 2 |
| $LMM_G$ | $\mathbf{61.07 \pm 5.57}$ | 4 | $62.14 \pm 4.04$ | 4 | $67.07 \pm 6.36$ | 6 | $66.43 \pm 6.61$ | 6 | $70.18 \pm 4.76$ | 7 |
| $LMM_{G,s}$ | $\mathbf{61.07 \pm 5.57}$ | 11 | $62.75 \pm 3.32$ | 12 | $\mathbf{67.91 \pm 5.80}$ | 16 | $\mathbf{66.40 \pm 6.90}$ | 19 | $70.43 \pm 5.57$ | 21 |
| $LMM_{nc}$ | $\mathbf{61.07 \pm 5.57}$ | 103 | $62.04 \pm 4.00$ | 87 | $65.47 \pm 6.56$ | 87 | $65.61 \pm 6.06$ | 113 | $67.01 \pm 4.58$ | 209 |
| Invcal | $38.74 \pm 5.43$ | 6 | $\mathbf{65.87 \pm 6.70}$ | 6 | $59.30 \pm 3.28$ | 6 | $61.53 \pm 4.17$ | 6 | $59.54 \pm 10.00$ | 6 |
| $AMM^{min}\ AMM_{EMM}$ | $53.89 \pm 6.82$ | 7 | $48.63 \pm 8.71$ | 7 | $53.24 \pm 8.02$ | 8 | $57.58 \pm 3.44$ | 9 | $63.64 \pm 11.82$ | 11 |
| $AMM^{min}\ AMM_{MM}$ | $\mathbf{60.45 \pm 5.58}$ | 5 | $63.33 \pm 4.99$ | 6 | $74.58 \pm 4.76$ | 6 | $72.43 \pm 1.39$ | 8 | $75.84 \pm 5.24$ | 7 |
| $AMM^{min}\ AMM_G$ | $\mathbf{60.45 \pm 5.58}$ | 17 | $\mathbf{64.16 \pm 6.99}$ | 18 | $74.18 \pm 4.34$ | 21 | $72.08 \pm 1.24$ | 22 | $\mathbf{75.94 \pm 4.55}$ | 24 |
| $AMM^{min}\ AMM_{G,s}$ | $\mathbf{60.45 \pm 5.58}$ | 52 | $\mathbf{64.20 \pm 7.24}$ | 57 | $74.29 \pm 4.50$ | 57 | $72.18 \pm 1.37$ | 66 | $75.77 \pm 4.44$ | 74 |
| $AMM^{min}\ AMM_{nc}$ | $\mathbf{60.45 \pm 5.58}$ | 118 | $63.20 \pm 6.09$ | 101 | $\mathbf{75.37 \pm 4.42}$ | 100 | $72.53 \pm 1.25$ | 130 | $\mathbf{75.99 \pm 5.26}$ | 225 |
| $AMM^{min}\ AMM_1$ | $37.08 \pm 4.42$ | 3 | $38.53 \pm 2.97$ | 3 | $41.89 \pm 2.07$ | 6 | $41.13 \pm 2.58$ | 9 | $47.09 \pm 9.40$ | 10 |
| $AMM^{min}\ AMM_{10ran}$ | $49.12 \pm 6.50$ | 36 | $60.31 \pm 5.57$ | 38 | $73.82 \pm 4.70$ | 44 | $72.07 \pm 3.22$ | 54 | $74.73 \pm 4.54$ | 72 |
| $AMM^{max}\ AMM_{EMM}$ | $46.45 \pm 3.30$ | 18 | $46.31 \pm 3.02$ | 19 | $67.34 \pm 13.42$ | 19 | $72.41 \pm 6.17$ | 20 | $74.58 \pm 4.63$ | 22 |
| $AMM^{max}\ AMM_{MM}$ | $52.47 \pm 8.88$ | 18 | $58.61 \pm 12.19$ | 18 | $65.14 \pm 21.84$ | 19 | $74.90 \pm 4.86$ | 20 | $74.88 \pm 3.75$ | 22 |
| $AMM^{max}\ AMM_G$ | $52.47 \pm 8.88$ | 54 | $56.12 \pm 12.25$ | 53 | $74.93 \pm 8.18$ | 57 | $73.87 \pm 4.55$ | 60 | $75.43 \pm 4.02$ | 67 |
| $AMM^{max}\ AMM_{G,s}$ | $52.47 \pm 8.88$ | 160 | $54.79 \pm 11.61$ | 158 | $74.84 \pm 8.12$ | 167 | $73.87 \pm 4.55$ | 180 | $75.40 \pm 4.05$ | 197 |
| $AMM^{max}\ AMM_{nc}$ | $52.47 \pm 8.88$ | 154 | $49.24 \pm 12.68$ | 137 | $65.11 \pm 21.84$ | 137 | $74.89 \pm 4.75$ | 167 | $74.70 \pm 3.71$ | 269 |
| $AMM^{max}\ AMM_1$ | $58.39 \pm 13.20$ | 17 | $61.04 \pm 14.43$ | 17 | $69.66 \pm 16.93$ | 17 | $\mathbf{76.49 \pm 3.29}$ | 18 | $75.44 \pm 3.65$ | 20 |
| $AMM^{max}\ AMM_{10ran}$ | $50.47 \pm 9.69$ | 168 | $56.78 \pm 10.89$ | 164 | $60.41 \pm 15.48$ | 160 | $61.62 \pm 18.81$ | 170 | $73.25 \pm 6.97$ | 191 |
| SVM alter-$\propto$ | $49.36 \pm 1.68$ | 34 | $49.59 \pm 1.58$ | 37 | $48.43 \pm 2.23$ | 40 | $48.85 \pm 1.55$ | 47 | $51.05 \pm 2.72$ | 64 |
| SVM conv-$\propto$ | $29.70 \pm 2.03$ | 6031 | $64.15 \pm 5.43$ | 6343 | $63.01 \pm 2.59$ | 6362 | $62.01 \pm 3.61$ | 6765 | $63.17 \pm 3.62$ | 7004 |
| Oracle | $79.43 \pm 2.88$ | <1 | $78.95 \pm 3.99$ | <1 | $79.18 \pm 1.70$ | <1 | $79.42 \pm 2.80$ | <1 | $79.02 \pm 3.62$ | <1 |

## Table 8: *heart*

| algorithm | 2 bags AUC | time(s) | 4 bags AUC | time(s) | 8 bags AUC | time(s) | 16 bags AUC | time(s) | 32 bags AUC | time(s) |
|---|---|---|---|---|---|---|---|---|---|---|
| EMM | $51.82 \pm 12.39$ | <1 | $50.43 \pm 23.03$ | <1 | $55.09 \pm 19.44$ | <1 | $49.55 \pm 17.47$ | <1 | $63.49 \pm 18.11$ | <1 |
| MM | $\mathbf{68.75 \pm 6.09}$ | <1 | $60.24 \pm 13.54$ | <1 | $80.35 \pm 9.42$ | <1 | $76.11 \pm 6.66$ | 1 | $83.50 \pm 6.22$ | 1 |
| $LMM_G$ | $\mathbf{68.75 \pm 6.09}$ | 3 | $68.04 \pm 8.53$ | 3 | $82.87 \pm 6.16$ | 4 | $\mathbf{82.92 \pm 1.28}$ | 4 | $85.85 \pm 3.84$ | 6 |
| $LMM_{G,s}$ | $\mathbf{68.75 \pm 6.09}$ | 9 | $69.04 \pm 6.52$ | 12 | $\mathbf{83.68 \pm 5.90}$ | 13 | $\mathbf{82.96 \pm 1.79}$ | 14 | $\mathbf{86.36 \pm 3.94}$ | 17 |
| $LMM_{nc}$ | $\mathbf{68.75 \pm 6.09}$ | 11 | $60.40 \pm 14.18$ | 12 | $80.24 \pm 9.74$ | 189 | $78.14 \pm 4.98$ | 42 | $84.47 \pm 5.06$ | 119 |
| Invcal | $28.84 \pm 4.96$ | 4 | $70.58 \pm 6.45$ | 4 | $37.33 \pm 10.31$ | 4 | $44.96 \pm 9.64$ | 4 | $62.76 \pm 15.05$ | 4 |
| $AMM^{min}\ AMM_{EMM}$ | $60.50 \pm 30.88$ | <1 | $63.36 \pm 28.50$ | 1 | $72.05 \pm 19.17$ | 1 | $80.87 \pm 15.51$ | 1 | $\mathbf{91.63 \pm 6.10}\uparrow$ | 2 |
| $AMM^{min}\ AMM_{MM}$ | $86.59 \pm 6.14$ | 1 | $80.57 \pm 16.72$ | 1 | $87.96 \pm 4.50$ | 2 | $90.04 \pm 5.14$ | 2 | $91.45 \pm 5.70\uparrow$ | 2 |
| $AMM^{min}\ AMM_G$ | $86.59 \pm 6.14$ | 5 | $86.70 \pm 5.45$ | 5 | $87.46 \pm 2.67$ | 6 | $\mathbf{91.06 \pm 2.87}$ | 7 | $91.55 \pm 5.93\uparrow$ | 9 |
| $AMM^{min}\ AMM_{G,s}$ | $86.59 \pm 6.14$ | 15 | $86.70 \pm 5.45$ | 16 | $88.31 \pm 4.00$ | 18 | $90.86 \pm 2.81$ | 21 | $91.55 \pm 5.93\uparrow$ | 27 |
| $AMM^{min}\ AMM_{nc}$ | $86.59 \pm 6.14$ | 13 | $78.97 \pm 16.78$ | 14 | $87.82 \pm 4.42$ | 21 | $90.48 \pm 3.53$ | 45 | $91.25 \pm 5.77$ | 125 |
| $AMM^{min}\ AMM_1$ | $\mathbf{90.62 \pm 5.82}$ | <1 | $89.19 \pm 5.90$ | 1 | $88.64 \pm 3.21$ | 1 | $90.78 \pm 2.10$ | 1 | $91.03 \pm 5.82$ | 1 |
| $AMM^{min}\ AMM_{10ran}$ | $78.38 \pm 30.44$ | 5 | $87.32 \pm 4.71$ | 6 | $89.85 \pm 2.31$ | 7 | $91.02 \pm 2.49$ | 9 | $90.47 \pm 6.39$ | 14 |
| $AMM^{max}\ AMM_{EMM}$ | $85.74 \pm 13.28$ | 3 | $84.60 \pm 10.87$ | 4 | $84.60 \pm 7.84$ | 3 | $89.83 \pm 2.72$ | 5 | $71.65 \pm 18.52$ | 6 |
| $AMM^{max}\ AMM_{MM}$ | $85.35 \pm 11.06$ | 4 | $82.43 \pm 9.76$ | 4 | $90.49 \pm 4.75$ | 4 | $89.92 \pm 2.90$ | | $89.35 \pm 6.98$ | 7 |
| $AMM^{max}\ AMM_G$ | $85.35 \pm 11.06$ | 13 | $87.18 \pm 6.56$ | 13 | $\mathbf{90.49 \pm 4.75}$ | 13 | $89.58 \pm 2.79$ | 16 | $88.55 \pm 9.71$ | 23 |
| $AMM^{max}\ AMM_{G,s}$ | $85.35 \pm 11.06$ | 39 | $\mathbf{90.49 \pm 5.05}$ | 40 | $\mathbf{90.58 \pm 4.77}$ | 40 | $89.58 \pm 2.79$ | 49 | $89.94 \pm 6.63$ | 67 |
| $AMM^{max}\ AMM_{nc}$ | $85.35 \pm 11.06$ | 20 | $82.73 \pm 9.23$ | 21 | $89.84 \pm 4.24$ | 30 | $90.06 \pm 3.20$ | 54 | $89.54 \pm 6.60$ | 140 |
| $AMM^{max}\ AMM_1$ | $72.77 \pm 37.27$ | 4 | $89.31 \pm 3.99$ | 3 | $89.68 \pm 3.79$ | 3 | $90.62 \pm 3.18$ | 5 | $87.97 \pm 9.42$ | 6 |
| $AMM^{max}\ AMM_{10ran}$ | $89.96 \pm 5.62$ | 32 | $89.93 \pm 5.02$ | 31 | $88.03 \pm 3.16$ | 30 | $90.80 \pm 3.61$ | 38 | $89.61 \pm 8.68$ | 54 |
| SVM alter-$\propto$ | $47.75 \pm 17.58$ | 15 | $59.72 \pm 18.21$ | 16 | $62.32 \pm 12.83$ | 20 | $58.49 \pm 10.98$ | 27 | $48.33 \pm 12.77$ | 47 |
| SVM conv-$\propto$ | $46.18 \pm 43.41$ | 1211 | $87.13 \pm 5.30$ | 1185 | $69.03 \pm 23.18$ | 1197 | $42.78 \pm 23.51$ | 1188 | $50.34 \pm 15.75$ | 1080 |
| Oracle | $91.72 \pm 3.95$ | <1 | $91.22 \pm 4.09$ | <1 | $91.27 \pm 2.88$ | <1 | $91.54 \pm 2.76$ | <1 | $91.42 \pm 5.46$ | <1 |

## Table 9: *ionosphere*

| algorithm | 2 bags AUC | time(s) | 4 bags AUC | time(s) | 8 bags AUC | time(s) | 16 bags AUC | time(s) | 32 bags AUC | time(s) |
|---|---|---|---|---|---|---|---|---|---|---|
| EMM | 44.28 ± 12.13 | <1 | 51.86 ± 8.01 | <1 | 50.69 ± 6.34 | <1 | 44.60 ± 3.91 | <1 | 48.91 ± 11.73 | <1 |
| MM | **64.81 ± 8.82** | <1 | 77.74 ± 5.23 | 1 | 78.95 ± 7.36 | 1 | 86.76 ± 2.96 | 1 | 88.13 ± 4.16 | 2 |
| $LMM_G$ | **64.81 ± 8.82** | 5 | 80.80 ± 2.32 | 6 | **83.46 ± 4.62** | 5 | 87.12 ± 2.23 | 7 | **88.24 ± 4.41** | 7 |
| $LMM_{G,s}$ | **64.81 ± 8.82** | 14 | **82.12 ± 2.50** | 15 | 83.24 ± 4.84 | 15 | **87.23 ± 1.57** | 17 | 87.99 ± 4.58 | 21 |
| $LMM_{nc}$ | **64.81 ± 8.82** | 20 | 79.39 ± 2.12 | 22 | 81.18 ± 6.40 | 32 | 87.05 ± 2.48 | 68 | **88.34 ± 4.32** | 182 |
| Invcal | 35.34 ± 8.76 | 5 | 44.78 ± 15.37 | 5 | 53.28 ± 9.02 | 5 | 53.52 ± 8.51 | 5 | 54.08 ± 9.53 | 5 |
| $AMM^{min}$ $AMM_{EMM}$ | 56.77 ± 6.42 | 2 | **85.07 ± 5.24** | 2 | 86.04 ± 5.21 | 2 | 86.81 ± 3.81 | 2 | 86.71 ± 3.54 | 3 |
| $AMM_{MM}$ | 46.67 ± 8.53 | 3 | 84.52 ± 4.60 | 2 | 84.23 ± 6.67 | 2 | 85.92 ± 4.48 | 3 | 87.77 ± 5.56 | 3 |
| $AMM_G$ | 46.67 ± 8.53 | 10 | 85.05 ± 4.11 | 9 | 85.28 ± 6.19 | 9 | 85.97 ± 3.19 | 11 | 88.85 ± 5.15 | 12 |
| $AMM_{G,s}$ | 46.67 ± 8.53 | 28 | 84.63 ± 3.80 | 26 | 85.28 ± 6.19 | 27 | 86.01 ± 4.37 | 30 | 88.85 ± 5.15 | 36 |
| $AMM_{nc}$ | 46.67 ± 8.53 | 24 | **85.16 ± 4.39** | 26 | 84.77 ± 6.45 | 36 | 85.96 ± 4.50 | 72 | 87.57 ± 5.23 | 174 |
| $AMM_1$ | 51.47 ± 13.46 | 1 | 83.65 ± 3.89 | 2 | **87.51 ± 4.24** | 2 | 86.76 ± 4.07 | 2 | 87.83 ± 5.05 | 2.11 |
| $AMM_{10ran}$ | 56.92 ± 22.42 | 10 | 80.39 ± 6.36 | 11 | 85.89 ± 5.52 | 12 | 87.32 ± 3.17 | 13 | 87.81 ± 6.52 | 15 |
| $AMM^{max}$ $AMM_{EMM}$ | 57.99 ± 8.96 | 10 | 76.31 ± 5.29 | 10 | 82.07 ± 4.47 | 11 | 86.99 ± 7.23 | 11 | 87.08 ± 5.86 | 12 |
| $AMM_{MM}$ | **74.57 ± 18.16** | 10 | 75.32 ± 4.74 | 10 | 78.65 ± 7.93 | 11 | 88.84 ± 3.10 | 12 | **90.01 ± 5.50** | 13 |
| $AMM_G$ | **74.57 ± 18.16** | 32 | 78.06 ± 5.11 | 33 | 83.24 ± 6.54 | 35 | 89.98 ± 3.08 ↑ | 38 | 88.41 ± 5.94 | 41 |
| $AMM_{G,s}$ | **74.57 ± 18.16** | 96 | 79.21 ± 4.58 | 98 | 83.36 ± 6.61 | 104 | **90.88 ± 3.11** ↑ | 112 | 88.41 ± 5.94 | 121 |
| $AMM_{nc}$ | **74.57 ± 18.16** | 47 | 75.80 ± 5.14 | 50 | 80.22 ± 6.95 | 61 | 88.05 ± 2.47 | 99 | 89.19 ± 5.45 | 198 |
| $AMM_1$ | 65.53 ± 17.30 | 10 | 77.29 ± 6.63 | 9 | 82.10 ± 7.95 | 10 | 85.45 ± 3.31 | 11 | 89.01 ± 7.02 | 12 |
| $AMM_{10ran}$ | 65.05 ± 16.59 | 85 | 79.60 ± 6.56 | 82 | 78.56 ± 4.77 | 88 | 88.44 ± 3.22 | 94 | 89.37 ± 6.67 | 109 |
| SVM alter-∝ | 43.07 ± 6.05 | 22 | 44.58 ± 4.95 | 24 | 69.24 ± 4.99 | 27 | 67.72 ± 12.25 | 55 | 59.67 ± 7.01 | 49 |
| conv-∝ | 36.67 ± 7.44 | 1316 | 44.55 ± 9.58 | 1280 | 57.84 ± 5.98 | 1788 | 65.93 ± 3.90 | 887 | 47.58 ± 11.29 | 1287 |
| Oracle | 90.07 ± 5.04 | <1 | 89.99 ± 4.23 | <1 | 90.08 ± 5.50 | <1 | 89.42 ± 6.34 | <1 | 90.22 ± 5.17 | <1 |

## Table 10: *vertebral column*

| algorithm | 2 bags AUC | time(s) | 4 bags AUC | time(s) | 8 bags AUC | time(s) | 16 bags AUC | time(s) | 32 bags AUC | time(s) |
|---|---|---|---|---|---|---|---|---|---|---|
| EMM | 57.91 ± 22.04 | <1 | 59.05 ± 10.46 | <1 | 51.43 ± 17.22 | <1 | 45.39 ± 23.81 | <1 | 61.30 ± 17.86 | <1 |
| MM | 77.45 ± 6.14 | <1 | 78.97 ± 3.54 | <1 | 79.85 ± 4.14 | <1 | 82.74 ± 2.11 | 1 | 87.45 ± 3.57 | 1 |
| $LMM_G$ | 77.45 ± 6.14 | 3 | 78.34 ± 2.82 | 3 | 81.93 ± 3.81 | 3 | 87.52 ± 2.71 | 5 | 90.43 ± 3.20 | 6 |
| $LMM_{G,s}$ | 77.45 ± 6.14 | 9 | 78.34 ± 2.82 | 8 | **83.87 ± 3.63** | 9 | **87.71 ± 2.56** | 13 | **91.06 ± 3.00** | 14 |
| $LMM_{nc}$ | 77.45 ± 6.14 | 31 | 78.43 ± 2.74 | 31 | 80.02 ± 4.02 | 35 | 83.50 ± 2.46 | 54 | 88.10 ± 3.57 | 122 |
| InvCal | 33.74 ± 24.95 | 4 | 36.46 ± 5.27 | 4 | 72.54 ± 5.79 | 4 | 61.89 ± 6.25 | 4 | 59.91 ± 8.79 | 4 |
| $AMM^{min}$ $AMM_{EMM}$ | **81.07 ± 8.12** | 2 | 78.56 ± 8.66 | 2 | 90.56 ± 3.44 | 2 | 92.08 ± 1.78 | 2 | 93.14 ± 2.04 | 3 |
| $AMM_{MM}$ | 75.64 ± 5.02 | 2 | 68.54 ± 4.90 | 2 | 87.10 ± 4.16 | 2 | **92.66 ± 1.99** | 3 | 93.50 ± 1.93 | 3 |
| $AMM_G$ | 75.64 ± 5.02 | 6 | 69.27 ± 5.69 | 7 | 87.57 ± 4.48 | 8 | 92.45 ± 1.89 | 10 | 93.59 ± 1.83 | 11 |
| $AMM_{G,s}$ | 75.64 ± 5.02 | 19 | 69.27 ± 5.69 | 22 | 87.86 ± 4.62 | 23 | 91.04 ± 3.82 | 30 | 92.97 ± 1.58 | 32 |
| $AMM_{nc}$ | 75.64 ± 5.02 | 34 | 68.49 ± 4.86 | 35 | 88.33 ± 5.17 | 39 | 91.26 ± 3.98 | 59 | **93.70 ± 2.09** | 127 |
| $AMM_1$ | 74.49 ± 6.08 | 1 | 68.66 ± 4.92 | 1 | 90.60 ± 3.18 | 2 | 92.41 ± 1.58 | 2 | 92.95 ± 1.75 | 2 |
| $AMM_{10ran}$ | 76.42 ± 4.80 | 12 | 75.75 ± 5.07 | 16 | **92.59 ± 0.22** | 18 | 92.15 ± 1.44 | 15 | 92.46 ± 1.79 | 19 |
| $AMM^{max}$ $AMM_{EMM}$ | 76.02 ± 12.70 | 4 | 78.42 ± 14.14 | 5 | 87.87 ± 1.94 | 5 | 87.88 ± 3.29 | 6 | 90.71 ± 2.79 | 8 |
| $AMM_{MM}$ | 75.31 ± 13.69 | 5 | **87.22 ± 3.13** | 5 | 88.43 ± 2.59 | 6 | 88.85 ± 2.39 | 6 | 90.29 ± 2.47 | 9 |
| $AMM_G$ | 75.31 ± 13.69 | 15 | 73.91 ± 16.06 | 17 | 87.89 ± 1.97 | 17 | 87.98 ± 3.27 | 21 | 90.29 ± 2.47 | 28 |
| $AMM_{G,s}$ | 75.31 ± 13.69 | 44 | 67.48 ± 16.70 | 50 | 87.89 ± 1.97 | 51 | 87.98 ± 3.27 | 63 | 90.18 ± 3.26 | 82 |
| $AMM_{nc}$ | 75.31 ± 13.69 | 43 | 82.97 ± 8.05 | 45 | 87.85 ± 2.00 | 49 | 88.91 ± 2.41 | 70 | 90.29 ± 2.47 | 144 |
| $AMM_1$ | 77.35 ± 13.61 | 4 | 70.14 ± 17.19 | 5 | 84.17 ± 2.66 | 5 | 89.12 ± 2.31 | 6 | 90.94 ± 3.06 | 8 |
| $AMM_{10ran}$ | 72.39 ± 14.33 | 36 | 82.49 ± 9.32 | 47 | 87.44 ± 1.52 | 47 | 85.79 ± 4.54 | 50 | 90.87 ± 2.53 | 69 |
| SVM alter-∝ | 40.88 ± 5.80 | 21 | 30.17 ± 7.47 | 23 | 68.26 ± 6.40 | 26 | 58.84 ± 21.21 | 33 | 37.17 ± 17.48 | 48 |
| conv-∝ | 77.72 ± 6.23 | 3624 | 72.28 ± 8.88 | 2292 | 36.21 ± 8.38 | 2328 | 45.01 ± 14.91 | 2481 | 70.49 ± 5.59 | 2306 |
| Oracle | 93.80 ± 1.06 | <1 | 93.83 ± 1.67 | <1 | 93.89 ± 1.89 | <1 | 93.83 ± 1.62 | <1 | 94.00 ± 1.42 | <1 |

## Table 11: *vote* (feature *physician-fee-freeze* was removed to make the problem harder)

| algorithm | 2 bags AUC | time(s) | 4 bags AUC | time(s) | 8 bags AUC | time(s) | 16 bags AUC | time(s) | 32 bags AUC | time(s) |
|---|---|---|---|---|---|---|---|---|---|---|
| EMM | 54.32 ± 8.79 | <1 | 45.47 ± 15.63 | <1 | 46.88 ± 6.06 | 1 | 55.20 ± 18.03 | 1 | 53.93 ± 10.59 | 1 |
| MM | 94.56 ± 2.04 | 1 | 95.37 ± 2.62 | 2 | 95.65 ± 0.85 | 2 | **96.33 ± 1.19** | 2 | 96.74 ± 1.50 | 2 |
| $LMM_G$ | 94.56 ± 2.04 | 7 | 95.93 ± 2.47 | 8 | 95.87 ± 1.12 | 8 | **96.41 ± 1.51** | 9 | **96.94 ± 1.67** | 10 |
| $LMM_{G,s}$ | 94.56 ± 2.04 | 20 | **96.03 ± 2.42** | 22 | **96.00 ± 1.18** | 23 | 96.38 ± 1.99 | 25 | 96.81 ± 2.09 | 28 |
| $LMM_{nc}$ | 94.56 ± 2.04 | 28 | 95.83 ± 2.34 | 31 | 95.71 ± 0.92 | 43 | 96.23 ± 1.58 | 85 | 96.81 ± 1.50 | 234 |
| Invcal | **94.85 ± 1.71** | 4 | 73.10 ± 2.21 | 4 | 77.86 ± 4.92 | 4 | 26.74 ± 6.82 | 4 | 79.77 ± 6.25 | 4 |
| $AMM^{min}$ $AMM_{EMM}$ | 93.67 ± 1.84 | 2 | 95.04 ± 3.01 | 2 | **96.18 ± 0.78** | 2 | 96.43 ± 1.31 | 2 | 96.94 ± 1.62 | 3 |
| $AMM_{MM}$ | 93.48 ± 2.31 | 2 | 95.12 ± 2.89 | 3 | 96.10 ± 0.82 | 3 | 96.15 ± 1.31 | 4 | **97.30 ± 1.58** | 4 |
| $AMM_G$ | 93.48 ± 2.31 | 10 | **95.61 ± 1.90** | 12 | 95.92 ± 1.02 | 11 | 96.41 ± 1.12 | 13 | **97.36 ± 1.47** | 15 |
| $AMM_{G,s}$ | 93.48 ± 2.31 | 29 | 94.87 ± 3.02 | 33 | 95.34 ± 0.98 | 35 | 96.11 ± 1.30 | 39 | **97.36 ± 1.47** | 46 |
| $AMM_{nc}$ | 93.48 ± 2.31 | 32 | 95.38 ± 2.38 | 35 | 95.81 ± 1.01 | 46 | 96.03 ± 1.48 | 89 | **97.38 ± 1.45** | 238 |
| $AMM_1$ | 93.57 ± 1.99 | 2 | 94.32 ± 3.36 | 2 | **96.25 ± 0.66** | 2 | 96.17 ± 1.20 | 2 | 96.83 ± 1.42 | 2 |
| $AMM_{10ran}$ | **93.84 ± 2.23** | 11 | 94.59 ± 3.56 | 11 | 95.85 ± 0.97 | 12 | **96.63 ± 1.32** | 15 | 96.66 ± 1.70 | 18 |
| $AMM^{max}$ $AMM_{EMM}$ | 91.68 ± 0.81 | 11 | 94.97 ± 2.24 | 12 | 94.94 ± 1 | 13 | 95.83 ± 1.36 | 14 | 96.60 ± 1.31 | 15 |
| $AMM_{MM}$ | 92.47 ± 0.38 | 12 | 93.43 ± 4.07 | 13 | 93.71 ± 1.34 | 14 | 95.40 ± 1.10 | 15 | 96.77 ± 1.31 | 17 |
| $AMM_G$ | 92.47 ± 0.38 | 40 | 94.34 ± 2.65 | 34 | 94.03 ± 0.81 | 43 | 95.65 ± 1.70 | 48 | 96.45 ± 1.52 | 53 |
| $AMM_{G,s}$ | 92.47 ± 0.38 | 124 | 94.22 ± 2.87 | 127 | 94.03 ± 0.81 | 132 | 96.01 ± 1.83 | 142 | 96.37 ± 1.39 | 160 |
| $AMM_{nc}$ | 92.47 ± 0.38 | 65 | 94.96 ± 3.48 | 66 | 94.07 ± 0.78 | 78 | 95.14 ± 1.18 | 124 | 96.74 ± 1.31 | 275 |
| $AMM_1$ | 91.60 ± 1.29 | 11 | 94.48 ± 2.14 | 12 | 94.34 ± 0.82 | 12 | 95.36 ± 1.56 | 13 | 96.54 ± 1.51 | 15 |
| $AMM_{10ran}$ | 90.49 ± 2.02 | 101 | 94.59 ± 2.85 | 103 | 94.19 ± 0.73 | 104 | 95.73 ± 1.83 | 112 | 96.21 ± 1.67 | 128 |
| SVM alter-∝ | 51.58 ± 3.27 | 19 | 62.74 ± 4.27 | 21 | 60.88 ± 3.50 | 25 | 63.01 ± 9.51 | 33 | 41.87 ± 7.12 | 57 |
| conv-∝ | 5.63 ± 2.03 | 1848 | 47.22 ± 4.92 | 1807 | 19.62 ± 5.91 | 1855 | 57.54 ± 11.22 | 1598 | 46.27 ± 9.48 | 1281 |
| Oracle | 97.11 ± 1.31 | <1 | 97.43 ± 2.25 | <1 | 97.06 ± 0.87 | <1 | 97.33 ± 1.38 | <1 | 97.52 ± 1.49 | <1 |

Table 12: *wine*

| algorithm | | 2 bags | | 4 bags | | 8 bags | | 16 bags | | 32 bags | |
|---|---|---|---|---|---|---|---|---|---|---|---|
| | | AUC | time(s) | AUC | time(s) | AUC | time(s) | AUC | time(s) | AUC | time(s) |
| | EMM | **70.38 ± 20.39** | <1 | 56.72 ± 29.85 | <1 | 55.42 ± 20.70 | <1 | 65.82 ± 21.45 | <1 | 46.85 ± 16.71 | <1 |
| | MM | 66.45 ± 5.42 | 1 | 82.41 ± 6.76 | 1 | 85.28 ± 4.80 | 1 | 90.35 ± 3.73 | 1 | 95.57 ± 2.45 | 1 |
| | LMM$_G$ | 66.45 ± 5.42 | 4 | 89.72 ± 3.73 | 5 | 90.69 ± 5.30 | 5 | 94.09 ± 3.45 | 5 | **97.74 ± 0.67** | 6 |
| | LMM$_{G,s}$ | 66.45 ± 4.412 | 13 | **93.32 ± 2.94** | 13 | **92.68 ± 6.06** | 14 | **95.53 ± 2.40** | 15 | 97.69 ± 0.90 | 19 |
| | LMM$_{nc}$ | 66.45 ± 5.42 | 9 | 84.00 ± 5.48 | 11 | 86.30 ± 4.18 | 18 | 91.10 ± 4.52 | 40 | 96.28 ± 2.06 | 116 |
| | Invcal | 58.96 ± 5.77 | 6 | 81.38 ± 4.59 | 6 | 55.18 ± 9.59 | 6 | 63.07 ± 12.61 | 6 | 71.01 ± 18.19 | 6 |
| AMM$^{min}$ | AMM$_{EMM}$ | 80.27 ± 18.08 | 1 | 90.33 ± 8.87 | 1 | 91.46 ± 10.59 | 1 | 88.97 ± 6.26 | 1 | 88.34 ± 22.79 | 2 |
| | AMM$_{MM}$ | 61.84 ± 9.20 | 2 | 85.56 ± 7.20 | 1 | 88.70 ± 8.31 | 2 | 93.78 ± 9.12 | 2 | 98.66 ± 1.11 | 2 |
| | AMM$_G$ | 61.84 ± 9.20 | 6 | 93.06 ± 7.88 | 7 | 93.42 ± 8.24 | 7 | 96.09 ± 8.18 | 7 | **99.33 ± 1.01** | 9 |
| | AMM$_{G,s}$ | 61.84 ± 9.20 | 17 | 94.87 ± 5.68 | 18 | 93.00 ± 8.95 | 20 | 96.09 ± 8.18 | 21 | **99.33 ± 1.01** | 27 |
| | AMM$_{nc}$ | 61.84 ± 9.20 | 10 | 87.03 ± 3.93 | 13 | 88.23 ± 7.90 | 20 | 97.49 ± 5.06 | 43 | **99.33 ± 1.01** | 119 |
| | AMM$_1$ | 82.21 ± 11.39 | <1 | 94.12 ± 6.34 | 1 | 99.60 ± 0.60 | 1 | 96.03 ± 7.57 | 1 | 97.03 ± 3.66 | 1 |
| | AMM$_{10ran}$ | 58.75 ± 31.30 | 4 | **99.47 ± 0.68** | 5 | 99.52 ± 0.45 | 6 | **99.59 ± 0.54** | 7 | 98.95 ± 1.66 | 10 |
| AMM$^{max}$ | AMM$_{EMM}$ | 74.23 ± 32.62 | 3 | 85.52 ± 17.48 | 4 | 99.67 ± 0.74 | 5 | 98.09 ± 3.09 | 6 | 92.00 ± 11.55 | 7 |
| | AMM$_{MM}$ | 88.23 ± 18.56 | 5 | 97.60 ± 2.40 | 4 | 87.42 ± 27.76 | 6 | 99.42 ± 0.79 | 7 | 98.61 ± 1.69 | 8 |
| | AMM$_G$ | 88.23 ± 18.56 | 15 | 88.41 ± 20 | 15 | **100.00 ± 0.00 ↑** | 19 | **99.63 ± 0.66** | 20 | 98.61 ± 1.69 | 25 |
| | AMM$_{G,s}$ | 88.23 ± 18.56 | 44 | 79.11 ± 23.90 | 44 | **100.00 ± 0.00 ↑** | 56 | **99.63 ± 0.66** | 59 | 98.61 ± 1.69 | 75 |
| | AMM$_{nc}$ | 88.23 ± 18.56 | 19 | 85.44 ± 19.04 | 21 | 86.17 ± 27.19 | 32 | 99.36 ± 0.74 | 56 | 98.61 ± 1.69 | 135 |
| | AMM$_1$ | 75.24 ± 21.10 | 3 | 80.45 ± 10.01 | 4 | 91.83 ± 14.63 | 5 | 91.79 ± 9.05 | 5 | 88.01 ± 9.78 | 7 |
| | AMM$_{10ran}$ | **97.54 ± 1.55** | 30 | 96.80 ± 3.94 | 32 | 99.46 ± 0.82 | 41 | 99.21 ± 0.79 | 47 | 98.54 ± 1.66 | 58 |
| SVM | alter-$\propto$ | 52.68 ± 2.54 | 14 | 36.53 ± 10.97 | 16 | 65.54 ± 2.26 | 19 | 29.15 ± 9.60 | 32 | 86.22 ± 11.93 | 44 |
| | conv-$\propto$ | 54.31 ± 4.63 | 831 | 70.23 ± 6.58 | 794 | 52.88 ± 13.86 | 840 | 55.60 ± 11.29 | 659 | 11.58 ± 7.84 | 495 |
| | Oracle | 99.69 ± 0.52 | <1 | 99.80 ± 0.44 | <1 | 99.60 ± 0.43 | <1 | 99.80 ± 0.44 | <1 | 99.78 ± 0.33 | <1 |

Figure 4: Relative AUC (wrt Oracle) vs entropy on *arrhythmia*

Figure 5: Relative AUC (wrt Oracle) vs entropy on *australian*

Figure 6: Relative AUC (wrt Oracle) vs entropy on *breastw*

Figure 7: Relative AUC (wrt Oracle) vs entropy on *colic*

Figure 8: Relative AUC (wrt Oracle) vs entropy on *german*

Figure 9: Relative AUC (wrt Oracle) vs entropy on *heart*

Figure 10: Relative AUC (wrt Oracle) vs entropy on *ionosphere*

Figure 11: Relative AUC (wrt Oracle) vs entropy on *vertebral column*

Figure 12: Relative AUC (wrt Oracle) vs entropy on *vote*

Figure 13: Relative AUC (wrt Oracle) vs entropy on *wine*

## Footnotes

[1]https://github.com/felixyu/pSVM