[Reviews · NeurIPS 2014]

Submitted by Assigned_Reviewer_3

This paper addresses the problem of learning with label proportions, wherein one only knows the proportions of the labels of bags of samples that are positive (or more generally that belong to one class). It makes very substantive contributions by:
a. analysis that shows that a general class of loss functions allow efficient learning via the mean operator without requiring homogeneity (as was the case with previous literature)
b. Developing fast learning algorithms for estimating the mean operator
c. allow the use of standard binary classifier learning algorithms to solve the LLP problem via reduction.
d. derive rdemacher style generalization bounds.
e. show via experimental accuracy that one can effectively learn classifiers accurately even in the LLP setting, thus showing that this setting does not really protect privacy

Excellent theoretical results, experimental results that are better than the state of the art, and finally explaining the implications for privacy protection make this a very enjoyable paper. It is also very well written, though it is quite dense because of the sheer volume of novel material that the authors try to cover in a short NIPS paper.
Summary: Thank you for a very enjoyable paper!

Excellent theoretical results, experimental results that are better than the state of the art, and finally explaining the implications for privacy protection make this a welcome contribution to NIPS.

Submitted by Assigned_Reviewer_20

The paper considers the problem of learning classifiers when each example in the training set, instead of an input/ouput pair
is a set of inputs (a bag) and the proportion of corresponding labels. The basic approach in the paper is a two steps procedure based on first estimating a so called mean operator, and then using this estimate to derive a classifier. The paper proposes a fast algorithm to estimate the mean operator (btw why would you call operator a function/vector?), with accompanying approximation guarantees. Such an algorithms is then used to initialize a suitable iterative procedure. The statistical properties of the proposed method are studied by means of generalization bounds based on Rademacher complexities. Extensive empirical evaluation are provided. The paper develops ideas in previous papers, in particular in [17].

On the good side, the paper is interesting and contains many results both from the algorithmic, theoretical and empirical point of view (there is a huge supplementary material). The proof seem flawless and the experiments very much

On the bad side, the paper is a tour de force. It is very dense and not self contained. The presentation of the material could definitely be improved.
Summary: Although the authors did not do a great job in organizing the material for a conference submission, I still think the material in the paper provide a good contribution.

Submitted by Assigned_Reviewer_44

Summary: This paper presents new algorithms for the task of learning with label proportions (LLP). One of the proposed algorithm, the Laplacian Mean Map algorithm, extends previously proposed algorithm called Mean Map algorithm. It doesn’t pose homogeneity assumption but uses the Laplacian of the bag similarities. The Alternating Mean Map algorithm, an iterative variant of the LMM that uses LMM for initialization, is also investigated. The paper presents generalization bounds for LMM and AMM. Experimental results using different datasets and domains show that the proposed methods achieve the state-of-the-art performance in many cases, yet both of the algorithms scale to a large dataset.

Quality: Detailed proof and thorough experiments clearly support the soundness and effectiveness of the proposed algorithms.

Clarity: The paper is written very clearly. The supplemental material well complement the proofs and detailed experimental results.

Originality: Although application of manifold regularization and iterative procedure themselves is a natural extension to the Mean Map algorithm, it is worth publishing.

Significance: Since the algorithms are easy to implement and scale to a large dataset, they will soon become strong baselines in LLP settings. Privacy-preserving technologies may also benefit from these algorithms.
Summary: This paper proposes two Laplacian-based algorithms for learning with label proportions. Theoretical analysis and experimental results show the soundness and effectiveness of the proposed algorithms. The paper itself is a good read; and the algorithms will become one of the standard algorithms in LLP tasks for their simplicity and performance.
Author Feedback
Author rebuttal: We would like to thank the reviewers for their constructive comments. As per “Assigned_Reviewer_20”’s comments, we are open to any suggestion that would optimise the paper’s content and its presentation at the conference, for example regarding the impact of our results on the rapidly growing concerns for privacy protection (as jointly underlined by Reviewers “Assigned_Reviewer_3” and “Assigned_Reviewer_44”).